# A Neural Collapse Perspective on Feature Evolution in Graph Neural Networks

**Vignesh Kothapalli** *
New York University

**Tom Tirer**
Bar-Ilan University

**Joan Bruna**
New York University

## Abstract

Graph neural networks (GNNs) have become increasingly popular for classification tasks on graph-structured data. Yet, the interplay between graph topology and feature evolution in GNNs is not well understood. In this paper, we focus on node-wise classification, illustrated with community detection on stochastic block model graphs, and explore the feature evolution through the lens of the "Neural Collapse" (NC) phenomenon. When training instance-wise deep classifiers (e.g. for image classification) beyond the zero training error point, NC demonstrates a reduction in the deepest features' within-class variability and an increased alignment of their class means to certain symmetric structures. We start with an empirical study that shows that a decrease in within-class variability is also prevalent in the node-wise classification setting, however, not to the extent observed in the instance-wise case. Then, we theoretically study this distinction. Specifically, we show that even an "optimistic" mathematical model requires that the graphs obey a strict structural condition in order to possess a minimizer with exact collapse. Interestingly, this condition is viable also for heterophilic graphs and relates to recent empirical studies on settings with improved GNNs' generalization. Furthermore, by studying the gradient dynamics of the theoretical model, we provide reasoning for the partial collapse observed empirically. Finally, we present a study on the evolution of within- and between-class feature variability across layers of a well-trained GNN and contrast the behavior with spectral methods.

## 1 Introduction

Graph neural networks [52] employ message-passing mechanisms to capture intricate topological relationships in data and have become de-facto standard architectures to handle data with non-Euclidean geometric structure [11, 12, 22, 24, 33, 60, 63, 66, 70]. However, the influence of topological information on feature learning in GNNs is yet to be fully understood [40, 64, 69, 72].

In this paper, we study the feature evolution in GNNs in a node-wise supervised classification setting. In order to gain insights into the role of topology, we focus on the controlled environment of the prominent stochastic block model (SBM) [1–3, 29, 45]. The SBM provides an effective framework to control the level of sparsity, homophily, and heterophily in the random graphs and facilitates analysis of GNN which relies solely on structural information [8, 16, 32, 40, 42, 49]. While inductive supervised learning on graphs is a relatively more difficult problem than transductive learning, it aligns with practical scenarios where nodes need to be classified in unseen graphs [24], and is also amenable to training GNNs that are deeper than conventional shallow Graph Convolution Network (GCN) models [14, 16, 32, 35, 47, 62, 62, 70].

The empirical and theoretical study of GNNs' feature evolution in this paper employs a "Neural Collapse" perspective [48]. When training Deep Neural Networks (DNNs) for classification, it

---

*Correspondence to: Vignesh Kothapalli (vk2115@nyu.edu)

37th Conference on Neural Information Processing Systems (NeurIPS 2023).

is common to continue optimizing the networks' parameters beyond the zero training error point [9, 28, 39], a stage that was referred to in [48] as the "terminal phase of training" (TPT). Papyan, Han, and Donoho [25, 48] have empirically shown that a phenomenon, dubbed Neural Collapse (NC), occurs during the TPT of plain DNNs[2] on standard instance-wise classification datasets. NC encompasses several simultaneous properties: (NC1) The within-class variability of the deepest features decreases (i.e., outputs of the penultimate layer for training samples from the same class tend to their mean); (NC2) After subtracting their global mean, the mean features of different classes become closer to a geometrical structure known as a simplex equiangular tight frame; (NC3) The last layer's weights exhibit alignment with the classes' mean features. A consequence of NC1-3 is that the classifier's decision rule becomes similar to the nearest class center in the feature space. We refer to [34] for a review on this topic.

The common approach to theoretically study the NC phenomenon is the "Unconstrained Features Model" (UFM) [31, 43]. The core idea behind this "optimistic" mathematical model is that the deepest features are considered to be freely optimizable. This idea has facilitated a recent surge of theoretical works in an effort to understand the global optimality conditions and gradient dynamics of these features and the last layer's weights in DNNs [18, 25, 37, 43, 54–56, 61, 65, 71, 74]. In our work, we extend NC analysis to settings where relational information in data is paramount, and creates a tension with the 'freeness' associated with the UFM model. In essence, we highlight the key differences when analyzing NC in GNNs by identifying structural conditions on the graphs, under which the global minimizers of the training objective exhibit full NC1. Interestingly, the structural conditions that we rigorously establish in this paper are aligned with the neighborhood conditions on heterophilic graphs that have been empirically hypothesized to facilitate learning by Ma et al. [40].

Our main contributions can be summarized as follows:

- We conduct an extensive empirical study that shows that a decrease in within-class variability is prevalent also in the deepest features of GNNs trained for node classification on SBMs. However, not to the extent observed in the instance-wise setting.
- We propose and analyze a graph-based UFM to understand the role of node neighborhood patterns and their community labels on NC dynamics. We prove that even this optimistic model requires a strict structural condition on the graphs in order to possess a minimizer with exact variability collapse. Then, we show that satisfying this condition is a rare event, which theoretically justifies the distinction between observations for GNNs and plain DNNs.
- Nevertheless, by studying the gradient dynamics of the graph-based UFM, we provide theoretical reasoning for the partial collapse during GNNs training.
- Finally, we study the evolution of features across the layers of well-trained GNNs and contrast the decrease in NC1 metrics along depth with a NC1 decrease along power iterations in spectral clustering methods.

## 2 Preliminaries and Problem Setup

We focus on supervised learning on graphs for *inductive* community detection. Formally, we consider a collection of $K$ undirected graphs $\{\mathcal{G}_k = (\mathcal{V}_k, \mathcal{E}_k)\}_{k=1}^K$, each with $N$ nodes, $C$ non-overlapping balanced communities and a node labelling ground truth function $y_k : \mathcal{V}_k \to \{\mathbf{e}_1, \dots, \mathbf{e}_C\}$. Here, $\forall c \in [C], \mathbf{e}_c \in \mathbb{R}^C$ indicates the standard basis vector, where we use the notation $[C] = \{1, \cdots, C\}$. The goal is to learn a parameterized GNN model $\psi_\Theta(.)$ which minimizes the empirical risk given by:

$$\min_\Theta \frac{1}{K} \sum_{k=1}^K \mathcal{L}(\psi_\Theta(\mathcal{G}_k), y_k(\mathcal{V}_k)) + \frac{\lambda}{2} \|\Theta\|_F^2, \tag{1}$$

where $\|\cdot\|_F$ represents the Frobenius norm, $\mathcal{L}$ is the loss function that is invariant to label permutations [16], and $\lambda > 0$ is the penalty parameter. We choose $\mathcal{L}$ based on the mean squared error (MSE) as:

$$\mathcal{L}(\psi_\Theta(\mathcal{G}_k), y_k) = \min_{\pi \in S_C} \frac{1}{2N} \|\psi_\Theta(\mathcal{G}_k) - \pi(y_k(\mathcal{V}_k))\|_2^2, \tag{2}$$

where $\pi$ belongs to the permutation group over $C$ elements. Using the MSE loss for training DNN classifiers has become increasingly popular recently. For example, Hui and Belkin [30] have

---

[2]Throughout the paper, by (plain) DNNs we mean networks that output an instance-wise prediction (e.g., image class rather than pixel class), while by GNNs we mean networks that output node-wise predictions.

performed an extensive empirical study that shows that training with MSE loss yields performance that is similar to (and sometimes even better than) training with CE loss. This choice also facilitates theoretical analyses [25, 55, 71].

## 2.1 Data model

We employ the Symmetric Stochastic Block Model (SSBM) to generate graphs $\{\mathcal{G}_k = (\mathcal{V}_k, \mathcal{E}_k)\}_{k=1}^K$. Stochastic block models (originated in [29]) are classical random graph models that have been extensively studied in statistics, physics, and computer science. In the SSBM model that is considered in this paper, each graph $\mathcal{G}_k$ is associated with an adjacency matrix $\mathbf{A}_k \in \mathbb{R}^{N \times N}$, degree matrix $\mathbf{D}_k = \text{diag}(\mathbf{A}_k \mathbf{1}) \in \mathbb{R}^{N \times N}$, and a random node features matrix $\mathbf{X}_k \in \mathbb{R}^{d \times N}$, with entries sampled from a normal distribution. Formally, if $\mathbf{P} \in \mathbb{R}^{C \times C}$ represents a symmetric matrix with diagonal entries $p$ and off-diagonal entries $q$, a random graph $\mathcal{G}_k$ is considered to be drawn from the distribution $\text{SSBM}(N, C, p, q)$ if an edge between vertices $v_i, v_j$ is formed with probability $(\mathbf{P})_{y_k(v_i), y_k(v_j)}$[3]. We choose the regime of exact recovery [1–3, 45] in sparse graphs where $p = \frac{a \ln(N)}{N}, q = \frac{b \ln(N)}{N}$ for parameters $a, b \geq 0$ such that $|\sqrt{a} - \sqrt{b}| > \sqrt{C}$. The need for exact recovery (information-theoretically) stems from the requirement that $\psi_\Theta$ should be able to reach TPT (Appendix B).

## 2.2 Graph neural networks

Inspired by the widely studied model of higher-order GNNs by Morris et al. [44], we design $\psi_\Theta$ based on a family of graph operators $\mathcal{F} = \{\mathbf{I}, \widehat{\mathbf{A}}_k\}, \forall k \in [K]$, and denote it as $\psi_\Theta^{\mathcal{F}}$. Formally, for a GNN $\psi_\Theta^{\mathcal{F}}$ with $L$ layers, the node features $\mathbf{H}_k^{(l)} \in \mathbb{R}^{d_l \times N}$ at layer $l \in [L]$ is given by:

$$
\begin{aligned}
\mathbf{X}_k^{(l)} &= \mathbf{W}_1^{(l)} \mathbf{H}_k^{(l-1)} + \mathbf{W}_2^{(l)} \mathbf{H}_k^{(l-1)} \widehat{\mathbf{A}}_k, \\
\mathbf{H}_k^{(l)} &= \sigma(\mathbf{X}_k^{(l)}),
\end{aligned}
\tag{3}
$$

where $\mathbf{H}_k^{(0)} = \mathbf{X}_k$, and $\sigma(\cdot)$ represents a point-wise activation function such as ReLU. $\mathbf{W}_1^{(l)}, \mathbf{W}_2^{(l)} \in \mathbb{R}^{d_l \times d_{l-1}}$ are the weight matrices and $\widehat{\mathbf{A}}_k = \mathbf{A}_k \mathbf{D}_k^{-1}$ is the normalized adjacency matrix, also known as the random-walk matrix. We also consider a simpler family without the identity operator $\mathcal{F}' = \{\widehat{\mathbf{A}}_k\}, \forall k \in [K]$ and analyze the GNN $\psi_\Theta^{\mathcal{F}'}$ with only graph convolution functionality. Formally, the node features $\mathbf{H}_k^{(l)} \in \mathbb{R}^{d_l \times N}$ for $\psi_\Theta^{\mathcal{F}'}$ is given by:

$$
\begin{aligned}
\mathbf{X}_k^{(l)} &= \mathbf{W}_2^{(l)} \mathbf{H}_k^{(l-1)} \widehat{\mathbf{A}}_k, \\
\mathbf{H}_k^{(l)} &= \sigma(\mathbf{X}_k^{(l)}).
\end{aligned}
\tag{4}
$$

Here, the subscript for the weight matrix $\mathbf{W}_2^{(l)}$ is retained to highlight that it acts on $\mathbf{H}_k^{(l-1)} \widehat{\mathbf{A}}_k$. Finally, we employ the training strategy of Chen et al. [16] and apply instance-normalization [58] on $\sigma(\mathbf{X}_k^{(l)}), \forall l \in \{1, \cdots, L-1\}$ to prevent training instability.

## 2.3 Tracking neural collapse in GNNs

In our setup, reaching zero training error (TPT) implies that the network perfectly classifies all the nodes (up to label permutations) in all the training graphs. To this end, we leverage the NC metrics introduced in [48, 55, 56, 74] and extend them to GNNs in an inductive setting. To begin with, let us consider a single graph $\mathcal{G}_k = (\mathcal{V}_k, \mathcal{E}_k), k \in [K]$ with a normalized adjacency matrix $\widehat{\mathbf{A}}_k$. Additionally, we denote $\mathbf{H}_k^{(l)} \in \mathbb{R}^{d_l \times N}$ as the output of layer $l \in [L-1]$, irrespective of the GNN design. Now, by dropping the subscript and superscript for notational convenience, we define the class means and the global mean of $\mathbf{H}$ as follows:

$$
\overline{\mathbf{h}}_c := \frac{1}{n} \sum_{i=1}^n \mathbf{h}_{c,i} \,, \forall c \in [C], \qquad \overline{\mathbf{h}}_G := \frac{1}{Cn} \sum_{c=1}^C \sum_{i=1}^n \mathbf{h}_{c,i},
\tag{5}
$$

where $n = N/C$ represents the number of nodes in each of the $C$ balanced communities, and $\mathbf{h}_{c,i}$ is the feature vector (a column in $\mathbf{H}$) associated with $v_{c,i} \in \mathcal{V}$, i.e., the $i^{th}$ node belonging to class

---

[3]In our setup, the nodes of sampled SSBM graphs are allowed to have self-edges.

$c \in [C]$. Next, let $\mathcal{N}(v_{c,i})$ denote all the neighbors of $v_{c,i}$ and let $\mathcal{N}_{c'}(v_{c,i})$ denote only the neighbors of $v_{c,i}$ that belong to class $c' \in [C]$. We define the class means and global mean of $\mathbf{H}\widehat{\mathbf{A}}$, which is unique to the GNN setting as follows:

$$\overline{\mathbf{h}}_c^{\mathcal{N}} := \frac{1}{n} \sum_{i=1}^n \mathbf{h}_{c,i}^{\mathcal{N}} \ , \forall c \in [C], \qquad \overline{\mathbf{h}}_G^{\mathcal{N}} := \frac{1}{Cn} \sum_{c=1}^C \sum_{i=1}^n \mathbf{h}_{c,i}^{\mathcal{N}}, \tag{6}$$

where $\mathbf{h}_{c,i}^{\mathcal{N}} = \left( \sum_{v_{c,j} \in \mathcal{N}_c(v_{c,i})} \mathbf{h}_{c,j} + \sum_{v_{c',j} \in \mathcal{N}_{c' \neq c}(v_{c,i})} \mathbf{h}_{c',j} \right) / |\mathcal{N}(v_{c,i})|$.

• **Variability collapse in features H**: For a given features matrix $\mathbf{H}$, let us define the within- and between-class covariance matrices, $\mathbf{\Sigma}_W(\mathbf{H})$ and $\mathbf{\Sigma}_B(\mathbf{H})$, as:

$$\mathbf{\Sigma}_W(\mathbf{H}) := \frac{1}{Cn} \sum_{c=1}^C \sum_{i=1}^n \left( \mathbf{h}_{c,i} - \overline{\mathbf{h}}_c \right) \left( \mathbf{h}_{c,i} - \overline{\mathbf{h}}_c \right)^\top, \tag{7}$$

$$\mathbf{\Sigma}_B(\mathbf{H}) := \frac{1}{C} \sum_{c=1}^C \left( \overline{\mathbf{h}}_c - \overline{\mathbf{h}}_G \right) \left( \overline{\mathbf{h}}_c - \overline{\mathbf{h}}_G \right)^\top. \tag{8}$$

To empirically track the within-class variability collapse with respect to the between-class variability, we define two NC1 metrics:

$$\mathcal{NC}_1(\mathbf{H}) = \frac{1}{C} \mathrm{Tr} \left( \mathbf{\Sigma}_W(\mathbf{H}) \mathbf{\Sigma}_B^\dagger(\mathbf{H}) \right), \qquad \widetilde{\mathcal{NC}}_1(\mathbf{H}) = \frac{\mathrm{Tr}\left( \mathbf{\Sigma}_W(\mathbf{H}) \right)}{\mathrm{Tr}\left( \mathbf{\Sigma}_B(\mathbf{H}) \right)}, \tag{9}$$

where $^\dagger$ denotes the Moore-Penrose pseudo-inverse and $\mathrm{Tr}(\cdot)$ denotes the trace of a matrix. Although $\mathcal{NC}_1$ is the original NC1 metric used by Papyan et al. [48], we consider also $\widetilde{\mathcal{NC}}_1$, which has been proposed by Tirer et al. [56] as an alternative metric that is more amenable to theoretical analysis.

• **Variability collapse in neighborhood-aggregated features H$\widehat{\mathbf{A}}$**: Similarly to the above, we track the within- and between-class variability of the "neighborhood-aggregated" features matrix $\mathbf{H}\widehat{\mathbf{A}}$ by $\mathbf{\Sigma}_W(\mathbf{H}\widehat{\mathbf{A}})$ and $\mathbf{\Sigma}_B(\mathbf{H}\widehat{\mathbf{A}})$ (computed using $\overline{\mathbf{h}}_c^{\mathcal{N}}$ and $\overline{\mathbf{h}}_G^{\mathcal{N}}$), as well as $\mathcal{NC}_1(\mathbf{H}\widehat{\mathbf{A}})$ and $\widetilde{\mathcal{NC}}_1(\mathbf{H}\widehat{\mathbf{A}})$. (See Appendix C for formal definitions.) Finally, we follow a simple approach and track the mean and variance of $\mathcal{NC}_1(\mathbf{H}), \widetilde{\mathcal{NC}}_1(\mathbf{H}), \mathcal{NC}_1(\mathbf{H}\widehat{\mathbf{A}}), \widetilde{\mathcal{NC}}_1(\mathbf{H}\widehat{\mathbf{A}})$ across all $K$ graphs in our experiments.

As the primary focus of our paper is the analysis of feature variability during training and inference, we defer the definition and examination of metrics based on NC2 and NC3 to Appendix C, H.

## 3 Evolution of penultimate layer features during training

In this section, we explore the evolution of the deepest features of GNNs during training. In Section 3.1, we present empirical results of GNNs in the setup that is detailed in Section 2, showing that a decrease in within-class feature variability is present in GNNs that reach zero training error, but not to the extent observed with plain DNNs. Then, in Section 3.2, we theoretically study a mathematical model that provides reasoning for the empirical observations.

### 3.1 Experiments

**Setup.** We focus on the training performance of GNNs $\psi_\Theta^{\mathcal{F}}, \psi_\Theta^{\mathcal{F}'}$ on sparse graphs and generate a dataset of $K = 1000$ random SSBM graphs with $C = 2, N = 1000, p = 0.025, q = 0.0017$. The networks $\psi_\Theta^{\mathcal{F}}, \psi_\Theta^{\mathcal{F}'}$ are composed of $L = 32$ layers with graph operator, ReLU activation, and instance-normalization functionality. The hidden feature dimension is set to $8$ across layers. They are trained for $8$ epochs using stochastic gradient descent (SGD) with a learning rate $0.004$, momentum $0.9$, and a weight decay of $5 \times 10^{-4}$. During training, we track the NC1 metrics for the penultimate layer features $\mathbf{H}_k^{(L-1)}$, by computing their mean and standard deviation across $k \in [K]$ graphs after every epoch. To measure the performance of the GNN, we compute the 'overlap' [16] between predicted communities and ground truth communities (up to permutations):

$$\mathrm{overlap}(\hat{y}, y) := \max_{\pi \in S_C} \left( \frac{1}{N} \sum_{i=1}^N \delta_{\hat{y}(v_i), \pi(y(v_i))} - \frac{1}{C} \right) / \left( 1 - \frac{1}{C} \right) \tag{10}$$

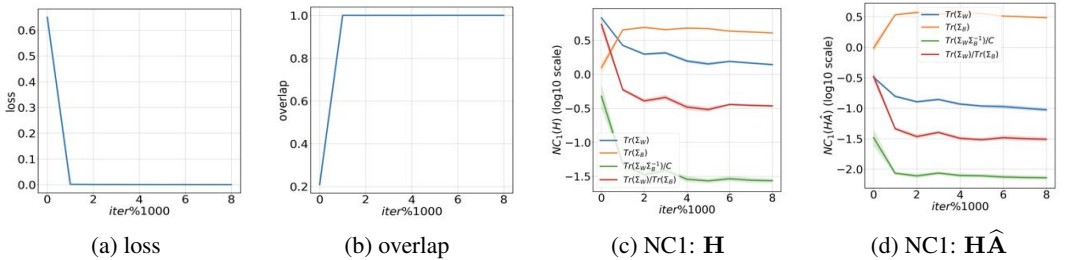

Figure 1: GNN $\psi_\Theta^{\mathcal{F}}$: Illustration of loss, overlap, and $\mathcal{NC}_1$ plots for $\mathbf{H}$, $\mathbf{H}\widehat{\mathbf{A}}$ during training.

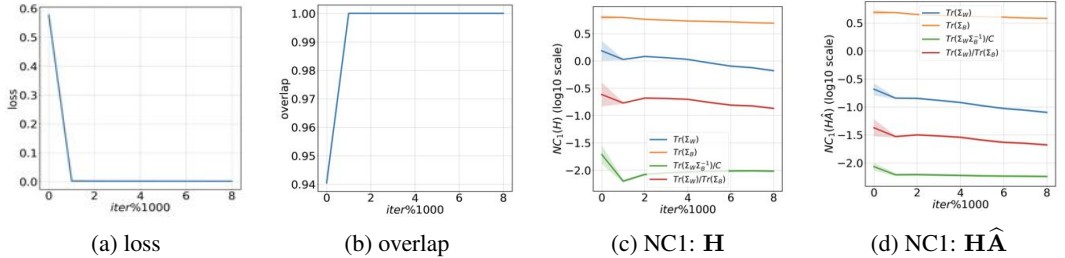

Figure 2: GNN $\psi_\Theta^{\mathcal{F}'}$: Illustration of loss, overlap, and $\mathcal{NC}_1$ plots for $\mathbf{H}$, $\mathbf{H}\widehat{\mathbf{A}}$ during training.

where $\hat{y}$ is the node labelling function based on GNN design and $\frac{1}{N} \sum_{i=1}^{N} \delta_{\hat{y}(v_i), \pi(y(v_i))}$ is the training accuracy ($\delta$ denotes the Kronecker delta). The overlap allows us to measure the improvements in performance over random guessing while retaining the indication that the GNN has reached TPT. Formally, when $\frac{1}{N} \sum_{i=1}^{N} \delta_{\hat{y}(v_i), \pi(y(v_i))} = 1$ (zero training error), then $\text{overlap}(\hat{y}, y) = 1$. We illustrate the empirical results in Figures 1 and 2, and present extensive experiments (showing similar behavior) along with infrastructure details in Appendix H[4].

**Observation:** The key takeaway is that $\mathcal{NC}_1(\mathbf{H}_k^{(L-1)})$, $\widetilde{\mathcal{NC}}_1(\mathbf{H}_k^{(L-1)})$ tend to reduce and plateau during TPT in $\psi_\Theta^{\mathcal{F}}$ and $\psi_\Theta^{\mathcal{F}'}$. Notice that even though we consider a controlled SSBM-based setting, the $\mathcal{NC}_1$ values observed here are higher than the values observed in the case of plain DNNs on real-world instance-wise datasets [48, 74]. Additionally, we can observe that trends for $\mathcal{NC}_1(\mathbf{H}_k^{(L-1)}\widehat{\mathbf{A}}_k)$, $\widetilde{\mathcal{NC}}_1(\mathbf{H}_k^{(L-1)}\widehat{\mathbf{A}}_k)$ are similar to those of $\mathcal{NC}_1(\mathbf{H}_k^{(L-1)})$, $\widetilde{\mathcal{NC}}_1(\mathbf{H}_k^{(L-1)})$.

### 3.2 Theoretical analysis

In this section, we provide a theory for this empirical behavior. Most, if not all, of the theoretical papers on NC, adopt the UFM approach, which treats the features as free optimization variables – disconnected from data [18, 25, 43, 55, 56, 74]. Here, we consider a graph-based adaptation of this approach, that we dubbed as gUFM. We consider GNNs of the form of $\psi_\Theta^{\mathcal{F}'}$, which is more tractable for mathematical analysis. Formally, by considering $\mathcal{L}$ to be the MSE loss, treating $\{\mathbf{H}_k^{(L-1)}\}_{k=1}^{K}$ as freely optimizable variables, and representing $\mathbf{W}_2^{(L)} \in \mathbb{R}^{C \times d_{L-1}}$, $\mathbf{H}_k^{(L-1)} \in \mathbb{R}^{d_{L-1} \times N}$ as $\mathbf{W}_2, \mathbf{H}_k$ (for notational convenience), the empirical risk based on the gUFM can be formulated as follows:

$$\widehat{\mathcal{R}}^{\mathcal{F}'}(\mathbf{W}_2, \{\mathbf{H}_k\}_{k=1}^{K}) := \frac{1}{K} \sum_{k=1}^{K} \left( \frac{1}{2N} \left\| \mathbf{W}_2 \mathbf{H}_k \widehat{\mathbf{A}}_k - \mathbf{Y} \right\|_F^2 + \frac{\lambda_{H_k}}{2} \left\| \mathbf{H}_k \right\|_F^2 \right) + \frac{\lambda_{W_2}}{2} \left\| \mathbf{W}_2 \right\|_F^2 \tag{11}$$

where $\mathbf{Y} \in \mathbb{R}^{C \times N}$ is the target matrix, which is composed of one-hot vectors associated with the different classes, and $\lambda_{W_2}, \lambda_{H_k} > 0$ are regularization hyperparameters. To simplify the analysis, let us assume that $\mathbf{Y} = \mathbf{I}_C \otimes \mathbf{1}_n^\top$, where $\otimes$ denotes the Kronecker product. Namely, the training data is balanced (a common assumption in UFM-based analyses in literature) with $n = N/C$ nodes per

---

[4]Code is available at: https://github.com/kvignesh1420/gnn_collapse

class in each graph and (without loss of generality) organized class-by-class. Note that for $K = 1$ (which allows omitting the graph index $k$) and no graphical structure, i.e., $\widehat{\mathbf{A}} = \mathbf{I}$ (since $\mathbf{A} = \mathbf{I}$), (11) reduces to the plain UFM that has been studied in [25, 55, 71]. In this case, it has been shown that any minimizer $(\mathbf{W}_2^*, \mathbf{H}^*)$ is *collapsed*, i.e., its features have *exactly zero* within-class variability:

$$\mathbf{h}_{c,1}^* = \cdots = \mathbf{h}_{c,n}^* = \overline{\mathbf{h}}_c^*, \quad \forall c \in [C], \tag{12}$$

which implies $\boldsymbol{\Sigma}_W(\mathbf{H}^*) = \mathbf{0}$. We will show now that the situation in gUFM is significantly different.

Considering the $K = 1$ case, we start by showing that, to have minimizers of (11) that possess the property in (12), the graph must obey a strict structural condition. For $K > 1$, having a minimizer $(\mathbf{W}_2^*, \{\mathbf{H}_k^*\})$ where, for some $j \in [K]$, $\mathbf{H}_j^*$ is collapsed directly follows from having the structural condition satisfied by the $j$-th graph (as shown in our proof, the sufficiency of the condition does not depend on the shared weights $\mathbf{W}_2$). On the other hand, generalizing the necessity of the structural condition to the case of $K > 1$ is technically challenging (see the appendix for details). For that reason, we state the condition in the following theorem only for $K = 1$. Note also that, showing that the condition is unlikely to be satisfied per graph is enough for explaining the plateaus above zero of NC metrics (computed over multiple graphs), which are demonstrated in Section 3.1.

**Theorem 3.1.** *Consider the gUFM in* (11) *with $K = 1$ and denote the fraction of neighbors of node $v_{c,i}$ that belong to class $c'$ as $s_{cc',i} = \frac{|\mathcal{N}_{c'}(v_{c,i})|}{|\mathcal{N}(v_{c,i})|}$. Let the condition **C** based on $s_{cc',i}$ be given by:*

$$(s_{c1,1}, \cdots, s_{cC,1}) = \cdots = (s_{c1,n}, \cdots, s_{cC,n}), \quad \forall c \in [C]. \tag{C}$$

*If a graph $\mathcal{G}$ satisfies condition **C**, then there exist minimizers of the gUFM that are collapsed (satisfying (12)). Conversely, when either $\sqrt{\lambda_H \lambda_{W_2}} = 0$, or $\sqrt{\lambda_H \lambda_{W_2}} > 0$ and $\mathcal{G}$ is regular (so that $\widehat{\mathbf{A}} = \widehat{\mathbf{A}}^\top$), if there exists a collapsed non-degenerate minimizer[5] of gUFM, then condition **C** necessarily holds.*

**Remark:** The proof is presented in Appendix D. The symmetry assumption on $\widehat{\mathbf{A}}$ (which implies that $\mathcal{G}$ is a regular graph) in the second part of the theorem has been made to pass technical obstacles in the proof rather than due to a true limitation. Thus, together with the results of our experiments (where no symmetry is enforced), we believe that this assumption can be dropped. Accordingly, we state the following conjecture.

**Conjecture 3.1.** *Consider the gUFM in* (11) *with $K = 1$ and condition **C** as stated in theorem 3.1. The minimizers of the gUFM are collapsed (satisfying (12)) iff the graph $\mathcal{G}$ satisfies condition **C**.*

Let us dwell on the implication of Theorem 3.1. The stated condition **C** essentially holds when any node $i \in [n]$ of a certain class $c$ obeys $(s_{c1,i}, \cdots, s_{cC,i}) = (s_{c1}, \cdots, s_{cC})$ for some $(s_{c1}, \cdots, s_{cC})$, a tuple of the ratio of neighbors ($\sum_{c'=1}^{C} s_{cc'} = 1$) independent of $i$. That is, $(s_{c1}, \cdots, s_{cC})$ must be the same for nodes within the same class but can be different for nodes belonging to different classes. For example, for a plain UFM this condition trivially holds, as $\widehat{\mathbf{A}} = \mathbf{I}$. Under the SSBM distribution, it is also easy to see that $\mathbb{E}\widehat{\mathbf{A}}$ satisfies this condition. However, for more practical graphs, such as those *drawn* from SSBM, the probability of having a graph that obeys condition **C** is negligible. This is shown in the following theorem.

**Theorem 3.2.** *Let $\mathcal{G} = (\mathcal{V}, \mathcal{E})$ be drawn from SSBM$(N, C, p, q)$. For $N >> C$, we have*

$$\mathbb{P}(\mathcal{G} \text{ obeys } \boldsymbol{C}) < \left(\sum_{t=0}^{n} \left[\binom{n}{t} q^t (1-q)^{n-t}\right]^n\right)^{\frac{C(C-1)}{2}}. \tag{13}$$

The proof is presented in Appendix E. It is not hard to see that as the number of per-class nodes $n$ increases, the probability of satisfying condition **C** decreases,[6] as numerically exemplified below.

**Numerical example.** Let's consider a setting with $C = 2, N = 1000, a = 3.75, b = 0.25$. This gives us $n = N/C = 500, p = 0.025, q = 0.0017$, for which $\mathbb{P}(\mathcal{G} \text{ obeys } \boldsymbol{C}) < 2.18 \times 10^{-188}$.

In Appendix E we further show by exhaustive computation of $\mathbb{P}(\mathcal{G} \text{ obeys } \boldsymbol{C})$ that its value is negligible even for smaller scale graphs. Thus, the probability of sampling a graph structure for which the gUFM minimizers exhibit exact collapse is practically 0.

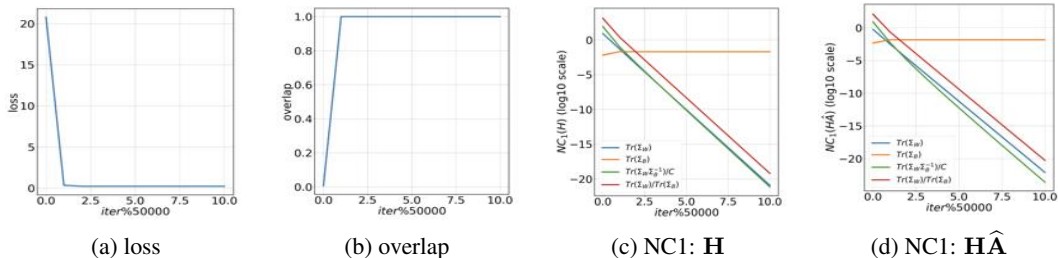

(a) loss      (b) overlap      (c) NC1: $\mathbf{H}$      (d) NC1: $\mathbf{H}\widehat{\mathbf{A}}$

Figure 3: gUFM for $\psi_{\Theta}^{\mathcal{F}'}$: Illustration of loss, overlap, and $\mathcal{NC}_1$ plots for $\mathbf{H}, \mathbf{H}\widehat{\mathbf{A}}$ during training on 10 SSBM graphs satisfying condition $\mathbf{C}$.

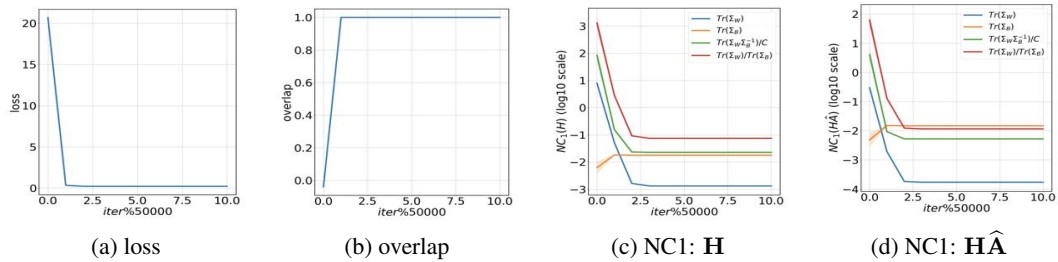

(a) loss      (b) overlap      (c) NC1: $\mathbf{H}$      (d) NC1: $\mathbf{H}\widehat{\mathbf{A}}$

Figure 4: gUFM for $\psi_{\Theta}^{\mathcal{F}'}$: Illustration of loss, overlap, and $\mathcal{NC}_1$ plots for $\mathbf{H}, \mathbf{H}\widehat{\mathbf{A}}$ during training on 10 SSBM graphs which do not satisfy condition $\mathbf{C}$.

**gUFM experiments.** For a better understanding of these results, we present small-scale experiments using the gUFM model on graphs that satisfy and do not satisfy condition $\mathbf{C}$. By training the gUFM (based on $\psi_{\Theta}^{\mathcal{F}'}$) on $K = 10$ graphs that satisfy condition $\mathbf{C}$, we can observe from Figure 3 that NC1 metrics on $\mathbf{H}, \mathbf{H}\widehat{\mathbf{A}}$ reduce significantly. On the other hand, these metrics plateau after sufficient reduction when the graphs fail to satisfy condition $\mathbf{C}$, as shown in Figure 4. In both the cases, the SSBM parameters are $C = 2, N = 1000, p = 0.025, q = 0.0017$, and the gUFM is trained using plain gradient descent for 50000 epochs with a learning rate of 0.1 and L2 regularization parameters $\lambda_{W_1} = \lambda_{W_2} = \lambda_H = 5 \times 10^{-3}$. Extensive experiments with varying choices of $N, C, p, q$, feature transformation based on $\psi_{\Theta}^{\mathcal{F}}$ and additional NC metrics are provided in Appendix H. (The additional NC metrics measure the alignment of the classes' mean features with simplex Equiangular Tight Frame (ETF) and Orthogonal Frame (OF) structures.)

**Remark.** Note that previous papers consider UFM configurations for which the minimizers possess exact NC, typically without any condition on the number of samples or on the hyperparameters of the settings. As the UFMs are "optimistic" models, in the sense that they ignore all the limitations on modifying the features that exist in the training of practical DNNs, such results can be understood as "zero-order" reasoning for practical NC behavior. On the other hand, here we show that even the optimistic gUFM will not yield perfectly collapsed minimizers for graph structures that are not rare. This provides a purer understanding of the gaps in GNNs' features from exact collapse and why these gaps are larger than for plain DNNs. We also highlight the observation that *condition $\mathbf{C}$ applies to homophilic as well as heterophilic graphs*, as the constraint on neighborhood ratios is independent of label similarity. Thus providing insights on the effectiveness of GNNs on highly heterophilic graphs as empirically observed by Ma et al. [40].

**Gradient flow:** By now, we have provided a theory for the distinction between the deepest features of GNNs and plain DNNs. Next, to provide reasoning for the partial collapse in GNNs, which is observed empirically, we turn to study the gradient dynamics of our gUFM.

We consider the $K = 1$ case and, following the common practice [25, 56], analyze the gradient flow along the "central path" — i.e., when $\mathbf{W}_2 = \mathbf{W}_2^*(\mathbf{H})$ is the optimal minimizer of $\widehat{\mathcal{R}}^{\mathcal{F}'}(\mathbf{W}_2, \mathbf{H})$

---

[5]Non-degenerate minimizers in the sense that $\mathbf{W}_2^*\mathbf{H}^* \in \mathbb{R}^{C \times N}$ is full-rank. This eliminates degenerate 'zero'-solutions which are obtained when the regularization hyper-parameters are large.

[6]Each term in each of the sums is the $n^{th}$ power of a number smaller than 1 (a binomial probability).

w.r.t. $\mathbf{W}_2$, which has a closed-form expression as a function of $\mathbf{H}$. The resulting gradient flow is:

$$\frac{d\mathbf{H}_t}{dt} = -\nabla \widehat{\mathcal{R}}^{\mathcal{F}'}(\mathbf{W}_2^*(\mathbf{H}_t), \mathbf{H}_t). \tag{14}$$

Similarly to [25, 56], we aim to gain insights on the evolution of $\mathbf{\Sigma}_W(\mathbf{H}_t)$ and $\mathbf{\Sigma}_B(\mathbf{H}_t)$ (in particular, their traces) along this flow. Yet, the presence of the structure matrix $\widehat{\mathbf{A}}$ significantly complicates the analysis compared to existing works (which are essentially restricted to $\widehat{\mathbf{A}} = \mathbf{I}$). Accordingly, we focus on the case of two classes, $C = 2$, and adopt a perturbation approach, analyzing the flow for a graph $\widehat{\mathbf{A}} = \mathbb{E}\widehat{\mathbf{A}} + \mathbf{E}$, where the expectation is taken with respect to the SSBM distribution and $\mathbf{E}$ is a sufficiently small perturbation matrix. Our results are stated in the following theorem.

**Theorem 3.3.** *Let $K = 1$, $C = 2$ and $\lambda_{W_2} > 0$. There exist $\alpha > 0$ and $E > 0$, such that for $0 < \lambda_H < \alpha$ and $0 < \|\mathbf{E}\| < E$, along the gradient flow stated in (14) associated with the graph $\widehat{\mathbf{A}} = \mathbb{E}\widehat{\mathbf{A}} + \mathbf{E}$, we have that: (1) $\mathrm{Tr}(\mathbf{\Sigma}_W(\mathbf{H}_t))$ decreases, and (2) $\mathrm{Tr}(\mathbf{\Sigma}_B(\mathbf{H}_t))$ increases. Accordingly, $\widehat{\mathcal{NC}}_1(\mathbf{H}_t)$ decreases.*

The proof is presented in Appendix F. The importance of the theorem comes from showing that even graphs that do not satisfy condition **C** (in the context of the analysis: perturbations around $\mathbb{E}\widehat{\mathbf{A}}$) exhibit reduction in the within-class covariance and increase in the between-class covariance of the features. This implies a reduction of NC1 metrics (to some extent), which is aligned with the empirical results in Section 3.1. Additionally, we highlight that since an increase in $\mathrm{Tr}(\mathbf{\Sigma}_B(\mathbf{H}_t))$ and decrease in $\mathrm{Tr}(\mathbf{\Sigma}_W(\mathbf{H}_t))$ is desirable for NC, this behavior of the penultimate layer's features can potentially serve as a remedy for the over-smoothing problem in GNNs (more details in Appendix A).

## 4 Feature separation across layers during inference

Till now, we have analyzed the feature evolution of the deepest GNN layer during training. In this section, we use these well-trained GNNs to classify nodes in unseen SSBM graphs and explore the depthwise evolution of features. In essence, we take an NC perspective on characterizing the weights of these well-trained networks that facilitate good generalization. To this end, we present empirical results demonstrating a gradual decrease of NC1 metrics along the network's depth. The observations hold a resemblance to the case with plain DNNs (shown empirically in [20, 55] and more recently in [26], and theoretically in [56]). To gain insights into this depthwise behavior we also compare it with the behavior of spectral clustering methods along their projected power iterations.

### 4.1 Experiments

**Setup.** We consider the $32-$layered networks $\psi_\Theta^{\mathcal{F}}, \psi_\Theta^{\mathcal{F}'}$ which have been designed and trained as per the setup in section 3.1 and have reached TPT. These networks are now tested on a dataset of $K = 100$ unseen random SSBM graphs with $C = 2, N = 1000, p = 0.025, q = 0.0017$. Additionally, we perform spectral clustering using projected power iterations on the Normalized Laplacian (NL) and Bethe-Hessian (BH) matrices [51] for each of the test graphs. The motivation behind this approach is to obtain an approximation of the Fiedler vector of NL/BH that sheds light on the hidden community structure [1, 4, 46, 67]. Formally, for a test graph $\mathcal{G} = (\mathcal{V}, \mathcal{E})$, the NL and BH matrices are given by:

$$\mathrm{NL}(\mathcal{G}) = \mathbf{I} - \mathbf{D}^{-1/2}\mathbf{A}\mathbf{D}^{-1/2}, \tag{15}$$

$$\mathrm{BH}(\mathcal{G}, r) = (r^2 - 1)\mathbf{I} - r\mathbf{A} + \mathbf{D}, \tag{16}$$

where $r \in \mathbb{R}$ is the BH scaling factor. Now, by treating $\mathbf{B}$ to be either NL or BH matrix, a projected power iteration to estimate the second largest eigenvector of $\widetilde{\mathbf{B}} = \|\mathbf{B}\| \mathbf{I} - \mathbf{B}$ is given by:

$$\mathbf{x}^{(l)} = \widetilde{\mathbf{B}}\mathbf{w}^{(l-1)}, \quad \text{where} \quad \mathbf{w}^{(l-1)} = \frac{\mathbf{x}^{(l-1)} - \langle \mathbf{x}^{(l-1)}, \mathbf{v}\rangle \mathbf{v}}{\left\|\mathbf{x}^{(l-1)} - \langle \mathbf{x}^{(l-1)}, \mathbf{v}\rangle \mathbf{v}\right\|_2}, \tag{17}$$

with the vector $\mathbf{v} \in \mathbb{R}^N$ denoting the largest eigenvector of $\widetilde{\mathbf{B}}$. Thus, we start with a random normal vector $\mathbf{w}^0 \in \mathbb{R}^N$ and iteratively compute the feature vector $\mathbf{x}^{(l)} \in \mathbb{R}^N$, which represents the 1-D feature for each node after $l$ iterations[7].

---

[7]Interestingly, the connection between message passing in GNNs and power iterations *without the normalization* has been explored in [36]. However, projection and normalization are paramount to our setup (with random features) for approximating the Fiedler vector.

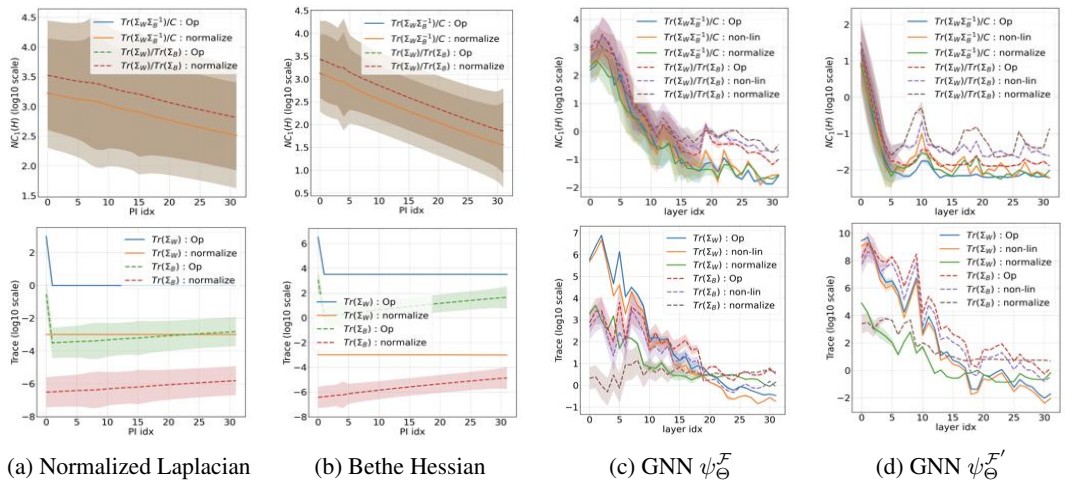

Figure 5: $\mathcal{NC}_1(\mathbf{H}), \widetilde{\mathcal{NC}}_1(\mathbf{H})$ metrics (top) and traces of covariance matrices (bottom) across projected power iterations for NL and BH (a,b), and across layers for GNNs $\psi_\Theta^{\mathcal{F}}$ and $\psi_\Theta^{\mathcal{F}'}$ (c,d).

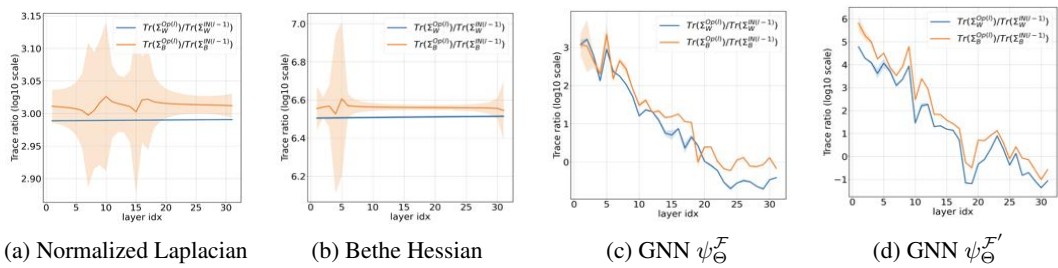

Figure 6: Ratio of traces of covariance matrices across projected power iterations for NL and BH (a,b), and across layers for GNNs $\psi_\Theta^{\mathcal{F}}$ and $\psi_\Theta^{\mathcal{F}'}$ (c,d).

### 4.2 Towards understanding depthwise behavior

From Figure 5, we can observe that the rate of decrease in NC1 metrics is much higher in $\psi_\Theta^{\mathcal{F}}$ and $\psi_\Theta^{\mathcal{F}'}$ (avg test overlap $= 1$) when compared to the baseline spectral approaches (avg test overlap NL$= 0.04$, BH$= 0.15$) with random normal feature initialization. For $\psi_\Theta^{\mathcal{F}}$ and $\psi_\Theta^{\mathcal{F}'}$, the NC1 metrics and traces of covariance matrices are tracked after each of the components of a layer: graph operator, ReLU and instance normalization. For spectral methods, the components are: the operator $\widetilde{\mathbf{B}}$ and the normalization. Interestingly, this rate seems to be relatively higher in $\psi_\Theta^{\mathcal{F}'}$ than in $\psi_\Theta^{\mathcal{F}}$, and the variance of metrics tends to reduce significantly across all the test graphs after a certain depth in $\psi_\Theta^{\mathcal{F}'}$ and $\psi_\Theta^{\mathcal{F}}$. Intuitively, the presence of $\mathbf{W}_1$ in $\psi_\Theta^{\mathcal{F}}$ seems to delay this reduction across layers. On the other hand, owing to the non-parametric nature of the spectral approaches, observe that the ratios $\text{Tr}(\mathbf{\Sigma}_B(\mathbf{x}^{(l)}))/\text{Tr}(\mathbf{\Sigma}_B(\mathbf{w}^{(l-1)})), \text{Tr}(\mathbf{\Sigma}_W(\mathbf{x}^{(l)}))/\text{Tr}(\mathbf{\Sigma}_W(\mathbf{w}^{(l-1)}))$ tend to be constant throughout all iterations. However, the GNNs behave differently as $\text{Tr}(\mathbf{\Sigma}_B(\mathbf{X}^{(l)}))/\text{Tr}(\mathbf{\Sigma}_B(\mathbf{H}^{(l-1)}))$, $\text{Tr}(\mathbf{\Sigma}_W(\mathbf{X}^{(l)}))/\text{Tr}(\mathbf{\Sigma}_W(\mathbf{H}^{(l-1)}))$ tend to decrease across depth (Figure 6).

For a better understanding of this phenomenon, we consider the case of $C = 2$ (without loss of generality) and assume that the $(l-1)^{th}$-layer features $\mathbf{H}^{(l-1)}$ of nodes belonging to class $c = 1, 2$ are drawn from distributions $\mathcal{D}_1, \mathcal{D}_2$ respectively. We do not make any assumptions on the nature of the distributions and simply consider $\boldsymbol{\mu}_1^{(l-1)}, \boldsymbol{\mu}_2^{(l-1)} \in \mathbb{R}^{d_{l-1}}$ and $\mathbf{\Sigma}_1^{(l-1)}, \mathbf{\Sigma}_2^{(l-1)} \in \mathbb{R}^{d_{l-1} \times d_{l-1}}$ as their mean vectors and covariance matrices, respectively. In the following theorem, we present bounds on the ratio of traces of feature covariance matrices after the graph operator is applied.

**Theorem 4.1.** *Let* $C = 2, \lambda_i(\cdot), \lambda_{-i}(\cdot)$ *indicate the* $i^{th}$ *largest and smallest eigenvalue of a matrix,* $\beta_1 = \frac{p-q}{p+q}, \beta_2 = \frac{p}{n(p+q)}, \beta_3 = \frac{p^2+q^2}{n(p+q)^2}$, *and denote*

$$\mathbf{T}_W = \mathbf{W}_1^{*(l)\top}\mathbf{W}_1^{*(l)} + \beta_2 \left[ \mathbf{W}_2^{*(l)\top}\mathbf{W}_1^{*(l)} + \mathbf{W}_1^{*(l)\top}\mathbf{W}_2^{*(l)} \right] + \beta_3 \mathbf{W}_2^{*(l)\top}\mathbf{W}_2^{*(l)},$$

$$\mathbf{T}_B = \left( \mathbf{W}_1^{*(l)} + \beta_1\mathbf{W}_2^{*(l)} \right)^\top \left( \mathbf{W}_1^{*(l)} + \beta_1\mathbf{W}_2^{*(l)} \right).$$

*Then, the ratios of traces* $\frac{\text{Tr}(\boldsymbol{\Sigma}_B(\mathbf{X}^{(l)}))}{\text{Tr}(\boldsymbol{\Sigma}_B(\mathbf{H}^{(l-1)}))}, \frac{\text{Tr}(\boldsymbol{\Sigma}_W(\mathbf{X}^{(l)}))}{\text{Tr}(\boldsymbol{\Sigma}_W(\mathbf{H}^{(l-1)}))}$ *for layer* $l \in \{2, \cdots, L\}$ *of a network* $\psi_\Theta^{\mathcal{F}}$ *are bounded as follows:*

$$\frac{\sum_{i=1}^{d_{l-1}} \lambda_{-i}\left(\boldsymbol{\Sigma}_B(\mathbf{H}^{(l-1)})\right) \lambda_i\left(\mathbf{T}_B\right)}{\sum_{i=1}^{d_{l-1}} \lambda_i\left(\boldsymbol{\Sigma}_B(\mathbf{H}^{(l-1)})\right)} \leq \frac{\text{Tr}(\boldsymbol{\Sigma}_B(\mathbf{X}^{(l)}))}{\text{Tr}(\boldsymbol{\Sigma}_B(\mathbf{H}^{(l-1)}))} \leq \frac{\sum_{i=1}^{d_{l-1}} \lambda_i\left(\boldsymbol{\Sigma}_B(\mathbf{H}^{(l-1)})\right) \lambda_i\left(\mathbf{T}_B\right)}{\sum_{i=1}^{d_{l-1}} \lambda_i\left(\boldsymbol{\Sigma}_B(\mathbf{H}^{(l-1)})\right)},$$

$$\frac{\sum_{i=1}^{d_{l-1}} \lambda_{-i}\left(\boldsymbol{\Sigma}_W(\mathbf{H}^{(l-1)})\right) \lambda_i\left(\mathbf{T}_W\right)}{\sum_{i=1}^{d_{l-1}} \lambda_i\left(\boldsymbol{\Sigma}_W(\mathbf{H}^{(l-1)})\right)} \leq \frac{\text{Tr}(\boldsymbol{\Sigma}_W(\mathbf{X}^{(l)}))}{\text{Tr}(\boldsymbol{\Sigma}_W(\mathbf{H}^{(l-1)}))} \leq \frac{\sum_{i=1}^{d_{l-1}} \lambda_i\left(\boldsymbol{\Sigma}_W(\mathbf{H}^{(l-1)})\right) \lambda_i\left(\mathbf{T}_W\right)}{\sum_{i=1}^{d_{l-1}} \lambda_i\left(\boldsymbol{\Sigma}_W(\mathbf{H}^{(l-1)})\right)}.$$

The proof is presented in Appendix G. To understand the implications of this result, first observe that by setting $\mathbf{W}_1^* = \mathbf{0}$ and modifying $\mathbf{T}_W = \beta_3\mathbf{W}_2^{*(l)\top}\mathbf{W}_2^{*(l)}, \mathbf{T}_B = \beta_1^2\mathbf{W}_2^{*(l)\top}\mathbf{W}_2^{*(l)}$, we can obtain a similar bound formulation for $\psi_\Theta^{\mathcal{F}'}$. To this end, as $\mathbf{T}_W, \mathbf{T}_B$ depend on the spectrum of $\mathbf{W}_2^{*(l)}$, the ratios $\frac{\text{Tr}(\boldsymbol{\Sigma}_B(\mathbf{X}^{(l)}))}{\text{Tr}(\boldsymbol{\Sigma}_B(\mathbf{H}^{(l-1)}))}, \frac{\text{Tr}(\boldsymbol{\Sigma}_W(\mathbf{X}^{(l)}))}{\text{Tr}(\boldsymbol{\Sigma}_W(\mathbf{H}^{(l-1)}))}$ are highly dependent on $\beta_1, \beta_3$. Notice that since $\mathbf{W}_1^{*(l)\top}\mathbf{W}_1^{*(l)}$ in $\mathbf{T}_W$ is not scaled by any factor that is inversely dependent on $n$, it tends to act as a spectrum controlling mechanism and the reduction in within-class variability of features in $\psi_\Theta^{\mathcal{F}}$ is relatively slow when compared to $\psi_\Theta^{\mathcal{F}'}$. Thus, justifying the empirical behavior that we observed in subplots 6c and 6d in Figure 6.

## 5   Conclusion

In this work, we studied the feature evolution in GNNs for inductive node classification tasks. Adopting a Neural Collapse (NC) perspective, we analyzed both empirically and theoretically the within- and between-class variability of features along the training epochs and along the layers during inference. We showed that a partial decrease in within-class variability (and NC1 metrics) is present in the GNNs' deepest features and provided theory that indicates that greater collapse is not expected when training GNNs on practical graphs (as it requires strict structural conditions). We also showed a depthwise decrease in variability metrics, which resembles the case with plain DNNs. Especially, by leveraging the analogy of feature transformation across layers in GNNs with spectral clustering along projected power iterations, we provided insights into this GNN behavior and distinctions between two GNN architectures.

Interestingly, the structural conditions on graphs for exact collapse, which we rigorously established in this paper, are aligned with those that have been empirically hypothesized to facilitate GNNs learning in [40] (outside the context of NC). As a direction for future research, one may try to use this connection to link NC behavior with the generalization performance of GNNs. Moreover, note that a reduction in NC1 metrics of the deepest features implies not only that the within-class variability decreases but also that the between-class variability is bounded from below. Therefore, methods that are based on promoting NC (e.g., by utilizing the established structural conditions) can potentially mitigate the over-smoothing problem in GNNs. See Appendix A for a formal statement on the relation of NC and over-smoothing and additional discussions on the potential usage of graph rewiring strategies for this goal.

## Acknowledgments and Disclosure of Funding

The authors would like to thank Jonathan Niles-Weed, Soledad Villar, Teresa Huang, Zhengdao Chen, Lei Chen and the anonymous reviewers for informative discussions and feedback. The authors acknowledge the NYU High Performance Computing services for providing the computing resources to run the experiments reported in this manuscript. V.K and J.B are partially supported by NSF DMS 2134216, NSF CAREER CIF 1845360, NSF IIS 1901091, and the Alfred P Sloan Foundation. T.T is partially supported by ISF grant no. 1940/23.

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

# A  Discussion on Oversmoothing and Graph Rewiring

In this appendix, we briefly discuss oversmoothing and graph rewiring from a neural collapse perspective along with some limitations and scope for future efforts.

## A.1  Oversmoothing

Repeated message-passing operations on the node features across the depth of a GNN lead to the 'oversmoothing' phenomenon, which leads to poor node classification performance [14, 15, 32, 47, 62, 64]. We adopt the formal definition of oversmoothing by Rusch et al. [50] to our setup as follows:

**Definition A.1.** *(Oversmoothing): For an undirected, connected graph $\mathcal{G} = (\mathcal{V}, \mathcal{E})$ with $|\mathcal{V}| = N$ and $l$-th layer hidden features $\mathbf{H}^l \in \mathbb{R}^{d_l \times N}$, a function $\mu : \mathbb{R}^{d_l \times N} \to \mathbb{R}_{\geq 0}$ is called a node-similarity measure if it satisfies:*

1. *$\exists \mathbf{c} \in \mathbb{R}^{d_l}$ with $\mathbf{H}_i = \mathbf{c}$ for all nodes $i \in \mathcal{V} \iff \mu(\mathbf{H}) = 0$, for $\mathbf{H} \in \mathbb{R}^{d_l \times N}$*

2. *$\mu(\mathbf{H} + \mathbf{T}) \leq \mu(\mathbf{H}) + \mu(\mathbf{T})$, for all $\mathbf{H}, \mathbf{T} \in \mathbb{R}^{d_l \times N}$.*

*Oversmoothing with respect to $\mu$ is now defined as the layer-wise exponential convergence of the node-similarity measure $\mu$ to zero, i.e,*

$$\mu(\mathbf{H}^l) \leq C_1 e^{-C_2 l}, \text{ for } l = 1, \cdots, L \text{ with some constants } C_1, C_2 > 0.$$

Observe from the definition that as the depth $l$ increases, all the node features converge to a single feature vector $\mathbf{c}$. This implies that *both* $\mathbf{\Sigma}_W(\mathbf{H}^{L-1}), \mathbf{\Sigma}_B(\mathbf{H}^{L-1}) \to 0$. In this context, if the penultimate layer features exhibit neural collapse during training, then $\mathbf{\Sigma}_W(\mathbf{H}^{L-1})$ decreases, and $\mathbf{\Sigma}_B(\mathbf{H}^{L-1})$ is bounded from below. Thus, potentially addressing the oversmoothing problem.

To this end, in the unconstrained features setting, Theorem 3.1 indicates that the graph must satisfy a strict structural condition for having an exact neural collapse solution as a minimizer of the empirical risk. Additionally, we obtained conditions on the amount of regularization needed for $\mathbf{\Sigma}_B(\mathbf{H}^{L-1})$ to increase along the gradient flow in Theorem 3.3 (also see Appendix F). However, since we cannot expect graphs to satisfy this condition (Theorem 3.2), a potential alternative is to explore graph rewiring techniques, which are discussed below.

## A.2  Graph Rewiring

Graph rewiring techniques aim to propagate messages between nodes via computational graphs that are suitable for the task [5–7, 19, 21, 23, 24, 57]. One of the popular examples is the Personalized PageRank (PPR) and Heat kernel (HK) based diffusion on graphs by Gasteiger et al. [21]. This approach leverages the diffusion matrix to facilitate message-passing between nodes many hops apart in the actual graph. However, the graph resulting from the diffusion operations tends to be quite dense. On the other hand, recent works on addressing over-squashing in long-range dependent tasks leverage the "curvature" information for rewiring [57].

In the context of our paper, one can aim to design a rewiring technique that can modify the input graph to satisfy condition $\mathbf{C}$. In previous work, Ma et al. [40] generate synthetic graphs based on CORA by ignoring the original graph and adding edges between nodes to satisfy certain uniform neighborhood distributions. Alternatively, exploring an edge re-weighting scheme as proposed by Yan et al. [64] can also be an interesting research direction. From a scalability perspective, a neighborhood sampling technique based on condition $\mathbf{C}$ can also aid in better representation learning. Additionally, note that condition $\mathbf{C}$ is not limited to homophilic graphs and can be extended to heterophilic settings as well [38, 73], provided the sum of slices of the columns in $\hat{\mathbf{A}} \in \mathbb{R}^{N \times N}$ are the same for all nodes in a class/community (refer to the necessity condition in the proof of Theorem 3.1 in Appendix D for more details). We believe that further research in this direction can shed some light on Conjecture 3.1 and improve our understanding of the nature of global minimizers.

### A.2.1  Measuring neighborhood similarity

A key question that pops up when rewiring a graph to achieve condition $\mathbf{C}$ is a trackable metric for node neighborhoods. The 'Cross-Class Neighborhood Similarity (CCNS)' metric by Ma et al. [40] serves as a good starting point to numerically track the cosine similarity in intra-class neighborhoods and dissimilarity of inter-class neighborhoods of nodes. The definition from Ma et al. [40] is given as:

**Definition A.2.** *Cross-Class Neighborhood Similarity (CCNS): Given a graph $\mathcal{G}$ and labels $\mathbf{y}$ for all nodes, the CCNS between classes $c, c'$ is $s(c, c') = \frac{1}{|\mathcal{V}_c||\mathcal{V}_{c'}|} \sum_{i \in \mathcal{V}_c, j \in \mathcal{V}_{c'}} cos(d(i), d(j))$ where $\mathcal{V}_c$ indicates the set of nodes in class $c$ and $d(i)$ indicates the empirical histogram (over $C$ classes) of node $i's$ neighbors' labels, and the function $cos(.;.)$ measures the cosine similarity.*

A limitation of this metric is that the CCNS between two classes is measured as an average of the cosine similarity of node neighborhood histograms while failing to incorporate the variance of these neighborhood similarities. Now, note that when condition $\mathbf{C}$ in Theorem 3.1 is combined with CCNS, we can ensure that the variance of the cosine similarities is zero for any pair of classes $c, c' \in [C]$. O*verall, better metrics based on different similarity measures and the condition $\mathbf{C}$, along with efficient rewiring techniques to maximize/minimize such metrics can be a valuable future effort in the community.*

## B    A Brief Note on Exact Recovery of Planted Communities

Phase transitions in recovering the planted communities of the Stochastic Block Model (SBM) graphs have been extensively studied in the literature. In this context, 'exact recovery' indicates a perfect assignment of nodes to their respective communities/clusters. For the bi-clustering problem (i.e., $C = 2$), one can date back to the works of Bui et al. [13] for min-cut based clustering, and Boppana [10] for a spectral clustering based approach (Refer to [1, 2] for a historical perspective and additional references on this topic). Along these series of developments to find thresholds in terms of $p, q$ for the exact recovery of communities, the seminal work of Abbe et al. [3] leveraged an information theoretic perspective to identify sharp thresholds in the logarithmic degree settings. The requirement for logarithmic degree can be understood from the following observation. If we consider $p = q$, then the SBM is essentially an Erdos-Renyi (ER) random graph model with edge-connection probability $p$. Based on the results by Erdős et al. [17], if $p = \frac{c \log N}{N}$, then a randomly sampled ER graph from $ER(N, \frac{c \log N}{N})$ is connected with high probability if and only if $c > 1$ (see section 2.5 in [1]). Thus, in order to achieve exact recovery, one must ensure that the SBM graph is connected along with some over-sampling of edges. In the Symmetric SBM case, when $p = \frac{a \log N}{N}, q = \frac{b \log N}{N}$ this condition can be represented using the $|\sqrt{a} - \sqrt{b}| > \sqrt{C}$ inequality (see Theorem 13 and the following remarks in [1]). Finally, observe that from a Neural Collapse perspective, we sample SSBM graphs in this regime to ensure that a GNN can achieve zero node-classification error and reach TPT.

## C  Additional neural collapse metrics

In this appendix, we define additional NC metrics pertaining to NC1-3 for our problem setup.

• **Variability collapse in neighborhood-aggregated features $\mathbf{H}\widehat{\mathbf{A}}$:** We track the within- and between-class variability of the "neighborhood-aggregated" features matrix $\mathbf{H}\widehat{\mathbf{A}}$ by defining the covariance matrices $\boldsymbol{\Sigma}_W\left(\mathbf{H}\widehat{\mathbf{A}}\right), \boldsymbol{\Sigma}_B\left(\mathbf{H}\widehat{\mathbf{A}}\right)$ as:

$$\boldsymbol{\Sigma}_W\left(\mathbf{H}\widehat{\mathbf{A}}\right) := \frac{1}{Cn}\sum_{c=1}^{C}\sum_{i=1}^{n}\left(\mathbf{h}_{c,i}^{\mathcal{N}} - \overline{\mathbf{h}}_c^{\mathcal{N}}\right)\left(\mathbf{h}_{c,i}^{\mathcal{N}} - \overline{\mathbf{h}}_c^{\mathcal{N}}\right)^{\top}$$

$$\boldsymbol{\Sigma}_B\left(\mathbf{H}\widehat{\mathbf{A}}\right) := \frac{1}{C}\sum_{c=1}^{C}\left(\overline{\mathbf{h}}_c^{\mathcal{N}} - \overline{\mathbf{h}}_G^{\mathcal{N}}\right)\left(\overline{\mathbf{h}}_c^{\mathcal{N}} - \overline{\mathbf{h}}_G^{\mathcal{N}}\right)^{\top}$$

To this end, we define the $\mathcal{NC}_1\left(\mathbf{H}\widehat{\mathbf{A}}\right), \widetilde{\mathcal{NC}}_1\left(\mathbf{H}\widehat{\mathbf{A}}\right)$ metrics as follows:

$$\mathcal{NC}_1\left(\mathbf{H}\widehat{\mathbf{A}}\right) = \frac{1}{C}\mathrm{Tr}\left(\boldsymbol{\Sigma}_W\left(\mathbf{H}\widehat{\mathbf{A}}\right)\boldsymbol{\Sigma}_B^{\dagger}\left(\mathbf{H}\widehat{\mathbf{A}}\right)\right), \qquad \widetilde{\mathcal{NC}}_1\left(\mathbf{H}\widehat{\mathbf{A}}\right) = \frac{\mathrm{Tr}\left(\boldsymbol{\Sigma}_W\left(\mathbf{H}\widehat{\mathbf{A}}\right)\right)}{\mathrm{Tr}\left(\boldsymbol{\Sigma}_B\left(\mathbf{H}\widehat{\mathbf{A}}\right)\right)}. \quad (18)$$

Their primary purpose is to track the within-class variability of neighborhood aggregated features $\mathbf{H}\widehat{\mathbf{A}}$ relative to their between-class variability, both now dependent on the topological structure of graph $\mathcal{G}$. The motivation to track these features arises from the P2 condition in the proof of theorem 3.1 in Appendix B. Essentially, this condition states that the neighborhood aggregated features should collapse to their respective class means for the minimizer to satisfy NC.

• **SNR for variance collapse:** We track the following 'Signal-to-Noise' ratios pertaining to variability collapse of $\mathbf{H}$ and $\mathbf{H}\widehat{\mathbf{A}}$:

$$SNR(\mathcal{NC}_1) := \frac{\left\|\mathbf{W}_1\left(\overline{\mathbf{H}}\otimes\mathbf{1}_n^{\top}\right)\right\|_F}{\left\|\mathbf{W}_1\left(\mathbf{H} - \overline{\mathbf{H}}\otimes\mathbf{1}_n^{\top}\right)\right\|_F} \tag{19}$$

$$SNR(\mathcal{NC}_1^{\mathcal{N}}) := \frac{\left\|\mathbf{W}_2\left(\overline{\mathbf{H}}^{\mathcal{N}}\otimes\mathbf{1}_n^{\top}\right)\right\|_F}{\left\|\mathbf{W}_2\left(\mathbf{H}\widehat{\mathbf{A}} - \overline{\mathbf{H}}^{\mathcal{N}}\otimes\mathbf{1}_n^{\top}\right)\right\|_F} \tag{20}$$

These SNR metrics provide an alternate perspective for us to empirically analyze the desirability of variance collapse. Here $\overline{\mathbf{H}} := \begin{bmatrix}\overline{\mathbf{h}}_1 & \cdots & \overline{\mathbf{h}}_C\end{bmatrix} \in \mathbb{R}^{d_l \times C}$ and $\overline{\mathbf{H}}^{\mathcal{N}} := \begin{bmatrix}\overline{\mathbf{h}}_1^{\mathcal{N}} & \cdots & \overline{\mathbf{h}}_C^{\mathcal{N}}\end{bmatrix} \in \mathbb{R}^{d_l \times C}$ are the class-mean matrices without and with neighborhood aggregation respectively.

• **Convergence of weights to a simplex ETF:** To track the convergence of the weights $\mathbf{W}_1, \mathbf{W}_2$ to a simplex ETF structure, we define $\mathcal{NC}_2^{ETF}(\mathbf{W}_1), \mathcal{NC}_2^{ETF}(\mathbf{W}_2)$ as:

$$\mathcal{NC}_2^{ETF}(\mathbf{W}_1) := \left\|\frac{\mathbf{W}_1\mathbf{W}_1^{\top}}{\left\|\mathbf{W}_1\mathbf{W}_1^{\top}\right\|_F} - \frac{1}{\sqrt{C-1}}\left(\mathbf{I}_C - \frac{1}{C}\mathbf{1}_C\mathbf{1}_C^{\top}\right)\right\|_F \tag{21}$$

$$\mathcal{NC}_2^{ETF}(\mathbf{W}_2) := \left\|\frac{\mathbf{W}_2\mathbf{W}_2^{\top}}{\left\|\mathbf{W}_2\mathbf{W}_2^{\top}\right\|_F} - \frac{1}{\sqrt{C-1}}\left(\mathbf{I}_C - \frac{1}{C}\mathbf{1}_C\mathbf{1}_C^{\top}\right)\right\|_F \tag{22}$$

• **Convergence of weights to an Orthogonal Frame (OF):** To track the convergence of the weights $\mathbf{W}_1, \mathbf{W}_2$ to an orthogonal frame structure, we define $\mathcal{NC}_2^{OF}(\mathbf{W}_1), \mathcal{NC}_2^{OF}(\mathbf{W}_2)$ as:

$$\mathcal{NC}_2^{OF}(\mathbf{W}_1) := \left\|\frac{\mathbf{W}_1\mathbf{W}_1^{\top}}{\left\|\mathbf{W}_1\mathbf{W}_1^{\top}\right\|_F} - \frac{\mathbf{I}_C}{\sqrt{C}}\right\|_F \tag{23}$$

$$\mathcal{NC}_2^{OF}(\mathbf{W}_2) := \left\|\frac{\mathbf{W}_2\mathbf{W}_2^{\top}}{\left\|\mathbf{W}_2\mathbf{W}_2^{\top}\right\|_F} - \frac{\mathbf{I}_C}{\sqrt{C}}\right\|_F \tag{24}$$

• **Convergence of features to a simplex ETF:** To track the convergence of the features $\mathbf{H}, \mathbf{H}\widehat{\mathbf{A}}$ to a simplex ETF structure, we define $\mathcal{NC}_2^{ETF}(\mathbf{H}), \mathcal{NC}_2^{ETF}(\mathbf{H}\widehat{\mathbf{A}})$ as:

$$\mathcal{NC}_2^{ETF}(\mathbf{H}) := \left\| \frac{\widetilde{\mathbf{H}}^\top \widetilde{\mathbf{H}}}{\left\| \widetilde{\mathbf{H}}^\top \widetilde{\mathbf{H}} \right\|_F} - \frac{1}{\sqrt{C-1}} \left( \mathbf{I}_C - \frac{1}{C} \mathbf{1}_C \mathbf{1}_C^\top \right) \right\|_F \tag{25}$$

$$\mathcal{NC}_2^{ETF}(\mathbf{H}\widehat{\mathbf{A}}) := \left\| \frac{\widetilde{\mathbf{H}}^{\mathcal{N}\top} \widetilde{\mathbf{H}}^{\mathcal{N}}}{\left\| \widetilde{\mathbf{H}}^{\mathcal{N}\top} \widetilde{\mathbf{H}}^{\mathcal{N}} \right\|_F} - \frac{1}{\sqrt{C-1}} \left( \mathbf{I}_C - \frac{1}{C} \mathbf{1}_C \mathbf{1}_C^\top \right) \right\|_F \tag{26}$$

where $\widetilde{\mathbf{H}} := \left[ \overline{\mathbf{h}}_1 - \overline{\mathbf{h}}_G \ \cdots \ \overline{\mathbf{h}}_C - \overline{\mathbf{h}}_G \right] \in \mathbb{R}^{d_l \times C}$ and $\widetilde{\mathbf{H}}^{\mathcal{N}} := \left[ \overline{\mathbf{h}}_1^{\mathcal{N}} - \overline{\mathbf{h}}_G^{\mathcal{N}} \ \cdots \ \overline{\mathbf{h}}_C^{\mathcal{N}} - \overline{\mathbf{h}}_G^{\mathcal{N}} \right] \in \mathbb{R}^{d_l \times C}$ are the re-centered class-means without and with neighborhood aggregation respectively.

• **Convergence of features to an OF:** To track the convergence of the features $\mathbf{H}, \mathbf{H}\widehat{\mathbf{A}}$ to an OF structure, we define $\mathcal{NC}_2^{OF}(\mathbf{H}), \mathcal{NC}_2^{OF}(\mathbf{H}\widehat{\mathbf{A}})$ as:

$$\mathcal{NC}_2^{OF}(\mathbf{H}) := \left\| \frac{\overline{\mathbf{H}}^\top \overline{\mathbf{H}}}{\left\| \overline{\mathbf{H}}^\top \overline{\mathbf{H}} \right\|_F} - \frac{\mathbf{I}_C}{\sqrt{C}} \right\|_F \tag{27}$$

$$\mathcal{NC}_2^{OF}(\mathbf{H}\widehat{\mathbf{A}}) := \left\| \frac{\overline{\mathbf{H}}^{\mathcal{N}\top} \overline{\mathbf{H}}^{\mathcal{N}}}{\left\| \overline{\mathbf{H}}^{\mathcal{N}\top} \overline{\mathbf{H}}^{\mathcal{N}} \right\|_F} - \frac{\mathbf{I}_C}{\sqrt{C}} \right\|_F \tag{28}$$

• **Generic alignment of weights and features:** To track the alignment of $\mathbf{W}_1$ with its dual $\mathbf{H}$, we define $\mathcal{NC}_3(\mathbf{W}_1, \mathbf{H})$ as:

$$\mathcal{NC}_3(\mathbf{W}_1, \mathbf{H}) := \left\| \frac{\mathbf{W}_1}{\|\mathbf{W}_1\|_F} - \frac{\overline{\mathbf{H}}^\top}{\|\overline{\mathbf{H}}\|_F} \right\|_F \tag{29}$$

Similarly, to track the alignment of $\mathbf{W}_2$ with its dual $\mathbf{H}\widehat{\mathbf{A}}$, we define $\mathcal{NC}_3(\mathbf{W}_2, \mathbf{H}\widehat{\mathbf{A}})$ as:

$$\mathcal{NC}_3(\mathbf{W}_2, \mathbf{H}\widehat{\mathbf{A}}) := \left\| \frac{\mathbf{W}_2}{\|\mathbf{W}_2\|_F} - \frac{\overline{\mathbf{H}}^{\mathcal{N}\top}}{\|\overline{\mathbf{H}}^{\mathcal{N}}\|_F} \right\|_F \tag{30}$$

• **Alignment of weights and features with respect to simplex ETF:** To track the alignment of $\mathbf{W}_1$ and its dual $\mathbf{H}$ with respect to a simplex ETF, we define $\mathcal{NC}_3^{ETF}(\mathbf{W}_1, \mathbf{H})$ as:

$$\mathcal{NC}_3^{ETF}(\mathbf{W}_1, \mathbf{H}) := \left\| \frac{\mathbf{W}_1 \widetilde{\mathbf{H}}}{\left\| \mathbf{W}_1 \widetilde{\mathbf{H}} \right\|_F} - \frac{1}{\sqrt{C-1}} \left( \mathbf{I}_C - \frac{1}{C} \mathbf{1}_C \mathbf{1}_C^\top \right) \right\|_F \tag{31}$$

Similarly, we track the alignment of $\mathbf{W}_2$ and its dual $\mathbf{H}\widehat{\mathbf{A}}$ with respect to a simplex ETF using:

$$\mathcal{NC}_3^{ETF}(\mathbf{W}_2, \mathbf{H}\widehat{\mathbf{A}}) := \left\| \frac{\mathbf{W}_2 \widetilde{\mathbf{H}}^{\mathcal{N}}}{\left\| \mathbf{W}_2 \widetilde{\mathbf{H}}^{\mathcal{N}} \right\|_F} - \frac{1}{\sqrt{C-1}} \left( \mathbf{I}_C - \frac{1}{C} \mathbf{1}_C \mathbf{1}_C^\top \right) \right\|_F \tag{32}$$

• **Alignment of weights and features with respect to OF:** To track the alignment of $\mathbf{W}_1$ and its dual $\mathbf{H}$ with respect to an OF, we define $\mathcal{NC}_3^{OF}(\mathbf{W}_1, \mathbf{H})$ as:

$$\mathcal{NC}_3^{OF}(\mathbf{W}_1, \mathbf{H}) := \left\| \frac{\mathbf{W}_1 \overline{\mathbf{H}}}{\left\| \mathbf{W}_1 \overline{\mathbf{H}} \right\|_F} - \frac{\mathbf{I}_C}{\sqrt{C}} \right\|_F \tag{33}$$

Similarly, we track the alignment of $\mathbf{W}_2$ and its dual $\mathbf{H}\widehat{\mathbf{A}}$ with respect to an OF using:

$$\mathcal{NC}_3^{OF}(\mathbf{W}_2, \mathbf{H}\widehat{\mathbf{A}}) := \left\| \frac{\mathbf{W}_2 \overline{\mathbf{H}}^{\mathcal{N}}}{\left\| \mathbf{W}_2 \overline{\mathbf{H}}^{\mathcal{N}} \right\|_F} - \frac{\mathbf{I}_C}{\sqrt{C}} \right\|_F \tag{34}$$

# D Proof of Theorem 3.1

In this appendix, we present the proof for Theorem 3.1 by analyzing the sufficiency and necessity conditions of the graph structure given by:

$$(s_{c1,1}, \cdots, s_{cC,1}) = \cdots = (s_{c1,n}, \cdots, s_{cC,n}), \quad \forall c \in [C], \tag{C}$$

where the fraction of neighbors of node $v_{c,i}$ that belong to class $c'$ as $s_{cc',i} = \frac{|\mathcal{N}_{c'}(v_{c,i})|}{|\mathcal{N}(v_{c,i})|}$.

**Proof sketch:** Our aim is to identify the structural properties of a graph $\mathcal{G}$ (especially $\widehat{\mathbf{A}}$), such that the features $\mathbf{H}$, which exhibit neural collapse are indeed the minimizers of the risk. First, we obtain a lower bound for the risk using Jensen's inequality and show that, for a 'collapsed' $\mathbf{H}$ to be the minimizer, it is sufficient if the graph $\mathcal{G}$ satisfies condition **C**. However, the inequality is applied on a convex function of $\{\mathbf{e}_{c,i}\}$ (standard basis vectors) that is not strictly convex, and so, this analysis does not imply necessity. Thus, we show the necessity of condition **C** for a 'collapsed' $\mathbf{H}$ to be a minimizer of the risk by analyzing the optimality conditions of the stationary points. Additional details of the sketch for the 'necessity' argument are also presented.

**Sufficiency:** We begin by revisiting the risk $\widehat{\mathcal{R}}^{\mathcal{F}'}$ for $K = 1$. For simplicity, we drop the superscript $\mathcal{F}'$, subscript $k$, and treat the unconstrained features of the corresponding graph $\mathcal{G} = (\mathcal{V}, \mathcal{E})$ as $\mathbf{H}$, and denote $\widehat{\mathbf{A}} = \mathbf{A}\mathbf{D}^{-1}$. The risk $\widehat{\mathcal{R}}(\mathbf{W}_2, \mathbf{H})$ is now given by:

$$\widehat{\mathcal{R}}(\mathbf{W}_2, \mathbf{H}) := \frac{1}{2N} \left\| \mathbf{W}_2 \mathbf{H}\widehat{\mathbf{A}} - \mathbf{Y} \right\|_F^2 + \frac{\lambda_{\mathbf{H}}}{2} \|\mathbf{H}\|_F^2 + \frac{\lambda_{\mathbf{W}_2}}{2} \|\mathbf{W}_2\|_F^2 \tag{35}$$

$$=: \widehat{\mathcal{L}}(\mathbf{W}_2, \mathbf{H}) + \frac{\lambda_{\mathbf{W}_2}}{2} \|\mathbf{W}_2\|_F^2 .$$

Now, by denoting $\mathbf{e}_{c,i} \in \mathbb{R}^N$ as the one-hot vector associated with the index of the feature column $\mathbf{h}_{c,i}$ (among the $N$ feature columns), we lower bound $\widehat{\mathcal{L}}(\mathbf{W}_2, \mathbf{H})$ as follows[8]:

$$\widehat{\mathcal{L}}(\mathbf{W}_2, \mathbf{H}) = \frac{1}{2N} \left\| \mathbf{W}_2 \mathbf{H}\widehat{\mathbf{A}} - \mathbf{Y} \right\|_F^2 + \frac{\lambda_H}{2} \|\mathbf{H}\|_F^2$$

$$= \frac{1}{2N} \sum_{c=1}^{C} \sum_{i=1}^{n} \left\| \mathbf{W}_2 \mathbf{H}\widehat{\mathbf{A}}\mathbf{e}_{c,i} - \mathbf{y}_c \right\|_F^2 + \frac{\lambda_H}{2} \sum_{c=1}^{C} \sum_{i=1}^{n} \|\mathbf{H}\mathbf{e}_{c,i}\|_2^2$$

$$= \frac{1}{2N} \sum_{c=1}^{C} \frac{n}{n} \sum_{i=1}^{n} \left\| \mathbf{W}_2 \mathbf{H}\widehat{\mathbf{A}}\mathbf{e}_{c,i} - \mathbf{y}_c \right\|_F^2 + \frac{\lambda_H}{2} \sum_{c=1}^{C} \frac{n}{n} \sum_{i=1}^{n} \|\mathbf{H}\mathbf{e}_{c,i}\|_2^2 \tag{36}$$

$$\geq \frac{1}{2N} \sum_{c=1}^{C} n \left\| \mathbf{W}_2 \mathbf{H}\widehat{\mathbf{A}}\frac{1}{n} \sum_{i=1}^{n} \mathbf{e}_{c,i} - \mathbf{y}_c \right\|_F^2 + \frac{\lambda_H}{2} \sum_{c=1}^{C} n \left\| \mathbf{H}\frac{1}{n} \sum_{i=1}^{n} \mathbf{e}_{c,i} \right\|_2^2$$

$$= \frac{1}{2N} \sum_{c=1}^{C} n \left\| \mathbf{W}_2 \frac{1}{n} \sum_{i=1}^{n} \mathbf{h}_{c,i}^{\mathcal{N}} - \mathbf{y}_c \right\|_F^2 + \frac{\lambda_H}{2} \sum_{c=1}^{C} n \left\| \frac{1}{n} \sum_{i=1}^{n} \mathbf{h}_{c,i} \right\|_2^2 .$$

Note that Jensen's inequality, which we used above, is applied on a convex function of $\mathbf{e}_{c,i}$'s that is not strictly convex. Therefore, the lower-bound in equation 36 can be attained despite having different $\mathbf{e}_{c,i}$'s, e.g., when the properties **P1** and **P2**, which are stated below, hold $\forall c \in [C]$:

- **P1:** $\mathbf{h}_{c,1} = \cdots = \mathbf{h}_{c,n} = \overline{\mathbf{h}}_c$
- **P2:** $\mathbf{h}_{c,1}^{\mathcal{N}} = \cdots = \mathbf{h}_{c,n}^{\mathcal{N}} = \overline{\mathbf{h}}_c^{\mathcal{N}}$

The first property **P1** indicates zero intra-class variability of $\mathbf{H}$, i.e., $\mathbf{\Sigma}_W(\mathbf{H}) \to \mathbf{0}$, and the second property **P2** indicates zero intra-class variability of $\mathbf{H}\widehat{\mathbf{A}}$ i.e, $\mathbf{\Sigma}_W(\mathbf{H}\widehat{\mathbf{A}}) \to \mathbf{0}$.

Recall condition **C**:

---

[8]When $K > 1$, observe that $\widehat{\mathcal{R}}(\mathbf{W}_2, \{\mathbf{H}\}_{k=1}^K) = \sum_{k=1}^{K} \widehat{\mathcal{L}}(\mathbf{W}_2, \mathbf{H}_k) + \frac{\lambda_{\mathbf{W}_2}}{2} \|\mathbf{W}_2\|_F^2$. Thus, each of the $\widehat{\mathcal{L}}(\mathbf{W}_2, \mathbf{H}_k)$ terms can be lower-bounded independently, resulting in a lower-bound for $\widehat{\mathcal{R}}(\mathbf{W}_2, \{\mathbf{H}\}_{k=1}^K)$ itself.

- **C:**    $(s_{c1,1}, \cdots, s_{cC,1}) = \cdots = (s_{c1,n}, \cdots, s_{cC,n}) = (s_{c1}, \cdots, s_{cC})$,

where $(s_{c1}, \cdots, s_{cC})$ (shared by any node $i \in [n]$ in class $c$) represents any suitable tuple of the ratio of neighbors per class[9].

We just need to show that if **C** is satisfied then **H** with both **P1** and **P2** exists. Equivalently, we can assume **P1** (which is in the "feasible set" of the optimization) and show that then **C** implies **P2**. Indeed, in this case

$$\mathbf{h}_{c,i}^{\mathcal{N}} = \frac{\sum_{\mathcal{N}_c(v_{c,i})} \mathbf{h}_{c,j} + \sum_{\mathcal{N}_{c' \neq c}(v_{c,i})} \mathbf{h}_{c',j}}{|\mathcal{N}(v_{c,i})|} \tag{37}$$

$$= \frac{\sum_{\mathcal{N}_c(v_{c,i})} \overline{\mathbf{h}}_c + \sum_{\mathcal{N}_{c' \neq c}(v_{c,i})} \overline{\mathbf{h}}_{c'}}{|\mathcal{N}(v_{c,i})|} = \sum_{c'=1}^{C} s_{cc'} \overline{\mathbf{h}}_{c'}. \tag{38}$$

Therefore **P2** holds: $\mathbf{h}_{c,1}^{\mathcal{N}} = \cdots = \mathbf{h}_{c,n}^{\mathcal{N}} = \overline{\mathbf{h}}_c^{\mathcal{N}}$. Accordingly, **C** is sufficient for having **H** that obeys **P1** and **P2**, and thus minimizes the risk.

**Sketch for 'Necessity':** The goal of this analysis is to nullify the possibility of having a minimizer that exhibits collapse (i.e., **H** which satisfies **P1**) for a graph that does not satisfy condition **C**. We do so by analyzing the optimality conditions of the stationary points satisfying **P1**, and obtaining conditions on the $\widehat{\mathbf{A}}$ matrix. Specifically, we obtain a system of linear equations that is shared by all $n$ nodes within a class, which in turn leads to condition **C**. Thus, proving its necessity.

**Necessity:** Analysing the necessity of condition **C** is relatively more complicated than the sufficiency case. Nonetheless, we prove the necessity by considering $K = 1$, and by leveraging the idea of a tight-convex alternative for $\widehat{\mathcal{R}}(\mathbf{W}_2, \mathbf{H})$ as been used in [55, 74] for the conventional UFMs[10]. Formally, assuming $d_{L-1} \geq C$, we minimize:

$$\widehat{\mathcal{R}}(\mathbf{Z}) := \frac{1}{2N} \|\mathbf{Z}\widehat{\mathbf{A}} - \mathbf{Y}\|_F^2 + \lambda_Z \|\mathbf{Z}\|_*, \tag{39}$$

where $\lambda_Z = \sqrt{\lambda_H \lambda_{W_2}}$, $\mathbf{Z} \in \mathbb{R}^{C \times N}$, and $\| \cdot \|_*$ denotes the nuclear norm. Namely, if $\mathbf{W}_2$ and $\mathbf{H}$ minimize the former, then $\mathbf{Z} = \mathbf{W}_2 \mathbf{H}$ minimizes the latter, which follows from:

$$\sqrt{\lambda_H \lambda_{W_2}} \|\mathbf{Z}\|_* = \min_{\mathbf{W}_2, \mathbf{H} \ s.t. \ \mathbf{W}_2\mathbf{H}=\mathbf{Z}} \left( \frac{\lambda_{W_2}}{2} \|\mathbf{W}_2\|_F^2 + \frac{\lambda_H}{2} \|\mathbf{H}\|_F^2 \right). \tag{40}$$

*We start with providing necessity analysis for the case $\lambda_Z = 0$. Later, we generalize this analysis to address the case $\lambda_Z > 0$.*

**Analysis for $\lambda_Z = 0$:** When $\lambda_Z = 0$, observe that $\mathbf{Z}\widehat{\mathbf{A}} = \mathbf{Y} = [\mathbf{y}_1, \ldots, \mathbf{y}_C] \otimes \mathbf{1}_n^\top = \mathbf{I}_C \otimes \mathbf{1}_n^\top$ gives us the minimum value for $\widehat{\mathcal{R}}(\mathbf{Z})$. Here $\mathbf{y}_c \in \mathbb{R}^C$ represent the one-hot label vectors corresponding to nodes of class $c \in [C]$.

Now, note that NC1 implies $\mathbf{H} = \overline{\mathbf{H}} \otimes \mathbf{1}_n^\top \iff \mathbf{Z} = \overline{\mathbf{Z}} \otimes \mathbf{1}_n^\top$ (where $\overline{\mathbf{H}} \in \mathbb{R}^{d_{L-1} \times C}$ and $\overline{\mathbf{Z}} = \mathbf{W}_2 \overline{\mathbf{H}} \in \mathbb{R}^{C \times C}$). Thus, by leveraging this Kronecker structure of $\mathbf{Z} = \overline{\mathbf{Z}} \otimes \mathbf{1}_n^\top = [\overline{\mathbf{z}}_1, \ldots, \overline{\mathbf{z}}_C] \otimes \mathbf{1}_n^\top$ and $\mathbf{Y} = [\mathbf{y}_1, \ldots, \mathbf{y}_C] \otimes \mathbf{1}_n^\top$, we formulate:

$$\begin{bmatrix} \overline{\mathbf{z}}_1 \otimes \mathbf{1}_n^\top & \cdots & \overline{\mathbf{z}}_C \otimes \mathbf{1}_n^\top \end{bmatrix} \begin{bmatrix} \mathbf{a}_{1,1}^1 & \cdots & \mathbf{a}_{1,1}^n & \cdots & \mathbf{a}_{C,1}^1 & \cdots & \mathbf{a}_{C,1}^n \\ \vdots & \vdots & \vdots & \ddots & \vdots & \vdots & \vdots \\ \mathbf{a}_{1,C}^1 & \cdots & \mathbf{a}_{1,C}^n & \cdots & \mathbf{a}_{C,C}^1 & \cdots & \mathbf{a}_{C,C}^n \end{bmatrix} \tag{41}$$
$$= \begin{bmatrix} \mathbf{y}_1 \otimes \mathbf{1}_n^\top & \cdots & \mathbf{y}_C \otimes \mathbf{1}_n^\top \end{bmatrix}$$

where $\mathbf{a}_{c,c'}^i \in \mathbb{R}^n$ represents the slice of columns in $\widehat{\mathbf{A}}$ corresponding to node $i \in [n]$ belonging to class $c \in [C]$ and forming edges with nodes from class $c' \in [C]$. Additionally, since $\widehat{\mathbf{A}}$ is the

---

[9]Note that the tuple can be different for nodes belonging to different classes, but must be the same for nodes within the same class.

[10]The $K = 1$ setting allows us to employ analysis strategies based on a tight convex formulation, which is not applicable for $K > 1$ settings. Specifically, we cannot generalize the problem stated for $\mathbf{Z} = \mathbf{W}_2 \mathbf{H}$ to multiple $\{\mathbf{H}_k\}$ as they share the same $\mathbf{W}_2$.

normalized adjacency matrix, we have $s_{cc',i} = \mathbf{1}_n^\top \mathbf{a}_{c,c'}^i \in \mathbb{R}$, which represents the sum of elements in $\mathbf{a}_{c,c'}^i$. This gives us:

$$\sum_{c'=1}^{C} s_{cc',i} = 1, \tag{42}$$

because $\widehat{\mathbf{A}} = \mathbf{A}\mathbf{D}^{-1}$ is the degree-normalized adjacency matrix. Now, multiplying the matrix $\mathbf{Z} = \overline{\mathbf{Z}} \otimes \mathbf{1}_n^\top$ with the first column of $\widehat{\mathbf{A}}$ gives us:

$$s_{11,1}\overline{\mathbf{z}}_1 + \cdots + s_{1C,1}\overline{\mathbf{z}}_C = \mathbf{y}_1 \tag{43}$$

Similarly, due to the block structure of $\mathbf{Z}, \mathbf{Y}$, we get for all $i \in [n]$ and $c = 1$:

$$s_{11,i}\overline{\mathbf{z}}_1 + \cdots + s_{1C,i}\overline{\mathbf{z}}_C = \mathbf{y}_1 \tag{44}$$

where $s_{11,i}\overline{\mathbf{z}}_1 + \cdots + s_{1C,i}\overline{\mathbf{z}}_C = \mathbf{y}_1$ itself can be written as $C$ linear equations (one for each of the vector components) as formulated below:

$$\begin{bmatrix} \overline{z}_{1,1} & \cdots & \overline{z}_{C,1} \\ \vdots & \ddots & \vdots \\ \overline{z}_{1,C} & \cdots & \overline{z}_{C,C} \end{bmatrix} \begin{bmatrix} s_{11,i} \\ \vdots \\ s_{1C,i} \end{bmatrix} = \mathbf{y}_1 \tag{45}$$

Now, by treating $\{s_{11,i}, \cdots, s_{1C,i}\}$ as the $C$ unknowns which satisfy equation 42, observe that the solution to this linear system remains the same for all nodes belonging to class $c = 1$. This implies:

$$s_{1c',1} = \cdots = s_{1c',n}, \quad \forall c' \in [C], \tag{46}$$

The generalization of this result for all $C$ classes essentially indicates that:

$$(s_{c1,1}, \cdots, s_{cC,1}) = \cdots = (s_{c1,n}, \cdots, s_{cC,n}), \quad \forall c \in [C], \tag{47}$$

which exactly represents condition **C** as stated above.

**Analysis for $\lambda_Z > 0$:** When $\mathbf{H}$ exhibits neural collapse, we have the optimality condition for the minimizer of equation 39 based on the sub-differential of the nuclear norm as follows:

$$\frac{1}{N}\left(\mathbf{Z}\widehat{\mathbf{A}} - \mathbf{Y}\right)\widehat{\mathbf{A}}^\top + \lambda_Z \mathbf{U}_Z \mathbf{V}_Z^\top = \mathbf{0}$$

$$\implies \frac{1}{N}\left((\overline{\mathbf{Z}} \otimes \mathbf{1}_n^\top)\widehat{\mathbf{A}} - \mathbf{I}_C \otimes \mathbf{1}_n^\top\right)\widehat{\mathbf{A}}^\top + \lambda_Z \mathbf{U}_Z \mathbf{V}_{\overline{Z}}^\top \otimes \frac{1}{\sqrt{n}}\mathbf{1}_n^\top = \mathbf{0}. \tag{48}$$

Where $\overline{\mathbf{Z}} \in \mathbb{R}^{C \times C}$ is the matrix of "collapsed" columns of $\mathbf{Z}$, and $\mathbf{U}_Z \in \mathbb{R}^{C \times C}, \mathbf{V}_Z \in \mathbb{R}^{N \times C}$ represent the left and right singular vectors of $\mathbf{Z}$. Additionally, since $\mathbf{Z}$ holds a "block" structure, we represent $\mathbf{V}_Z^\top = \mathbf{V}_{\overline{Z}}^\top \otimes \frac{1}{\sqrt{n}}\mathbf{1}_n^\top$, where $\mathbf{V}_{\overline{Z}} \in \mathbb{R}^{C \times C}$.

• **Matrix Quadratic Form:** By considering $-N\lambda_Z \mathbf{U}_Z \mathbf{V}_{\overline{Z}}^\top \otimes \frac{1}{\sqrt{n}}\mathbf{1}_n^\top = \mathbf{B}$, we get:

$$(\mathbf{Z}\widehat{\mathbf{A}} - \mathbf{Y})\widehat{\mathbf{A}}^\top = \mathbf{B}, \tag{49}$$

as our optimality condition. Analyzing this condition along the same lines as $\lambda_Z = 0$ is non-trivial due to the outer product of $\widehat{\mathbf{A}}\widehat{\mathbf{A}}^\top$. To address this complication, we leverage the fact that SSBM graphs tend to be regular with a high probability as $N$ increases. Thus, by assuming that $\widehat{\mathbf{A}}$ is symmetric, we obtain the following matrix quadratic form:

$$\mathbf{Z}\widehat{\mathbf{A}}^2 - \mathbf{Y}\widehat{\mathbf{A}} - \mathbf{B} = \mathbf{0}. \tag{50}$$

As $\mathbf{Z}$ is rectangular, we can take the pseudo-inverse and obtain:

$$\widehat{\mathbf{A}}^2 - \mathbf{Z}^\dagger \mathbf{Y}\widehat{\mathbf{A}} - \mathbf{Z}^\dagger \mathbf{B} = \mathbf{0}. \tag{51}$$

Since $\mathbf{Z}^\dagger \mathbf{Y}, \mathbf{Z}^\dagger \mathbf{B} \in \mathbb{R}^{N \times N}$, we can treat $\widehat{\mathbf{A}}$ as the variable matrix and leverage the results on quadratic matrix equations by Higham and Kim [27]. Especially, we leverage theorem 3 in Higham and Kim [27] and employ a generalized Schur decomposition technique to obtain a condition on $\widehat{\mathbf{A}}$ (as shown in the following lemma). This condition on $\widehat{\mathbf{A}}$ allows us to obtain a system of linear equations that are shared by all $n$ nodes within a class (similar to the $\lambda_Z = 0$ case). Thus, establishing the necessity of condition **C**.

**Lemma D.1.** *Let* $\mathbf{F} = \begin{bmatrix} \mathbf{0} & \mathbf{I} \\ \mathbf{Z}^\dagger\mathbf{B} & \mathbf{Z}^\dagger\mathbf{Y} \end{bmatrix}, \mathbf{G} = \begin{bmatrix} \mathbf{I} & \mathbf{0} \\ \mathbf{0} & \mathbf{I} \end{bmatrix} = \mathbf{I} \in \mathbb{R}^{2N \times 2N}$, *and the generalized Schur decomposition of* $\mathbf{F}, \mathbf{G}$ *be given by:*

$$\mathbf{Q}^\top\mathbf{F}\mathbf{K} = \mathbf{T}, \qquad \mathbf{Q}^\top\mathbf{G}\mathbf{K} = \mathbf{E},$$

*where* $\mathbf{Q}, \mathbf{K}$ *are unitary and* $\mathbf{T}, \mathbf{E}$ *are upper triangular.* $\mathbf{K} = \begin{bmatrix} \mathbf{K}_{11} & \mathbf{K}_{12} \\ \mathbf{K}_{21} & \mathbf{K}_{22} \end{bmatrix} \in \mathbb{R}^{2N \times 2N}$ *is a* $2 \times 2$ *block matrix with blocks of size* $N \times N$. *Then* $\widehat{\mathbf{A}}$ *satisfies:* $\mathbf{K}_{11}\widehat{\mathbf{A}} = \mathbf{K}_{21}$.

*Proof.* The proof is relatively straightforward once we observe that $\widehat{\mathbf{A}}$ satisfies:

$$\mathbf{F}\begin{bmatrix} \mathbf{I} \\ \widehat{\mathbf{A}} \end{bmatrix} = \mathbf{G}\begin{bmatrix} \mathbf{I} \\ \widehat{\mathbf{A}} \end{bmatrix}\widehat{\mathbf{A}}. \tag{52}$$

Now, the condition $\mathbf{K}_{11}\widehat{\mathbf{A}} = \mathbf{K}_{21}$ is a direct consequence of theorem 3 in [27], which leverages the QR decomposition of $\begin{bmatrix} \mathbf{I} \\ \widehat{\mathbf{A}} \end{bmatrix}$ using $\mathbf{K}$ as the orthogonal matrix. $\square$

• **Kronecker structure of F:** To leverage the relationship between $\widehat{\mathbf{A}}$ and $\mathbf{K}$ as per Lemma D.1, a closer look at $\mathbf{F}$ is required. Observe that:

$$\mathbf{F} = \begin{bmatrix} \mathbf{0} & \mathbf{I} \\ \mathbf{Z}^\dagger\mathbf{B} & \mathbf{Z}^\dagger\mathbf{Y} \end{bmatrix} = \begin{bmatrix} \mathbf{Z}^\dagger\mathbf{0} & \mathbf{Z}^\dagger\mathbf{Z} \\ \mathbf{Z}^\dagger\mathbf{B} & \mathbf{Z}^\dagger\mathbf{Y} \end{bmatrix} = \begin{bmatrix} \mathbf{Z}^\dagger & \mathbf{0} \\ \mathbf{0} & \mathbf{Z}^\dagger \end{bmatrix}\begin{bmatrix} \mathbf{0} & \mathbf{Z} \\ \mathbf{B} & \mathbf{Y} \end{bmatrix}. \tag{53}$$

By expanding the Kronecker structures of $\mathbf{Z}, \mathbf{B}, \mathbf{Y}$, we get:

$$\begin{aligned}
\mathbf{F} &= \begin{bmatrix} \mathbf{Z}^\dagger & \mathbf{0} \\ \mathbf{0} & \mathbf{Z}^\dagger \end{bmatrix}\begin{bmatrix} \mathbf{0} \otimes \mathbf{1}_n^\top & \overline{\mathbf{Z}} \otimes \mathbf{1}_n^\top \\ \overline{\mathbf{B}} \otimes \mathbf{1}_n^\top & \mathbf{I}_C \otimes \mathbf{1}_n^\top \end{bmatrix} \\
&= \begin{bmatrix} \mathbf{0} & \cdots & \mathbf{0} & \mathbf{Z}^\dagger\overline{\mathbf{z}}_1 & \cdots & \mathbf{Z}^\dagger\overline{\mathbf{z}}_C \\ \mathbf{Z}^\dagger\overline{\mathbf{b}}_1 & \cdots & \mathbf{Z}^\dagger\overline{\mathbf{b}}_C & \mathbf{Z}^\dagger\mathbf{e}_1 & \cdots & \mathbf{Z}^\dagger\mathbf{e}_C \end{bmatrix} \otimes \mathbf{1}_n^\top.
\end{aligned} \tag{54}$$

Observe that the pseudo-inverse $\mathbf{Z}^\dagger$ can be represented by:

$$\mathbf{Z}^\dagger = (\mathbf{V}_{\overline{Z}} \otimes \frac{1}{\sqrt{n}}\mathbf{1}_n)\mathbf{S}_Z^\dagger\mathbf{U}_Z^\top \tag{55}$$

which also holds a "block" structure (but with respect to rows, instead of columns):

$$\mathbf{Z}^\dagger = \frac{1}{\sqrt{n}}\begin{bmatrix} \mathbf{v}_1^\top \\ \vdots \\ \mathbf{v}_C^\top \end{bmatrix}\mathbf{S}_Z^\dagger\mathbf{U}_Z^\top = \frac{1}{\sqrt{n}}\begin{bmatrix} \mathbf{v}_1^\top\mathbf{S}_Z^\dagger\mathbf{U}_Z^\top \\ \vdots \\ \mathbf{v}_C^\top\mathbf{S}_Z^\dagger\mathbf{U}_Z^\top \end{bmatrix}_{N \times C}. \tag{56}$$

Where $\mathbf{v}_j \in \mathbb{R}^C, \forall j \in [C]$ and $\mathbf{v}_j^\top$ is the $j^{th}$ row of $\mathbf{V}_{\overline{Z}} \in \mathbb{R}^{C \times C}$. With this formulation, a matrix-vector product term in $\mathbf{F}$, for instance $\mathbf{Z}^\dagger\overline{\mathbf{b}}_i, i \in [C]$, can be given as:

$$\mathbf{Z}^\dagger\overline{\mathbf{b}}_i = \frac{1}{\sqrt{n}}\begin{bmatrix} \mathbf{v}_1^\top\mathbf{S}_Z^\dagger\mathbf{U}_Z^\top\overline{\mathbf{b}}_i \\ \vdots \\ \mathbf{v}_C^\top\mathbf{S}_Z^\dagger\mathbf{U}_Z^\top\overline{\mathbf{b}}_i \end{bmatrix}_{N \times 1} = (\mathbf{V}_{\overline{Z}}\mathbf{S}_Z^\dagger\mathbf{U}_Z^\top\overline{\mathbf{b}}_i) \otimes \frac{1}{\sqrt{n}}\mathbf{1}_n. \tag{57}$$

By considering the following notational simplifications:

$$\begin{aligned}
\widetilde{\mathbf{b}}_i &= \mathbf{V}_{\overline{Z}}\mathbf{S}_Z^\dagger\mathbf{U}_Z^\top\overline{\mathbf{b}}_i \\
\widetilde{\mathbf{z}}_i &= \mathbf{V}_{\overline{Z}}\mathbf{S}_Z^\dagger\mathbf{U}_Z^\top\overline{\mathbf{z}}_i \\
\widetilde{\mathbf{e}}_i &= \mathbf{V}_{\overline{Z}}\mathbf{S}_Z^\dagger\mathbf{U}_Z^\top\mathbf{e}_i,
\end{aligned} \tag{58}$$

we can represent $\mathbf{F}$ as:

$$\mathbf{F} = \begin{bmatrix} \mathbf{0} & \cdots & \mathbf{0} & \mathbf{Z}^\dagger \bar{\mathbf{z}}_1 & \cdots & \mathbf{Z}^\dagger \bar{\mathbf{z}}_C \\ \mathbf{Z}^\dagger \bar{\mathbf{b}}_1 & \cdots & \mathbf{Z}^\dagger \bar{\mathbf{b}}_C & \mathbf{Z}^\dagger \mathbf{e}_1 & \cdots & \mathbf{Z}^\dagger \mathbf{e}_C \end{bmatrix} \otimes \mathbf{1}_n^\top$$

$$= \begin{bmatrix} \mathbf{0} & \cdots & \mathbf{0} & \widetilde{\mathbf{z}}_1 \otimes \frac{1}{\sqrt{n}}\mathbf{1}_n & \cdots & \widetilde{\mathbf{z}}_C \otimes \frac{1}{\sqrt{n}}\mathbf{1}_n \\ \widetilde{\mathbf{b}}_1 \otimes \frac{1}{\sqrt{n}}\mathbf{1}_n & \cdots & \widetilde{\mathbf{b}}_C \otimes \frac{1}{\sqrt{n}}\mathbf{1}_n & \widetilde{\mathbf{e}}_1 \otimes \frac{1}{\sqrt{n}}\mathbf{1}_n & \cdots & \widetilde{\mathbf{e}}_1 \otimes \frac{1}{\sqrt{n}}\mathbf{1}_n \end{bmatrix} \otimes \mathbf{1}_n^\top \qquad (59)$$

$$= \begin{bmatrix} \mathbf{0} & \cdots & \mathbf{0} & \widetilde{\mathbf{z}}_1 & \cdots & \widetilde{\mathbf{z}}_C \\ \widetilde{\mathbf{b}}_1 & \cdots & \widetilde{\mathbf{b}}_C & \widetilde{\mathbf{e}}_1 & \cdots & \widetilde{\mathbf{e}}_1 \end{bmatrix}_{2C \times 2C} \otimes \frac{1}{\sqrt{n}}\mathbf{1}_n \otimes \mathbf{1}_n^\top.$$

• **Schur decomposition of F:** For notational simplicity, let :

$$\mathbf{F} = \widetilde{\mathbf{F}} \otimes \frac{1}{\sqrt{n}}\mathbf{1}_n \otimes \mathbf{1}_n^\top$$

$$\text{Where:} \quad \widetilde{\mathbf{F}} = \begin{bmatrix} \mathbf{0} & \cdots & \mathbf{0} & \widetilde{\mathbf{z}}_1 & \cdots & \widetilde{\mathbf{z}}_C \\ \widetilde{\mathbf{b}}_1 & \cdots & \widetilde{\mathbf{b}}_C & \widetilde{\mathbf{e}}_1 & \cdots & \widetilde{\mathbf{e}}_1 \end{bmatrix}_{2C \times 2C}. \qquad (60)$$

Since $\mathbf{G} = \mathbf{I}$, the diagonal entries of $\mathbf{S}$ must equal the eigenvalues of $\mathbf{I}$. This condition is satisfied when $\mathbf{Q} = \mathbf{K}$. This also simplifies the generalized Schur decomposition for square matrices $\mathbf{F}, \mathbf{G}$ to the standard Schur decomposition of $\mathbf{F}$. Now, let the schur decomposition of $\mathbf{F} \in \mathbb{R}^{2N \times 2N}, \widetilde{\mathbf{F}} \in \mathbb{R}^{2C \times 2C}$ be given as:

$$\mathbf{F} = \mathbf{K}\mathbf{T}\mathbf{K}^\top, \widetilde{\mathbf{F}} = \widetilde{\mathbf{K}}\widetilde{\mathbf{T}}\widetilde{\mathbf{K}}^\top. \qquad (61)$$

Here $\mathbf{K}, \mathbf{T} \in \mathbb{R}^{2N \times 2N}$ and $\widetilde{\mathbf{K}}, \widetilde{\mathbf{T}} \in \mathbb{R}^{2C \times 2C}$. To find a relation between $\mathbf{K}, \mathbf{T}, \widetilde{\mathbf{K}}, \widetilde{\mathbf{T}}$, we can leverage the Schur decomposition properties of Kronecker products [53, 59] and obtain:

$$\mathbf{F} = \widetilde{\mathbf{F}} \otimes \frac{1}{\sqrt{n}}\mathbf{1}_n \otimes \mathbf{1}_n^\top \qquad (62)$$

$$= \widetilde{\mathbf{F}} \otimes \frac{1}{\sqrt{n}}\mathbf{1}_n\mathbf{1}_n^\top \qquad (63)$$

$$\mathbf{K}\mathbf{T}\mathbf{K}^\top = \left(\widetilde{\mathbf{K}}\widetilde{\mathbf{T}}\widetilde{\mathbf{K}}^\top\right) \otimes \left(\mathbf{J}\mathbf{O}\mathbf{J}^\top\right) \qquad (64)$$

$$= \left(\widetilde{\mathbf{K}} \otimes \mathbf{J}\right)\left(\widetilde{\mathbf{T}} \otimes \mathbf{O}\right)\left(\widetilde{\mathbf{K}}^\top \otimes \mathbf{J}^\top\right). \qquad (65)$$

Where $\mathbf{J}, \mathbf{O} \in \mathbb{R}^{n \times n}$ are unitary and upper triangular respectively, and are the Schur decomposition factors of $\frac{1}{\sqrt{n}}\mathbf{1}_n\mathbf{1}_n^\top \in \mathbb{R}^{n \times n}$.

• **Linear systems:** In matrix form, $\mathbf{K} = \widetilde{\mathbf{K}} \otimes \mathbf{J}$ can be represented as:

$$\mathbf{K} = \left[ \begin{array}{c|c} \widetilde{\mathbf{K}}_{11} \otimes \mathbf{J} & \widetilde{\mathbf{K}}_{12} \otimes \mathbf{J} \\ \widetilde{\mathbf{K}}_{21} \otimes \mathbf{J} & \widetilde{\mathbf{K}}_{22} \otimes \mathbf{J} \end{array} \right]_{2N \times 2N}. \qquad (66)$$

Where $\widetilde{\mathbf{K}}_{11}, \widetilde{\mathbf{K}}_{12}, \widetilde{\mathbf{K}}_{21}, \widetilde{\mathbf{K}}_{22} \in \mathbb{R}^{C \times C}$. Now, observe that $\mathbf{K}_{11}\widehat{\mathbf{A}} = \mathbf{K}_{21}$ (based on Lemma D.1) can be reformulated as:

$$\left(\widetilde{\mathbf{K}}_{11} \otimes \mathbf{J}\right)\widehat{\mathbf{A}} = \left(\widetilde{\mathbf{K}}_{21} \otimes \mathbf{J}\right). \qquad (67)$$

Now, as per equation 41, we leverage the same line of analysis that we followed for the $\lambda_Z = 0$ case. For notational simplicity, we represent the unitary matrix $\mathbf{J} \in \mathbb{R}^{n \times n}$ in column format as follows:

$$\mathbf{J} = [\mathbf{j}_1 \quad \cdots \quad \mathbf{j}_n], \qquad (68)$$

where $\mathbf{j}_i \in \mathbb{R}^n, i \in [n]$ are linearly independent vectors. Now, by multiplying the first row of $\widetilde{\mathbf{K}}_{11} \otimes \mathbf{J}$ and the first column of $\widehat{\mathbf{A}}$, we get:

$$\begin{bmatrix} (\widetilde{\mathbf{K}}_{11})_{1,1} \begin{bmatrix} \mathbf{j}_1 & \cdots & \mathbf{j}_n \end{bmatrix} & \cdots & (\widetilde{\mathbf{K}}_{11})_{1,C} \begin{bmatrix} \mathbf{j}_1 & \cdots & \mathbf{j}_n \end{bmatrix} \end{bmatrix} \begin{bmatrix} a_{1,1} \\ \vdots \\ a_{n,1} \\ \hline \vdots \\ \hline a_{(C-1)n+1,1} \\ \vdots \\ a_{Cn,1} \end{bmatrix} = (\widetilde{\mathbf{K}}_{21})_{1,1} \mathbf{j}_1.$$

This translates to the following linear equation:

$$a_{1,1}(\widetilde{\mathbf{K}}_{11})_{1,1}\mathbf{j}_1 + \cdots + a_{n,1}(\widetilde{\mathbf{K}}_{11})_{1,1}\mathbf{j}_n + \cdots \\ + a_{(C-1)n+1,1}(\widetilde{\mathbf{K}}_{11})_{1,C}\mathbf{j}_1 + \cdots + a_{Cn,1}(\widetilde{\mathbf{K}}_{11})_{1,C}\mathbf{j}_n = (\widetilde{\mathbf{K}}_{21})_{1,1}\mathbf{j}_1. \tag{69}$$

Due to linear independence of vectors $\mathbf{j}_i, i \in [n]$, we obtain the following $n$ equations pertaining to the coefficients of $\mathbf{j}_i$:

$$a_{1,1}(\widetilde{\mathbf{K}}_{11})_{1,1} + \cdots + a_{(C-1)n+1,1}(\widetilde{\mathbf{K}}_{11})_{1,C} = (\widetilde{\mathbf{K}}_{21})_{1,1}$$
$$a_{2,1}(\widetilde{\mathbf{K}}_{11})_{1,1} + \cdots + a_{(C-1)n+2,1}(\widetilde{\mathbf{K}}_{11})_{1,C} = 0$$
$$\vdots$$
$$a_{n,1}(\widetilde{\mathbf{K}}_{11})_{1,1} + \cdots + a_{Cn,1}(\widetilde{\mathbf{K}}_{11})_{1,C} = 0.$$

By adding all these equations, we get:

$$s_{11,1}(\widetilde{\mathbf{K}}_{11})_{1,1} + \cdots + s_{1C,1}(\widetilde{\mathbf{K}}_{11})_{1,C} = (\widetilde{\mathbf{K}}_{21})_{1,1}. \tag{70}$$

By following the same approach for the other rows of $\widetilde{\mathbf{K}}_{11} \otimes \mathbf{K}_{1_n}$ and the first column of $\widehat{\mathbf{A}}$, we get the following system of equations for node $v_{1,1}$:

$$\begin{bmatrix} (\widetilde{\mathbf{K}}_{11})_{1,1} & \cdots & (\widetilde{\mathbf{K}}_{11})_{1,C} \\ \vdots & \ddots & \vdots \\ (\widetilde{\mathbf{K}}_{11})_{C,1} & \cdots & (\widetilde{\mathbf{K}}_{11})_{C,C} \end{bmatrix} \begin{bmatrix} s_{11,1} \\ \vdots \\ s_{1C,1} \end{bmatrix} = \begin{bmatrix} (\widetilde{\mathbf{K}}_{21})_{1,1} \\ \vdots \\ (\widetilde{\mathbf{K}}_{21})_{C,1} \end{bmatrix}. \tag{71}$$

The same line of analysis can be applied for all the rows of $\widetilde{\mathbf{K}}_{11} \otimes \mathbf{K}_{1_n}$ and the second column of $\widehat{\mathbf{A}}$, to get the following system of equations for node $v_{1,2}$:

$$\begin{bmatrix} (\widetilde{\mathbf{K}}_{11})_{1,1} & \cdots & (\widetilde{\mathbf{K}}_{11})_{1,C} \\ \vdots & \ddots & \vdots \\ (\widetilde{\mathbf{K}}_{11})_{C,1} & \cdots & (\widetilde{\mathbf{K}}_{11})_{C,C} \end{bmatrix} \begin{bmatrix} s_{11,2} \\ \vdots \\ s_{1C,2} \end{bmatrix} = \begin{bmatrix} (\widetilde{\mathbf{K}}_{21})_{1,1} \\ \vdots \\ (\widetilde{\mathbf{K}}_{21})_{C,1} \end{bmatrix} \tag{72}$$

Thus, it is straightforward that the systems of equations are the same for all $n$ nodes belonging to class $c = 1$, as the procedure of row and column multiplication remains the same. Thus, the $C$ unknowns in these linear systems have the same solution for all $n$ nodes belonging to class $c = 1$. It is straightforward to extend this to any class $c \in [C]$ and obtain:

$$(s_{c1,1}, \cdots, s_{cC,1}) = \cdots = (s_{c1,n}, \cdots, s_{cC,n}), \quad \forall c \in [C], \tag{73}$$

which exactly represents the condition $\mathbf{C}$ as per the theorem.

# E Proof of Theorem 3.2

In this appendix, we derive the upper bound on the probability of sampling the desired neighborhood for condition **C**.

Recall that to satisfy condition **C**, the requirement w.r.t $\widehat{\mathbf{A}}$ is for $\left( \frac{|\mathcal{N}_1(v_{c,i})|}{|\mathcal{N}(v_{c,i})|}, \cdots, \frac{|\mathcal{N}_C(v_{c,i})|}{|\mathcal{N}(v_{c,i})|} \right), \forall i \in [n]$ to be the same for a given $c \in [C]$. To this end, we are primarily concerned with the probabilities of edges between nodes $v_i, v_j, 1 \leq i \leq j \leq N$. Thus, as per preliminaries, the probability matrix $\mathbf{P}$ can be given in block form as:

$$
\mathbf{P} = \begin{bmatrix} p\mathbf{1}_n\mathbf{1}_n^\top & q\mathbf{1}_n\mathbf{1}_n^\top & \cdots & q\mathbf{1}_n\mathbf{1}_n^\top \\ \vdots & p\mathbf{1}_n\mathbf{1}_n^\top & \cdots & q\mathbf{1}_n\mathbf{1}_n^\top \\ \vdots & & \ddots & \ddots & \vdots \\ \vdots & & \cdots & \cdots & p\mathbf{1}_n\mathbf{1}_n^\top \end{bmatrix}_{N \times N}
$$

Where we are only concerned with the diagonal and upper triangular values[11]. Now, for a pair of classes $c, c' \in [C]$, we are concerned with the block probability matrix $p\mathbf{1}_n\mathbf{1}_n^\top$ when $c = c'$ and $q\mathbf{1}_n\mathbf{1}_n^\top$ when $c \neq c'$ for sampling edges between nodes. Observe that sampling edges within a community based on diagonal block matrix $p\mathbf{1}_n\mathbf{1}_n^\top$ is the same as sampling an Erdos-Renyi graph with edge probability $p$. Similarly, sampling edges between communities based on off-diagonal block matrix $q\mathbf{1}_n\mathbf{1}_n^\top$ is the same as sampling a bipartite graph with edge probability $q$.

**Concentration of pairwise neighbor ratios:** To begin with, consider the set of nodes belonging to class $c \in [C]$ as $\Omega_c = \{v_{c,1}, \cdots, v_{c,n}\}$. Now, to satisfy condition **C**, the fraction of neighbors of nodes $v_{c,i}, v_{c,j}, i \neq j \in [n]$ that belong to class $c' \in [C]$ should be equal, i.e $s_{cc',i} = s_{cc',j}$. Formally, this leads to:

$$
\frac{|\mathcal{N}_{c'}(v_{c,i})|}{|\mathcal{N}(v_{c,i})|} = \frac{|\mathcal{N}_{c'}(v_{c,j})|}{|\mathcal{N}(v_{c,j})|} \implies \frac{|\mathcal{N}_{c'}(v_{c,i})|}{|\mathcal{N}_{c'}(v_{c,j})|} = \frac{|\mathcal{N}(v_{c,i})|}{|\mathcal{N}(v_{c,j})|}
$$

Without loss of generality, observe that $|\mathcal{N}_{c'}(v_{c,i})|$ is the sum of $n$ independent Bernoulli random variables $\gamma_{c,i}^{c',l}, \forall l \in [n]$ with $\mathbb{P}(\gamma_{c,i}^{c',l} = 1) = q$. This implies that $\mathbb{E}|\mathcal{N}_{c'}(v_{c,i})| = nq$. Now, we apply the Chernoff bound to obtain:

$$
\mathbb{P}\left( ||\mathcal{N}_{c'}(v_{c,i})| - nq| \geq \delta nq \right) \leq 2e^{-tnq\delta^2}
$$

Where $\delta \in [0, 1]$ and $t > 0$ is a constant. By choosing $\delta = \sqrt{\frac{(r+1)\ln n}{tnq}}$ for sufficiently large $r > 0, N >> C$, we get $\delta = O(1)$ as $q = \frac{b \ln N}{N}$. Now, by taking a union bound over all the nodes in the class, we get:

$$
\mathbb{P}\left( \forall i \in [n], ||\mathcal{N}_{c'}(v_{c,i})| - nq| \geq \delta nq \right) \leq 2ne^{-(r+1)\ln n}
$$

Thus, with a probability at-least $1 - 2n^{-r}$, we get:

$$
|\mathcal{N}_{c'}(v_{c,i})| = nq\left(1 \pm O(1)\right)
$$

By applying the same line of argument to $|\mathcal{N}_{c'}(v_{c,j})|$ and assuming a sufficiently large value of $n$, we get with a probability at-least $1 - 4n^{-r}$ that:

$$
\frac{|\mathcal{N}_{c'}(v_{c,i})|}{|\mathcal{N}_{c'}(v_{c,j})|} \to 1
$$

To this end, we assume that $|\mathcal{N}_{c'}(v_{c,i})| = |\mathcal{N}_{c'}(v_{c,j})|, \forall c \in [C], i \neq j \in [n]$ with a high probability for the rest of the analysis. *The consequence of this assumption is that all the nodes belonging to the same class have the same degree. However, it is not necessary that the graph itself is regular.*

**Off-diagonal blocks:** Without loss of generality, consider the set of nodes belonging to a pair of classes $c \neq c' \in [C]$ as $\Omega_c = \{v_{c,1}, \cdots, v_{c,n}\}, \Omega_{c'} = \{v_{c',1}, \cdots, v_{c',n}\}$ respectively. Now, we need

---

[11]Due to symmetry, one can equivalently consider the lower triangular values and proceed with sums of columns instead of sums of rows.

to ensure that every node $v_{c,i} \in \Omega_c$ is connected to (say) exactly $t_{cc'} \in \mathbb{R}, 0 \leq t_{cc'} \leq n$, nodes in $\Omega_{c'}$. If $E_{c,i}^{c'}(t_{cc'})$ indicates that such an event occurs for node $v_{c,i} \in \Omega_c$ with respect to $\Omega_{c'}$, we can formally represent it as the sum of $n$ independent Bernoulli random variables $\gamma_{c,i}^{c';j}, i \in [n], \forall j \in \{1, \ldots, n\}$ with $\mathbb{P}(\gamma_{c,i}^{c';j} = 1) = q$ sum to $t_{cc'}$ as follows:

$$\mathbb{P}(E_{c,i}^{c'}(t_{cc'})) = \binom{n}{t_{cc'}} q^{t_{cc'}} (1-q)^{n-t_{cc'}}.$$

Now, we are concerned with an intersection of events pertaining to all nodes in $\Omega_c$, which ensures that each node has exactly $t_{cc'}$ neighbors in $\Omega_{c'}$. By considering the event $E_c^{c'}(t_{cc'}) = \bigcap_{i=1}^n E_{c,i}^{c'}(t_{cc'})$ and leveraging the fact that edges are sampled independently, we obtain:

$$\mathbb{P}\left(E_c^{c'}(t_{cc'})\right) = \mathbb{P}\left(\bigcap_{i=1}^n E_{c,i}^{c'}(t_{cc'})\right) = \left[\binom{n}{t_{cc'}} q^{t_{cc'}} (1-q)^{n-t_{cc'}}\right]^n.$$

Now, to account for all possible values of $0 \leq t_{cc'} \leq n$, we compute the probability of the union of events $\widetilde{E}_c^{c'} = \bigcup_{t_{cc'}=0}^n E_c^{c'}(t_{cc'})$ as:

$$\mathbb{P}\left(\widetilde{E}_c^{c'}\right) \leq \sum_{t_{cc'}=0}^n \left[\binom{n}{t_{cc'}} q^{t_{cc'}} (1-q)^{n-t_{cc'}}\right]^n.$$

Note that this result can be applied to any distinct pair of classes $c \neq c' \in [C]$ based on the characteristics of the SSBM. Since we have $\binom{C}{2} = \frac{C(C-1)}{2}$ combinations of distinct communities, the probability of occurrence of all the corresponding events is given by:

$$\mathbb{P}\left(\bigcap_{c=1}^{C-1} \bigcap_{c'=c+1}^{C} \widetilde{E}_c^{c'}\right) \leq \prod_{c=1}^{C-1} \prod_{c'=c+1}^{C} \sum_{t_{cc'}=0}^n \left[\binom{n}{t_{cc'}} q^{t_{cc'}} (1-q)^{n-t_{cc'}}\right]^n. \tag{74}$$

Observe that the binomial expansion of $(q + 1 - q)^n$ is given by:

$$1 = (q + 1 - q)^n = \sum_{t_{cc'}=0}^n \binom{n}{t_{cc'}} q^{t_{cc'}} (1-q)^{n-t_{cc'}}, \tag{75}$$

where each term, say $f(t_{cc'}) = \binom{n}{t_{cc'}} q^{t_{cc'}} (1-q)^{n-t_{cc'}}$ in the sum is strictly less than 1. Additionally,

$$1^n = \left(\sum_{t_{cc'}=0}^n f(t_{cc'})\right)^n = \sum_{k_0+\cdots+k_n=n, k_0,\cdots,k_n \geq 0} \binom{n}{k_0, \cdots, k_n} \prod_{t_{cc'}=0}^n f(t_{cc'})^{k_{t_{cc'}}} \tag{76}$$

$$= \sum_{t_{cc'}=0}^n f(t_{cc'})^n + \sum_{k_0+\cdots+k_n=n, 0 \leq k_0,\cdots,k_n < n} \binom{n}{k_0, \cdots, k_n} \prod_{t_{cc'}=0}^n f(t_{cc'})^{k_{t_{cc'}}}. \tag{77}$$

This gives us:

$$\sum_{t_{cc'}=0}^n f(t_{cc'})^n = 1 - \sum_{k_0+\cdots+k_n=n, 0 \leq k_0,\cdots,k_n < n} \binom{n}{k_0, \cdots, k_n} \prod_{t_{cc'}=0}^n f(t_{cc'})^{k_{t_{cc'}}}. \tag{78}$$

Observe that there are $n + 1$ terms in $\sum_{t_{cc'}=0}^n f(t_{cc'}) = 1$ and the maximum value that $f(t_{cc'})$ can take is $< 1$. Thus, each of the $f(t_{cc'})$ terms tend to zero after taking the $n^{th}$ power, as $n \to \infty$, leading to $\sum_{t_{cc'}=0}^n f(t_{cc'})^n \to 0$.

**Diagonal blocks:** Handling the exact event probabilities when $c = c' \in [C]$ is not so straightforward due to symmetry constraints. To begin with, observe that the sum of $n$ independent Bernoulli random variables $\gamma_{c,i}^{c,j}, i \in [n], \forall j \in \{1, \ldots, n\}$ with $\mathbb{P}(\gamma_{c,i}^{c,j} = 1) = p$ sum to $t_{cc}$ is given by:

$$\mathbb{P}\left(E_{c,i}^c(t_{cc})\right) = \mathbb{P}\left(\sum_{j=1}^n \gamma_{c,i}^{c,j} = t_{cc}\right) = \binom{n}{t_{cc}} p^{t_{cc}} (1-p)^{n-t_{cc}}$$

Since $\mathbb{P}(E^c_{c,j}(t_{cc}))$ for $j > i \in [n]$ is conditional on $\mathbb{P}(E^c_{c,i}(t_{cc}))$, the desired probability for the event $E^c_c(t_{cc}) = \bigcap_{i=1}^n E^c_{c,i}(t_{cc})$ can be formulated as:

$$\mathbb{P}\left(E^c_c(t_{cc})\right) = \mathbb{P}\left(\bigcap_{i=1}^n E^c_{c,i}(t_{cc})\right)$$

$$= \mathbb{P}\left(E^c_{c,1}(t_{cc})\right) \cdot \mathbb{P}\left(E^c_{c,2}(t_{cc})\big|E^c_{c,1}(t_{cc})\right) \cdots \mathbb{P}\left(E^c_{c,n}(t_{cc})\bigg|\bigcap_{i=1}^{n-1} E^c_{c,i}(t_{cc})\right)$$

Obtaining a clean expression for the diagonal case is complicated by the fact that the events $E^c_{c,1}(t_{cc}), E^c_{c,2}(t_{cc}), \cdots, E^c_{c,n}(t_{cc})$ are not independent. Since we have already shown in the off-diagonal case that the $\mathbb{P}\left(\bigcap_{c=1}^{C-1}\bigcap_{c'=c+1}^C \widetilde{E}^{c'}_c\right) \to 0$ as $n \to \infty$, we proceed with the simpler inequality $\mathbb{P}\left(E^c_c(t_{cc})\right) < 1$ to convey the message of this theorem. Thus, we estimate the probability of the intersection of all events pertaining to $c, c' \in [C]$ as:

$$\mathbb{P}\left(\bigcap_{c=1}^C\bigcap_{c'=c}^C \widetilde{E}^{c'}_c\right) < \left(\prod_{c=1}^{C-1}\prod_{c'=c+1}^C \sum_{t_{cc'}=0}^n \left[\binom{n}{t_{cc'}}q^{t_{cc'}}(1-q)^{n-t_{cc'}}\right]^n\right)$$

$$\implies \mathbb{P}\left(\bigcap_{c=1}^C\bigcap_{c'=c}^C \widetilde{E}^{c'}_c\right) < \left(\sum_{t=0}^n \left[\binom{n}{t}q^t(1-q)^{n-t}\right]^n\right)^{\frac{C(C-1)}{2}} \tag{79}$$

### E.1 Illustration with exhaustive combinations for a small graph

A key assumption in our theoretical analysis has been $N >> C$, which allowed us to assume that nodes belonging to the same class have the same degree. However, for an intuitive understanding of condition **C**, let us consider an extremely simple graph with $N = 4, C = 2$. This leads to the following adjacency matrix formulation:

$$\mathbf{A} = \begin{bmatrix} \gamma_{11} & \gamma_{12} & \gamma_{13} & \gamma_{14} \\ \gamma_{12} & \gamma_{22} & \gamma_{23} & \gamma_{24} \\ \gamma_{13} & \gamma_{23} & \gamma_{33} & \gamma_{34} \\ \gamma_{14} & \gamma_{24} & \gamma_{34} & \gamma_{44} \end{bmatrix} \tag{80}$$

Where $\gamma_{ij}$ represents a Bernoulli random variable depending on $p, q$. We defer assigning values to $p, q$ until the end as we are interested in the 'realizations' of $\mathbf{A}$ that satisfy condition **C**. Observe that there are only 10 random variables in $\mathbf{A}$ due to symmetry (4 on the diagonal and 6 on the upper-triangular part). Since each can either take a value of $0, 1$, there are 1024 unique realizations of $\mathbf{A}$. To this end, condition **C** can be represented as:

$$\frac{\gamma_{11} + \gamma_{12}}{\gamma_{11} + \gamma_{12} + \gamma_{13} + \gamma_{14}} = \frac{\gamma_{12} + \gamma_{22}}{\gamma_{12} + \gamma_{22} + \gamma_{23} + \gamma_{24}} \qquad and$$

$$\frac{\gamma_{13} + \gamma_{14}}{\gamma_{11} + \gamma_{12} + \gamma_{13} + \gamma_{14}} = \frac{\gamma_{23} + \gamma_{24}}{\gamma_{12} + \gamma_{22} + \gamma_{23} + \gamma_{24}} \qquad and$$

$$\frac{\gamma_{13} + \gamma_{23}}{\gamma_{13} + \gamma_{23} + \gamma_{33} + \gamma_{34}} = \frac{\gamma_{14} + \gamma_{24}}{\gamma_{14} + \gamma_{24} + \gamma_{34} + \gamma_{44}} \qquad and \tag{81}$$

$$\frac{\gamma_{33} + \gamma_{34}}{\gamma_{13} + \gamma_{23} + \gamma_{33} + \gamma_{34}} = \frac{\gamma_{34} + \gamma_{44}}{\gamma_{14} + \gamma_{24} + \gamma_{34} + \gamma_{44}} \qquad .$$

Based on our simulations for graphs with self-edges, less than $1/10$ of the 1024 realizations satisfy this property. A few are illustrated below:

$$\mathbf{A} = \begin{bmatrix} 1 & 1 & 0 & 1 \\ 1 & 1 & 1 & 0 \\ 0 & 1 & 1 & 0 \\ 1 & 0 & 0 & 1 \end{bmatrix}, \qquad \mathbf{A} = \begin{bmatrix} 1 & 1 & 0 & 1 \\ 1 & 1 & 1 & 0 \\ 0 & 1 & 0 & 1 \\ 1 & 0 & 1 & 0 \end{bmatrix}$$

$$\mathbf{A} = \begin{bmatrix} 1 & 0 & 1 & 0 \\ 0 & 1 & 0 & 1 \\ 1 & 0 & 1 & 0 \\ 0 & 1 & 0 & 1 \end{bmatrix}, \qquad \mathbf{A} = \begin{bmatrix} 1 & 0 & 0 & 1 \\ 0 & 1 & 1 & 0 \\ 0 & 1 & 0 & 1 \\ 1 & 0 & 1 & 0 \end{bmatrix}$$

By considering $p = 0.2, q = 0.05$, the probability of sampling such $\mathbf{A}$ is $\approx 0.046$. Interestingly, when $p = 0.05, q = 0.2$, the probability increases to $\approx 0.06$. Now, by increasing the density of edges in the graph, i.e., when $p = 0.4, q = 0.1$ and $p = 0.1, q = 0.4$, we get probability values $\approx 0.166, 0.178$ respectively. Now, let us consider the case where $N = 8, C = 2$. In this case, we have 36 Bernoulli random variables $\gamma_{ij}, i \leq j \in \{1, \cdots, 8\}$. The number of possible values for $\mathbf{A}$ turns out to be $2^{36} = 68,719,476,736$, for which the brute force approach to validate condition $\mathbf{C}$ is not efficient. To this end, we follow a simple Monte-Carlo approach and draw $1,000,000$ random graphs from SSBM($N = 8, C = 2, p = 0.5, q = 0.2$). We observed that only $\approx 800$ graphs out of $1,000,000$ satisfied condition $\mathbf{C}$, i.e., a probability of $\approx 0.0008$. A few are illustrated below:

$$\mathbf{A} = \begin{bmatrix} 0 & 1 & 1 & 1 & 0 & 0 & 0 & 0 \\ 1 & 1 & 0 & 1 & 0 & 0 & 0 & 0 \\ 1 & 0 & 1 & 1 & 0 & 0 & 0 & 0 \\ 1 & 1 & 1 & 0 & 0 & 0 & 0 & 0 \\ 0 & 0 & 0 & 0 & 0 & 1 & 1 & 0 \\ 0 & 0 & 0 & 0 & 1 & 1 & 1 & 1 \\ 0 & 0 & 0 & 0 & 1 & 1 & 1 & 1 \\ 0 & 0 & 0 & 0 & 0 & 1 & 1 & 0 \end{bmatrix}, \quad \mathbf{A} = \begin{bmatrix} 1 & 1 & 1 & 1 & 0 & 1 & 0 & 0 \\ 1 & 1 & 1 & 1 & 0 & 0 & 1 & 0 \\ 1 & 1 & 1 & 1 & 1 & 0 & 0 & 0 \\ 1 & 1 & 1 & 1 & 0 & 0 & 0 & 1 \\ 0 & 0 & 1 & 0 & 0 & 1 & 1 & 1 \\ 1 & 0 & 0 & 0 & 1 & 1 & 0 & 1 \\ 0 & 1 & 0 & 0 & 1 & 0 & 1 & 1 \\ 0 & 0 & 0 & 1 & 1 & 1 & 1 & 0 \end{bmatrix}$$

$$\mathbf{A} = \begin{bmatrix} 0 & 1 & 1 & 0 & 0 & 1 & 0 & 1 \\ 1 & 1 & 0 & 1 & 0 & 1 & 1 & 1 \\ 1 & 0 & 1 & 1 & 1 & 1 & 1 & 0 \\ 0 & 1 & 1 & 1 & 1 & 0 & 1 & 1 \\ 0 & 0 & 1 & 1 & 0 & 1 & 0 & 1 \\ 1 & 1 & 1 & 0 & 1 & 0 & 1 & 1 \\ 0 & 1 & 1 & 1 & 0 & 1 & 1 & 1 \\ 1 & 1 & 0 & 1 & 1 & 1 & 1 & 0 \end{bmatrix}, \quad \mathbf{A} = \begin{bmatrix} 1 & 1 & 1 & 0 & 1 & 1 & 0 & 0 \\ 1 & 0 & 1 & 1 & 1 & 0 & 1 & 0 \\ 1 & 1 & 0 & 1 & 0 & 1 & 0 & 1 \\ 0 & 1 & 1 & 1 & 0 & 0 & 1 & 1 \\ 1 & 1 & 0 & 0 & 1 & 1 & 1 & 0 \\ 1 & 0 & 1 & 0 & 1 & 0 & 1 & 1 \\ 0 & 1 & 0 & 1 & 1 & 1 & 0 & 1 \\ 0 & 0 & 1 & 1 & 0 & 1 & 1 & 1 \end{bmatrix}$$

Thus, the takeaway is that, even for small-medium scale homophilic and heterophilic SSBM graphs, random graphs that satisfy condition $\mathbf{C}$ are rarely sampled.

# F   Proof of Theorem 3.3

In this appendix, we analyse the gradient flow stated in (14) and establish the results stated in Theorem 3.3. Our analysis is inspired by the one in [56], but significantly differs from it, as we need to overcome the complexity introduced by the graph structure matrix.

Based on the setup of gUFM for GNN $\psi^{\mathcal{F}'}$, we analyze the risk $\widehat{\mathcal{R}}^{\mathcal{F}'}$ given as follows:

$$\widehat{\mathcal{R}}^{\mathcal{F}'}(\mathbf{W}_2, \{\mathbf{H}_k\}_{k=1}^K) := \frac{1}{K} \sum_{k=1}^K \left( \frac{1}{2N} \left\| \mathbf{W}_2 \mathbf{H}_k \widehat{\mathbf{A}}_k - \mathbf{Y} \right\|_F^2 + \frac{\lambda_{H_k}}{2} \|\mathbf{H}_k\|_2^2 \right) + \frac{\lambda_{W_2}}{2} \|\mathbf{W}_2\|_2^2$$

By taking the derivatives of $\widehat{\mathcal{R}}^{\mathcal{F}'}$ with respect to $(\mathbf{W}_2, \mathbf{H}_k)$, we get:

$$\frac{\partial \widehat{\mathcal{R}}^{\mathcal{F}'}}{\partial \mathbf{W}_2} = \frac{1}{KN} \sum_{k=1}^K (\mathbf{W}_2 \mathbf{H}_k \widehat{\mathbf{A}}_k - \mathbf{Y})(\mathbf{H}_k \widehat{\mathbf{A}}_k)^\top + \lambda_{W_2} \mathbf{W}_2$$

$$\frac{\partial \widehat{\mathcal{R}}^{\mathcal{F}'}}{\partial \mathbf{H}_k} = \frac{1}{KN} \left[ \mathbf{W}_2^\top (\mathbf{W}_2 \mathbf{H}_k \widehat{\mathbf{A}}_k - \mathbf{Y}) \widehat{\mathbf{A}}_k^\top \right] + \frac{1}{K} \lambda_{H_k} \mathbf{H}_k$$

Now, by setting $\frac{\partial \widehat{\mathcal{R}}^{\mathcal{F}'}}{\partial \mathbf{W}_2} = 0$ we get the closed form representation of $\mathbf{W}_2$ in terms of $\mathbf{H}$.

$$\frac{\partial \widehat{\mathcal{R}}^{\mathcal{F}'}}{\partial \mathbf{W}_2} = \frac{1}{KN} \sum_{k=1}^K (\mathbf{W}_2 \mathbf{H}_k \widehat{\mathbf{A}}_k - \mathbf{Y})(\mathbf{H}_k \widehat{\mathbf{A}}_k)^\top + \lambda_{W_2} \mathbf{W}_2 = 0$$

$$\implies \frac{1}{K} \sum_{k=1}^K \left( \mathbf{W}_2 \mathbf{H}_k \widehat{\mathbf{A}}_k \widehat{\mathbf{A}}_k^\top \mathbf{H}_k^\top - \mathbf{Y} \widehat{\mathbf{A}}_k^\top \mathbf{H}_k^\top \right) + \lambda_{W_2} N \mathbf{W}_2 = 0$$

$$\implies \mathbf{W}_2 \left[ \frac{1}{K} \sum_{k=1}^K \mathbf{H}_k \widehat{\mathbf{A}}_k \widehat{\mathbf{A}}_k^\top \mathbf{H}_k^\top + \lambda_{W_2} N \mathbf{I} \right] = \frac{1}{K} \sum_{k=1}^K \mathbf{Y} \widehat{\mathbf{A}}_k^\top \mathbf{H}_k^\top$$

Thus, the ideal value of $\mathbf{W}_2^*$ is given by:

$$\mathbf{W}_2^* = \left[ \frac{1}{K} \sum_{k=1}^K \mathbf{Y} \widehat{\mathbf{A}}_k^\top \mathbf{H}_k^\top \right] \left[ \frac{1}{K} \sum_{k=1}^K \mathbf{H}_k \widehat{\mathbf{A}}_k \widehat{\mathbf{A}}_k^\top \mathbf{H}_k^\top + \lambda_{W_2} N \mathbf{I} \right]^{-1}$$

Now, under the assumption that $K = 1, C = 2$, we drop the subscript for $\mathbf{H}_k, \widehat{\mathbf{A}}_k$ to get:

$$\mathbf{W}_2^* = \left( \mathbf{Y} \widehat{\mathbf{A}}^\top \mathbf{H}^\top \right) \left( \mathbf{H} \widehat{\mathbf{A}} \widehat{\mathbf{A}}^\top \mathbf{H}^\top + \lambda_{W_2} N \mathbf{I} \right)^{-1} \tag{82}$$

To this end, we return to our risk minimization formulation for a single graph with these optimal values as follows:

$$\widehat{\mathcal{R}}^{\mathcal{F}'}(\mathbf{H}) = \frac{1}{2N} \left\| \mathbf{W}_2^* \mathbf{H} \widehat{\mathbf{A}} - \mathbf{Y} \right\|_F^2 + \frac{\lambda_{W_2}}{2} \|\mathbf{W}_2^*\|_2^2 + \frac{\lambda_H}{2} \|\mathbf{H}\|_2^2 \tag{83}$$

## F.1   $\widehat{\mathbf{A}}$ as a perturbation of $\mathbb{E}\widehat{\mathbf{A}}$

Since we are dealing with SSBM graphs, we can formulate $\widehat{\mathbf{A}}$ as the perturbed version of its expected value. Formally, $\widehat{\mathbf{A}} = \mathbb{E}\widehat{\mathbf{A}} + \mathbf{E}$ where $\mathbb{E}\widehat{\mathbf{A}} \in \mathbb{R}^{N \times N}$ is the expected normalized adjacency matrix and $\mathbf{E} \in \mathbb{R}^{N \times N}$ is the perturbation matrix. $\mathbb{E}\widehat{\mathbf{A}}$ can be written in block matrix form as:

$$\mathbb{E}\widehat{\mathbf{A}} = \frac{1}{np + nq} \begin{bmatrix} p\mathbf{1}_n\mathbf{1}_n^\top & q\mathbf{1}_n\mathbf{1}_n^\top \\ q\mathbf{1}_n\mathbf{1}_n^\top & p\mathbf{1}_n\mathbf{1}_n^\top \end{bmatrix}_{N \times N} \tag{84}$$

Where the $\mathbb{E}\widehat{\mathbf{A}}$ has eigenvalues $1, \frac{p-q}{p+q}$ associated with eigenvectors $\frac{1}{\sqrt{N}}\mathbf{1}$ and $\mathbf{c} = \frac{1}{\sqrt{N}}\begin{bmatrix} \mathbf{1}_n \\ -\mathbf{1}_n \end{bmatrix}_N$ respectively. Thus, we can represent $\mathbb{E}\widehat{\mathbf{A}}$ using its spectral information as follows:

$$
\begin{aligned}
\mathbb{E}\widehat{\mathbf{A}} &= \frac{2}{N(p+q)}\left( \frac{p+q}{2}\mathbf{1}_N\mathbf{1}_N^\top + \frac{p-q}{2}\mathbf{c}\mathbf{c}^\top \right) \\
&= \begin{bmatrix} \mathbf{1}_N & \mathbf{c} \end{bmatrix}\begin{bmatrix} \alpha_1 & 0 \\ 0 & \alpha_2 \end{bmatrix}\begin{bmatrix} \mathbf{1}_N & \mathbf{c} \end{bmatrix}^\top \\
&= \begin{bmatrix} \mathbf{1}_N & \mathbf{c} \end{bmatrix}\begin{bmatrix} \sqrt{\alpha_1} & 0 \\ 0 & \sqrt{\alpha_2} \end{bmatrix}\begin{bmatrix} \sqrt{\alpha_1} & 0 \\ 0 & \sqrt{\alpha_2} \end{bmatrix}^\top \begin{bmatrix} \mathbf{1}_N & \mathbf{c} \end{bmatrix}^\top \\
&= \mathbf{Q}\mathbf{Q}^\top
\end{aligned}
$$
(85)

Where $\alpha_1 = \frac{1}{N}, \alpha_2 = \frac{p-q}{(p+q)N}$, and $\mathbf{Q} = \begin{bmatrix} \mathbf{1}_N & \mathbf{c} \end{bmatrix}\begin{bmatrix} \sqrt{\alpha_1} & 0 \\ 0 & \sqrt{\alpha_2} \end{bmatrix} \in \mathbb{R}^{N\times 2}$ is the factor matrix.

### F.2 Preliminary results

Let us recall the definitions of the within- and between-class covariance matrices, $\boldsymbol{\Sigma}_W(\mathbf{H})$ and $\boldsymbol{\Sigma}_B(\mathbf{H})$, for computing the preliminary results:

$$
\boldsymbol{\Sigma}_W(\mathbf{H}) := \frac{1}{Cn}\sum_{c=1}^{C}\sum_{i=1}^{n}\left( \mathbf{h}_{c,i} - \overline{\mathbf{h}}_c \right)\left( \mathbf{h}_{c,i} - \overline{\mathbf{h}}_c \right)^\top,
$$

$$
\boldsymbol{\Sigma}_B(\mathbf{H}) := \frac{1}{C}\sum_{c=1}^{C}\left( \overline{\mathbf{h}}_c - \overline{\mathbf{h}}_G \right)\left( \overline{\mathbf{h}}_c - \overline{\mathbf{h}}_G \right)^\top.
$$

Here, the class means $\overline{\mathbf{h}}_c, \forall c \in [C]$ and the global mean $\overline{\mathbf{h}}_G$ of $\mathbf{H}$ are given as follows:

$$
\overline{\mathbf{h}}_c := \frac{1}{n}\sum_{i=1}^{n}\mathbf{h}_{c,i}, \forall c \in [C], \qquad \overline{\mathbf{h}}_G := \frac{1}{Cn}\sum_{c=1}^{C}\sum_{i=1}^{n}\mathbf{h}_{c,i}.
$$

### F.2.1 Relating $\widetilde{\boldsymbol{\Sigma}}_B(\mathbf{H}), \boldsymbol{\Sigma}_B(\mathbf{H})$

Since $C = 2$ in our analysis and $\overline{\mathbf{h}}_G = \frac{\overline{\mathbf{h}}_1 + \overline{\mathbf{h}}_2}{2}$ due to balanced communities, note that:

$$
\begin{aligned}
\boldsymbol{\Sigma}_B(\mathbf{H}) &= \frac{1}{2}\sum_{c=1}^{2}\left( \overline{\mathbf{h}}_c - \overline{\mathbf{h}}_G \right)\left( \overline{\mathbf{h}}_c - \overline{\mathbf{h}}_G \right)^\top \\
&= \frac{1}{2}\left( \left( \overline{\mathbf{h}}_1 - \overline{\mathbf{h}}_G \right)\left( \overline{\mathbf{h}}_1 - \overline{\mathbf{h}}_G \right)^\top + \left( \overline{\mathbf{h}}_2 - \overline{\mathbf{h}}_G \right)\left( \overline{\mathbf{h}}_2 - \overline{\mathbf{h}}_G \right)^\top \right) \\
&= \frac{1}{4}\left( \overline{\mathbf{h}}_1 - \overline{\mathbf{h}}_2 \right)\left( \overline{\mathbf{h}}_1 - \overline{\mathbf{h}}_2 \right)^\top \\
&= \frac{1}{4}\left( \overline{\mathbf{h}}_1\overline{\mathbf{h}}_1^\top + \overline{\mathbf{h}}_2\overline{\mathbf{h}}_2^\top - \overline{\mathbf{h}}_1\overline{\mathbf{h}}_2^\top - \overline{\mathbf{h}}_2\overline{\mathbf{h}}_1^\top \right) \\
&= \frac{1}{4}\left( 2\widetilde{\boldsymbol{\Sigma}}_B(\mathbf{H}) - \overline{\mathbf{h}}_1\overline{\mathbf{h}}_2^\top - \overline{\mathbf{h}}_2\overline{\mathbf{h}}_1^\top \right)
\end{aligned}
$$

Thus, we get the following relation between $\widetilde{\boldsymbol{\Sigma}}_B(\mathbf{H}), \boldsymbol{\Sigma}_B(\mathbf{H})$:

$$
2\widetilde{\boldsymbol{\Sigma}}_B(\mathbf{H}) - 4\boldsymbol{\Sigma}_B(\mathbf{H}) = \overline{\mathbf{h}}_1\overline{\mathbf{h}}_2^\top + \overline{\mathbf{h}}_2\overline{\mathbf{h}}_1^\top
$$
(86)

Additionally, we can extend this result to the following:

$$
\begin{aligned}
2\widetilde{\boldsymbol{\Sigma}}_B(\mathbf{H}) - 4\boldsymbol{\Sigma}_B(\mathbf{H}) &= \overline{\mathbf{h}}_1\overline{\mathbf{h}}_2^\top + \overline{\mathbf{h}}_2\overline{\mathbf{h}}_1^\top \\
&= \left( 2\overline{\mathbf{h}}_G - \overline{\mathbf{h}}_2 \right)\overline{\mathbf{h}}_2^\top + \left( 2\overline{\mathbf{h}}_G - \overline{\mathbf{h}}_1 \right)\overline{\mathbf{h}}_1^\top \\
&= 2\overline{\mathbf{h}}_G\left( \overline{\mathbf{h}}_1^\top + \overline{\mathbf{h}}_2^\top \right) - \overline{\mathbf{h}}_1\overline{\mathbf{h}}_1^\top - \overline{\mathbf{h}}_2\overline{\mathbf{h}}_2^\top \\
&= 4\boldsymbol{\Sigma}_G(\mathbf{H}) - 2\widetilde{\boldsymbol{\Sigma}}_B(\mathbf{H})
\end{aligned}
$$

Where $\Sigma_G = \overline{\mathbf{h}}_G \overline{\mathbf{h}}_G^\top$. This gives us:

$$\widetilde{\Sigma}_B(\mathbf{H}) - \Sigma_G(\mathbf{H}) = \Sigma_B(\mathbf{H}) \tag{87}$$

### F.2.2 Expanding $\mathbf{H}\widehat{\mathbf{A}}\mathbf{H}^\top$

Based on the perturbed representation of $\widehat{\mathbf{A}} = \mathbb{E}\widehat{\mathbf{A}} + \mathbf{E}$, we can modify $\mathbf{H}\widehat{\mathbf{A}}\mathbf{H}^\top$ and $\mathbf{H}\widehat{\mathbf{A}}^\top\mathbf{H}^\top$ as:

$$\mathbf{H}\widehat{\mathbf{A}}\mathbf{H}^\top = \mathbf{H}\left[\mathbb{E}\widehat{\mathbf{A}}\right]\mathbf{H}^\top + \mathbf{H}\mathbf{E}\mathbf{H}^\top$$
$$= [\mathbf{H}\mathbf{Q}][\mathbf{H}\mathbf{Q}]^\top + \mathbf{H}\mathbf{E}\mathbf{H}^\top$$
$$\mathbf{H}\widehat{\mathbf{A}}^\top\mathbf{H}^\top = \mathbf{H}\left[\mathbb{E}\widehat{\mathbf{A}}\right]^\top\mathbf{H}^\top + \mathbf{H}\mathbf{E}^\top\mathbf{H}^\top$$
$$= \mathbf{H}\left[\mathbb{E}\widehat{\mathbf{A}}\right]\mathbf{H}^\top + \mathbf{H}\mathbf{E}^\top\mathbf{H}^\top$$
$$= [\mathbf{H}\mathbf{Q}][\mathbf{H}\mathbf{Q}]^\top + \mathbf{H}\mathbf{E}^\top\mathbf{H}^\top$$

Observe that $\mathbf{H}\mathbf{Q}$ can be broken down in terms of $\mathbf{h}_{c,i}, c \in [C], i \in [n]$ as follows:

$$\mathbf{H}\mathbf{Q} = [\mathbf{h}_{1,1} \quad \cdots \quad \mathbf{h}_{1,n} \quad \mathbf{h}_{2,1} \quad \cdots \quad \mathbf{h}_{2,n}]_{d_L-1 \times N} [\mathbf{1}_N \quad \mathbf{c}]_{N \times 2} \begin{bmatrix} \sqrt{\alpha_1} & 0 \\ 0 & \sqrt{\alpha_2} \end{bmatrix}_{2 \times 2}$$
$$= \left[\sqrt{\alpha_1} \sum_{c=1}^2 \sum_{i=1}^n \mathbf{h}_{c,i} \quad \sqrt{\alpha_2}\left(\sum_{i=1}^n \mathbf{h}_{1,i} - \sum_{i=1}^n \mathbf{h}_{2,i}\right)\right]$$
$$= \left[2n\sqrt{\alpha_1}\overline{\mathbf{h}}_G \quad n\sqrt{\alpha_2}\left(\overline{\mathbf{h}}_1 - \overline{\mathbf{h}}_2\right)\right]$$

This leads to the following expansion of $[\mathbf{H}\mathbf{Q}][\mathbf{H}\mathbf{Q}]^\top$:

$$[\mathbf{H}\mathbf{Q}][\mathbf{H}\mathbf{Q}]^\top = \left[2n\sqrt{\alpha_1}\overline{\mathbf{h}}_G \quad n\sqrt{\alpha_2}\left(\overline{\mathbf{h}}_1 - \overline{\mathbf{h}}_2\right)\right] \begin{bmatrix} 2n\sqrt{\alpha_1}\overline{\mathbf{h}}_G^\top \\ n\sqrt{\alpha_2}\left(\overline{\mathbf{h}}_1 - \overline{\mathbf{h}}_2\right)^\top \end{bmatrix}$$
$$= 4n^2\alpha_1\overline{\mathbf{h}}_G\overline{\mathbf{h}}_G^\top + n^2\alpha_2\left(\overline{\mathbf{h}}_1 - \overline{\mathbf{h}}_2\right)\left(\overline{\mathbf{h}}_1 - \overline{\mathbf{h}}_2\right)^\top \tag{88}$$
$$= 4n^2\alpha_1\overline{\mathbf{h}}_G\overline{\mathbf{h}}_G^\top + 4n^2\alpha_2\left(\overline{\mathbf{h}}_1 - \overline{\mathbf{h}}_G\right)\left(\overline{\mathbf{h}}_1 - \overline{\mathbf{h}}_G\right)^\top$$

Since $\overline{\mathbf{h}}_G = \frac{\overline{\mathbf{h}}_1 + \overline{\mathbf{h}}_2}{2}$ (due to balanced classes). This also implies that:

$$[\mathbf{H}\mathbf{Q}][\mathbf{H}\mathbf{Q}]^\top = 4n^2\alpha_1\overline{\mathbf{h}}_G\overline{\mathbf{h}}_G^\top + 4n^2\alpha_2\left(\overline{\mathbf{h}}_2 - \overline{\mathbf{h}}_G\right)\left(\overline{\mathbf{h}}_2 - \overline{\mathbf{h}}_G\right)^\top \tag{89}$$

Thus, by taking the average of values in equation 88, 89, we get:

$$[\mathbf{H}\mathbf{Q}][\mathbf{H}\mathbf{Q}]^\top = 4n^2\alpha_1\Sigma_G(\mathbf{H}) + 4n^2\alpha_2\Sigma_B(\mathbf{H})$$
$$= 4n^2\left[\frac{1}{N}\Sigma_G(\mathbf{H}) + \frac{(p-q)}{(p+q)N}\Sigma_B(\mathbf{H})\right]$$
$$= 2n\left[\widetilde{\Sigma}_B(\mathbf{H}) - \Sigma_B(\mathbf{H}) + \frac{(p-q)}{(p+q)}\Sigma_B(\mathbf{H})\right]$$
$$= 2n\left[\widetilde{\Sigma}_B(\mathbf{H}) - \frac{2q}{(p+q)}\Sigma_B(\mathbf{H})\right]$$

Finally, $\mathbf{H}\widehat{\mathbf{A}}\mathbf{H}^\top$ can be simplified to:

$$\mathbf{H}\widehat{\mathbf{A}}\mathbf{H}^\top = 2n\left[\widetilde{\Sigma}_B(\mathbf{H}) - \frac{2q}{(p+q)}\Sigma_B(\mathbf{H})\right] + \Delta_1 \tag{90}$$

Where $\Delta_1 = \mathbf{H}\mathbf{E}\mathbf{H}^\top$ corresponds to the first order perturbation term. Similarly,

$$\mathbf{H}\widehat{\mathbf{A}}^\top\mathbf{H}^\top = 2n\left[\widetilde{\Sigma}_B(\mathbf{H}) - \frac{2q}{(p+q)}\Sigma_B(\mathbf{H})\right] + \Delta_1^\top \tag{91}$$

### F.2.3 Expanding $\mathbf{H}\widehat{\mathbf{A}}\widehat{\mathbf{A}}^\top\mathbf{H}^\top$

Expanding $\widehat{\mathbf{A}}\widehat{\mathbf{A}}^\top$ can be done along the same lines:

$$\widehat{\mathbf{A}}\widehat{\mathbf{A}}^\top = \left[\mathbb{E}\widehat{\mathbf{A}} + \mathbf{E}\right]\left[\mathbb{E}\widehat{\mathbf{A}} + \mathbf{E}\right]^\top = \left[\mathbb{E}\widehat{\mathbf{A}}\right]^2 + \mathbf{E}\mathbf{E}^\top + \mathbf{E}\left[\mathbb{E}\widehat{\mathbf{A}}\right] + \left[\mathbb{E}\widehat{\mathbf{A}}\right]\mathbf{E}^\top$$

$$\left[\mathbb{E}\widehat{\mathbf{A}}\right]^2 = [\mathbf{1}_N \quad \mathbf{c}]\begin{bmatrix} \alpha_1 & 0 \\ 0 & \alpha_2 \end{bmatrix}[\mathbf{1}_N \quad \mathbf{c}]^\top[\mathbf{1}_N \quad \mathbf{c}]\begin{bmatrix} \alpha_1 & 0 \\ 0 & \alpha_2 \end{bmatrix}[\mathbf{1}_N \quad \mathbf{c}]^\top$$

$$= [\mathbf{1}_N \quad \mathbf{c}]\begin{bmatrix} \alpha_1 & 0 \\ 0 & \alpha_2 \end{bmatrix}\begin{bmatrix} N\alpha_1 & 0 \\ 0 & N\alpha_2 \end{bmatrix}[\mathbf{1}_N \quad \mathbf{c}]^\top$$

$$= \mathbf{Q}\begin{bmatrix} \sqrt{\alpha_1 N} & 0 \\ 0 & \sqrt{\alpha_2 N} \end{bmatrix}\begin{bmatrix} \sqrt{\alpha_1 N} & 0 \\ 0 & \sqrt{\alpha_2 N} \end{bmatrix}^\top \mathbf{Q}^\top$$

Based on the formulations above, $\mathbf{H}\left[\mathbb{E}\widehat{\mathbf{A}}\right]^2\mathbf{H}^\top$ can be given by:

$$\mathbf{H}\left[\mathbb{E}\widehat{\mathbf{A}}\right]^2\mathbf{H}^\top = \left[\mathbf{HQ}\begin{bmatrix} \sqrt{\alpha_1 N} & 0 \\ 0 & \sqrt{\alpha_2 N} \end{bmatrix}\right]\left[\mathbf{HQ}\begin{bmatrix} \sqrt{\alpha_1 N} & 0 \\ 0 & \sqrt{\alpha_2 N} \end{bmatrix}\right]^\top$$

$$= [2n\alpha_1\sqrt{N}\overline{\mathbf{h}}_G \quad n\alpha_2\sqrt{N}\,(\overline{\mathbf{h}}_1 - \overline{\mathbf{h}}_2)]\begin{bmatrix} 2n\alpha_1\sqrt{N}\overline{\mathbf{h}}_G^\top \\ n\alpha_2\sqrt{N}\,(\overline{\mathbf{h}}_1 - \overline{\mathbf{h}}_2)^\top \end{bmatrix} \tag{92}$$

$$= 4n^2\alpha_1^2 N\overline{\mathbf{h}}_G\overline{\mathbf{h}}_G^\top + n^2\alpha_2^2 N\,(\overline{\mathbf{h}}_1 - \overline{\mathbf{h}}_2)\,(\overline{\mathbf{h}}_1 - \overline{\mathbf{h}}_2)^\top$$

$$= 4n^2\alpha_1^2 N\overline{\mathbf{h}}_G\overline{\mathbf{h}}_G^\top + 4n^2\alpha_2^2 N\,(\overline{\mathbf{h}}_1 - \overline{\mathbf{h}}_G)\,(\overline{\mathbf{h}}_1 - \overline{\mathbf{h}}_G)^\top$$

Since $\overline{\mathbf{h}}_G = \frac{\overline{\mathbf{h}}_1 + \overline{\mathbf{h}}_2}{2}$ (due to balanced classes). This also implies that:

$$\mathbf{H}\left[\mathbb{E}\widehat{\mathbf{A}}\right]^2\mathbf{H}^\top = 4n^2\alpha_1^2 N\overline{\mathbf{h}}_G\overline{\mathbf{h}}_G^\top + 4n^2\alpha_2^2 N\,(\overline{\mathbf{h}}_2 - \overline{\mathbf{h}}_G)\,(\overline{\mathbf{h}}_2 - \overline{\mathbf{h}}_G)^\top \tag{93}$$

Thus, based on taking the average of values in equation 92, 93, we get:

$$\mathbf{H}\left[\mathbb{E}\widehat{\mathbf{A}}\right]^2\mathbf{H}^\top = 4n^2\alpha_1^2 N\mathbf{\Sigma}_G(\mathbf{H}) + 4n^2\alpha_2^2 N\mathbf{\Sigma}_B(\mathbf{H})$$

$$= 8n^3\left[\frac{1}{N^2}\mathbf{\Sigma}_G(\mathbf{H}) + \frac{(p-q)^2}{N^2(p+q)^2}\mathbf{\Sigma}_B(\mathbf{H})\right]$$

$$= 2n\left[\widetilde{\mathbf{\Sigma}}_B(\mathbf{H}) - \frac{4pq}{(p+q)^2}\mathbf{\Sigma}_B(\mathbf{H})\right]$$

Finally, $\mathbf{H}\widehat{\mathbf{A}}\widehat{\mathbf{A}}^\top\mathbf{H}^\top$ can be simplified to:

$$\mathbf{H}\widehat{\mathbf{A}}\widehat{\mathbf{A}}^\top\mathbf{H}^\top = 2n\left[\widetilde{\mathbf{\Sigma}}_B(\mathbf{H}) - \frac{4pq}{(p+q)^2}\mathbf{\Sigma}_B(\mathbf{H})\right] + \Delta_2 \tag{94}$$

Where $\Delta_2 = \mathbf{H}\left[\mathbf{E}\mathbf{E}^\top + \mathbf{E}\left[\mathbb{E}\widehat{\mathbf{A}}\right] + \left[\mathbb{E}\widehat{\mathbf{A}}\right]\mathbf{E}^\top\right]\mathbf{H}^\top$ is a symmetric matrix corresponding to the first and second order perturbation terms.

### F.2.4 Expanding $\mathbf{Y}\widehat{\mathbf{A}}^\top\mathbf{H}^\top, \mathbf{Y}\mathbf{H}^\top$

We follow similar line of expansions for $\mathbf{Y}\widehat{\mathbf{A}}^\top\mathbf{H}^\top$ to get:

$$\mathbf{Y}\widehat{\mathbf{A}}^\top\mathbf{H}^\top = \mathbf{Y}\left[\mathbb{E}\widehat{\mathbf{A}}\right]\mathbf{H}^\top + \mathbf{Y}\mathbf{E}^\top\mathbf{H}^\top = (\mathbf{I}_2 \otimes \mathbf{1}_n^\top)\left[\mathbb{E}\widehat{\mathbf{A}}\right]\mathbf{H}^\top + \mathbf{Y}\mathbf{E}^\top\mathbf{H}^\top$$

$$= (\mathbf{I}_2 \otimes \mathbf{1}_n^\top)\frac{1}{np + nq}\begin{bmatrix} np\overline{\mathbf{h}}_1^\top + nq\overline{\mathbf{h}}_2^\top \\ \vdots \\ nq\overline{\mathbf{h}}_1^\top + np\overline{\mathbf{h}}_2^\top \\ \vdots \end{bmatrix} + \mathbf{Y}\mathbf{E}^\top\mathbf{H}^\top \tag{95}$$

$$= \frac{n}{p+q}\begin{bmatrix} p\overline{\mathbf{h}}_1^\top + q\overline{\mathbf{h}}_2^\top \\ q\overline{\mathbf{h}}_1^\top + p\overline{\mathbf{h}}_2^\top \end{bmatrix} + \Delta_3$$

Where $\Delta_3 = \mathbf{Y}\mathbf{E}^\top\mathbf{H}^\top$ is the first order perturbation term. Next, observe that:

$$\mathbf{Y}\mathbf{H}^\top = (\mathbf{I}_2 \otimes \mathbf{1}_n^\top)\mathbf{H}^\top = n\overline{\mathbf{H}}^\top \tag{96}$$

## F.2.5 Expanding $\mathbf{H}\widehat{\mathbf{A}}\mathbf{Y}^\top\mathbf{Y}\widehat{\mathbf{A}}^\top\mathbf{H}^\top$

$$
\begin{aligned}
\mathbf{H}\widehat{\mathbf{A}}\mathbf{Y}^\top\mathbf{Y}\widehat{\mathbf{A}}^\top\mathbf{H}^\top &= \mathbf{H}\left[\mathbb{E}\widehat{\mathbf{A}} + \mathbf{E}\right]\mathbf{Y}^\top\mathbf{Y}\left[\mathbb{E}\widehat{\mathbf{A}} + \mathbf{E}\right]^\top\mathbf{H}^\top \\
&= \left(\frac{n}{p+q}\begin{bmatrix} p\overline{\mathbf{h}}_1^\top + q\overline{\mathbf{h}}_2^\top \\ q\overline{\mathbf{h}}_1^\top + p\overline{\mathbf{h}}_2^\top \end{bmatrix}^\top + \Delta_3^\top\right)\left(\frac{n}{p+q}\begin{bmatrix} p\overline{\mathbf{h}}_1^\top + q\overline{\mathbf{h}}_2^\top \\ q\overline{\mathbf{h}}_1^\top + p\overline{\mathbf{h}}_2^\top \end{bmatrix} + \Delta_3\right) \\
&= \frac{n^2}{(p+q)^2}\begin{bmatrix} p\overline{\mathbf{h}}_1 + q\overline{\mathbf{h}}_2 & q\overline{\mathbf{h}}_1 + p\overline{\mathbf{h}}_2 \end{bmatrix}\begin{bmatrix} p\overline{\mathbf{h}}_1^\top + q\overline{\mathbf{h}}_2^\top \\ q\overline{\mathbf{h}}_1^\top + p\overline{\mathbf{h}}_2^\top \end{bmatrix} + \widetilde{\Delta}_3 \\
&= \frac{n^2}{(p+q)^2}\left[(p^2 + q^2)\left(\overline{\mathbf{h}}_1\overline{\mathbf{h}}_1^\top + \overline{\mathbf{h}}_2\overline{\mathbf{h}}_2^\top\right) + (2pq)\left(\overline{\mathbf{h}}_1\overline{\mathbf{h}}_2^\top + \overline{\mathbf{h}}_2\overline{\mathbf{h}}_1^\top\right)\right] + \widetilde{\Delta}_3 \\
&\stackrel{(a)}{=} \frac{n^2}{(p+q)^2}\left[2(p^2+q^2)\widetilde{\boldsymbol{\Sigma}}_B(\mathbf{H}) + (2pq)\left(2\widetilde{\boldsymbol{\Sigma}}_B(\mathbf{H}) - 4\boldsymbol{\Sigma}_B(\mathbf{H})\right)\right] + \widetilde{\Delta}_3 \\
&= 2n^2\left[\widetilde{\boldsymbol{\Sigma}}_B(\mathbf{H}) - \frac{4pq}{(p+q)^2}\boldsymbol{\Sigma}_B(\mathbf{H})\right] + \widetilde{\Delta}_3
\end{aligned}
$$

$$\tag{97}$$

Where $\widetilde{\Delta}_3 = \mathbf{H}\left(\mathbf{E}\mathbf{Y}^\top\mathbf{Y}\mathbf{E}^\top + \left[\mathbb{E}\widehat{\mathbf{A}}\right]\mathbf{Y}^\top\mathbf{Y}\mathbf{E}^\top + \mathbf{E}\mathbf{Y}^\top\mathbf{Y}\left[\mathbb{E}\widehat{\mathbf{A}}\right]\right)\mathbf{H}^\top$ and the equality $(a)$ is based on equation 86.

## F.3 Trace formulation of risk

Now, note that the risk can be formulated in terms of matrix traces as follows:

$$
\widehat{\mathcal{R}}^{\mathcal{F}'}(\mathbf{H}) = \frac{1}{2N}\mathrm{Tr}\left\{\left(\mathbf{W}_2^*\mathbf{H}\widehat{\mathbf{A}} - \mathbf{Y}\right)\left(\mathbf{W}_2^*\mathbf{H}\widehat{\mathbf{A}} - \mathbf{Y}\right)^\top\right\} + \frac{\lambda_{W_2}}{2}\mathrm{Tr}\left\{\mathbf{W}_2^*\mathbf{W}_2^{*\top}\right\} + \\
\frac{\lambda_H}{2}\mathrm{Tr}\left\{\mathbf{H}\mathbf{H}^\top\right\} \tag{98}
$$

Where the term $\left(\mathbf{W}_2^*\mathbf{H}\widehat{\mathbf{A}} - \mathbf{Y}\right)\left(\mathbf{W}_2^*\mathbf{H}\widehat{\mathbf{A}} - \mathbf{Y}\right)^\top$ can be expanded as follows:

$$
\begin{aligned}
\left(\mathbf{W}_2^*\mathbf{H}\widehat{\mathbf{A}} - \mathbf{Y}\right)\left(\mathbf{W}_2^*\mathbf{H}\widehat{\mathbf{A}} - \mathbf{Y}\right)^\top &= \mathbf{W}_2^*\mathbf{H}\widehat{\mathbf{A}}\widehat{\mathbf{A}}^\top\mathbf{H}^\top\mathbf{W}_2^{*\top} - \mathbf{W}_2^*\mathbf{H}\widehat{\mathbf{A}}\mathbf{Y}^\top \\
&\quad - \mathbf{Y}\widehat{\mathbf{A}}^\top\mathbf{H}^\top\mathbf{W}_2^{*\top} + \mathbf{Y}\mathbf{Y}^\top
\end{aligned}
$$

Since $\mathbf{W}_2^*\left[\mathbf{H}\widehat{\mathbf{A}}\widehat{\mathbf{A}}^\top\mathbf{H}^\top + \lambda_{W_2}N\mathbf{I}\right] = \mathbf{Y}\widehat{\mathbf{A}}^\top\mathbf{H}^\top$, we can multiply $\mathbf{W}_2^{*\top}$ on both sides and get:

$$
\mathbf{W}_2^*\mathbf{H}\widehat{\mathbf{A}}\widehat{\mathbf{A}}^\top\mathbf{H}^\top\mathbf{W}_2^{*\top} = \mathbf{Y}\widehat{\mathbf{A}}^\top\mathbf{H}^\top\mathbf{W}_2^{*\top} - \lambda_{W_2}N\mathbf{W}_2^*\mathbf{W}_2^{*\top}
$$

Using these simplifications and matrix trace properties, the risk can be modified as:

$$
\begin{aligned}
&= \frac{1}{2N}\mathrm{Tr}\left\{\left(\mathbf{W}_2^*\mathbf{H}\widehat{\mathbf{A}} - \mathbf{Y}\right)\left(\mathbf{W}_2^*\mathbf{H}\widehat{\mathbf{A}} - \mathbf{Y}\right)^\top\right\} + \frac{\lambda_{W_2}}{2}\mathrm{Tr}\left\{\mathbf{W}_2^*\mathbf{W}_2^{*\top}\right\} + \frac{\lambda_H}{2}\mathrm{Tr}\left\{\mathbf{H}\mathbf{H}^\top\right\} \\
&= \frac{1}{2N}\mathrm{Tr}\left\{-\mathbf{W}_2^*\mathbf{H}\widehat{\mathbf{A}}\mathbf{Y}^\top + \mathbf{Y}\mathbf{Y}^\top\right\} + \frac{\lambda_H}{2}\mathrm{Tr}\left\{\mathbf{H}\mathbf{H}^\top\right\} \\
&= \frac{1}{2N}\mathrm{Tr}\left\{-\mathbf{Y}\widehat{\mathbf{A}}^\top\mathbf{H}^\top\left[\mathbf{H}\widehat{\mathbf{A}}\widehat{\mathbf{A}}^\top\mathbf{H}^\top + \lambda_{W_2}N\mathbf{I}\right]^{-1}\mathbf{H}\widehat{\mathbf{A}}\mathbf{Y}^\top + \mathbf{Y}\mathbf{Y}^\top\right\} + \frac{\lambda_H}{2}\mathrm{Tr}\left\{\mathbf{H}\mathbf{H}^\top\right\} \\
&= \frac{1}{2N}\mathrm{Tr}\left\{-\mathbf{H}\widehat{\mathbf{A}}\mathbf{Y}^\top\mathbf{Y}\widehat{\mathbf{A}}^\top\mathbf{H}^\top\left[\mathbf{H}\widehat{\mathbf{A}}\widehat{\mathbf{A}}^\top\mathbf{H}^\top + \lambda_{W_2}N\mathbf{I}\right]^{-1}\right\} \\
&\quad + \frac{1}{2N}\mathrm{Tr}\left\{\mathbf{Y}\mathbf{Y}^\top\right\} + \frac{\lambda_H}{2}\mathrm{Tr}\left\{\mathbf{H}\mathbf{H}^\top\right\}
\end{aligned}
$$

Where the covariance matrix formulations for $\mathbf{H}\widehat{\mathbf{A}}\widehat{\mathbf{A}}^\top\mathbf{H}^\top, \mathbf{H}\widehat{\mathbf{A}}\mathbf{Y}^\top\mathbf{Y}\widehat{\mathbf{A}}^\top\mathbf{H}^\top$ can be leveraged to formulate the risk as:

$$
\begin{aligned}
\widehat{\mathcal{R}}^{\mathcal{F}'}(\mathbf{H}) = -\frac{1}{2N}\mathrm{Tr}\Bigg\{ &\left[ 2n^2\left(\widetilde{\boldsymbol{\Sigma}}_B(\mathbf{H}) - \frac{4pq}{(p+q)^2}\boldsymbol{\Sigma}_B(\mathbf{H})\right) + \widetilde{\Delta}_3 \right] \\
&\left[ 2n\left(\widetilde{\boldsymbol{\Sigma}}_B(\mathbf{H}) - \frac{4pq}{(p+q)^2}\boldsymbol{\Sigma}_B(\mathbf{H})\right) + \Delta_2 + \lambda_{W_2}N\mathbf{I} \right]^{-1} \Bigg\} + \frac{1}{2} \\
&+ \frac{\lambda_H}{2}\mathrm{Tr}\left\{ N\widetilde{\boldsymbol{\Sigma}}_T(\mathbf{H}) \right\}
\end{aligned}
\tag{99}
$$

### F.4 Trace evolution of covariance matrices

Now, we analyze the traces of $\frac{d\boldsymbol{\Sigma}_W}{dt}, \frac{d\boldsymbol{\Sigma}_B}{dt}$ along the gradient flow:

$$
\frac{d\mathbf{H}_t}{dt} = -\nabla\widehat{\mathcal{R}}^{\mathcal{F}'}(\mathbf{H}_t). \tag{100}
$$

Let $\partial_{kjl}$ represent the derivative of $l^{th}$ entry of $\mathbf{h}_{k,j}$. For notational simplicity, we also consider $\boldsymbol{\Sigma}_W = \boldsymbol{\Sigma}_W(\mathbf{H}), \boldsymbol{\Sigma}_B = \boldsymbol{\Sigma}_B(\mathbf{H}), \widetilde{\boldsymbol{\Sigma}}_B = \widetilde{\boldsymbol{\Sigma}}_B(\mathbf{H}), \widetilde{\boldsymbol{\Sigma}}_T = \widetilde{\boldsymbol{\Sigma}}_T(\mathbf{H})$. This leads to:

$$
\begin{aligned}
\partial_{kjl}\boldsymbol{\Sigma}_B &= \frac{1}{2n}\left( \mathbf{e}_l\left(\overline{\mathbf{h}}_k - \overline{\mathbf{h}}_G\right)^\top + \left(\overline{\mathbf{h}}_k - \overline{\mathbf{h}}_G\right)\mathbf{e}_l^\top \right) \\
\partial_{kjl}\widetilde{\boldsymbol{\Sigma}}_B &= \frac{1}{2n}\left( \mathbf{e}_l\overline{\mathbf{h}}_k^\top + \overline{\mathbf{h}}_k\mathbf{e}_l^\top \right) \\
\partial_{kjl}\boldsymbol{\Sigma}_W &= \frac{1}{2n}\left( \mathbf{e}_l\left(\mathbf{h}_{k,j} - \overline{\mathbf{h}}_k\right)^\top + \left(\mathbf{h}_{k,j} - \overline{\mathbf{h}}_k\right)\mathbf{e}_l^\top \right) \\
\partial_{kjl}\widetilde{\boldsymbol{\Sigma}}_T &= \frac{1}{2n}\left( \mathbf{e}_l\mathbf{h}_{k,j}^\top + \mathbf{h}_{k,j}\mathbf{e}_l^\top \right)
\end{aligned}
\tag{101}
$$

Now, considering $\mathbf{J} = 2n\left(\widetilde{\boldsymbol{\Sigma}}_B - \frac{4pq}{(p+q)^2}\boldsymbol{\Sigma}_B\right)$, we formulate $\partial_{kjl}\widehat{\mathcal{R}}^{\mathcal{F}'}(\mathbf{H})$ as:

$$
\partial_{kjl}\widehat{\mathcal{R}}^{\mathcal{F}'}(\mathbf{H}) = \frac{-n}{2N}\mathrm{Tr}\left\{ \partial_{kjl}\left( \left[\mathbf{J} + \frac{\widetilde{\Delta}_3}{n}\right][\mathbf{J} + \Delta_2 + \lambda_{W_2}N\mathbf{I}]^{-1} \right) \right\} + \frac{N\lambda_H}{4n}\left(2\mathbf{e}_l^\top\mathbf{h}_{k,j}\right)
$$

Since $C = 2$ in our analysis, the derivative expands into:

$$
\begin{aligned}
\partial_{kjl}\widehat{\mathcal{R}}^{\mathcal{F}'}(\mathbf{H}) = &-\frac{1}{4}\mathrm{Tr}\left\{ \partial_{kjl}\left(\mathbf{J} + \frac{\widetilde{\Delta}_3}{n}\right)[\mathbf{J} + \Delta_2 + \lambda_{W_2}N\mathbf{I}]^{-1} \right\} \\
&-\frac{1}{4}\mathrm{Tr}\left\{ \left[\mathbf{J} + \frac{\widetilde{\Delta}_3}{n}\right]\partial_{kjl}\left([\mathbf{J} + \Delta_2 + \lambda_{W_2}N\mathbf{I}]^{-1}\right) \right\} + \lambda_H\mathbf{e}_l^\top\mathbf{h}_{k,j}
\end{aligned}
$$

Where the second term can be expanded as:

$$
\begin{aligned}
&\frac{1}{4}\mathrm{Tr}\left\{ \left[\mathbf{J} + \frac{\widetilde{\Delta}_3}{n}\right]\partial_{kjl}\left([\mathbf{J} + \Delta_2 + \lambda_{W_2}N\mathbf{I}]^{-1}\right) \right\} \\
&= -\frac{1}{4}\mathrm{Tr}\left\{ \left[\mathbf{J} + \frac{\widetilde{\Delta}_3}{n}\right][\mathbf{J} + \Delta_2 + \lambda_{W_2}N\mathbf{I}]^{-1}\partial_{kjl}(\mathbf{J} + \Delta_2)[\mathbf{J} + \Delta_2 + \lambda_{W_2}N\mathbf{I}]^{-1} \right\} \\
&= -\frac{1}{4}\mathrm{Tr}\left\{ [\mathbf{J} + \Delta_2 + \lambda_{W_2}N\mathbf{I}]^{-1}\left[\mathbf{J} + \frac{\widetilde{\Delta}_3}{n}\right][\mathbf{J} + \Delta_2 + \lambda_{W_2}N\mathbf{I}]^{-1}\partial_{kjl}(\mathbf{J} + \Delta_2) \right\}
\end{aligned}
$$

Now, by expanding $\partial_{kjl}(\mathbf{J})$ in terms of covariance matrix derivatives, we get:

$$\partial_{kjl}(\mathbf{J}) = \partial_{kjl}2n\left(\widetilde{\boldsymbol{\Sigma}}_B - \frac{4pq}{(p+q)^2}\boldsymbol{\Sigma}_B\right)$$

$$= \left(\mathbf{e}_l\overline{\mathbf{h}}_k^\top + \overline{\mathbf{h}}_k\mathbf{e}_l^\top\right) - \frac{4pq}{(p+q)^2}\left(\mathbf{e}_l\left(\overline{\mathbf{h}}_k - \overline{\mathbf{h}}_G\right)^\top + \left(\overline{\mathbf{h}}_k - \overline{\mathbf{h}}_G\right)\mathbf{e}_l^\top\right)$$

$$= \mathbf{e}_l\left(\left(\frac{p-q}{p+q}\right)^2\overline{\mathbf{h}}_k + \frac{4pq}{(p+q)^2}\overline{\mathbf{h}}_G\right)^\top + \left(\left(\frac{p-q}{p+q}\right)^2\overline{\mathbf{h}}_k + \frac{4pq}{(p+q)^2}\overline{\mathbf{h}}_G\right)\mathbf{e}_l^\top$$

This leads to the following formulation for $\partial_{kjl}\widehat{\mathcal{R}}^{\mathcal{F}'}(\mathbf{H})$:

$$\partial_{kjl}\widehat{\mathcal{R}}^{\mathcal{F}'}(\mathbf{H}) = -\frac{1}{4}\mathrm{Tr}\left\{[\mathbf{J} + \Delta_2 + \lambda_{W_2}N\mathbf{I}]^{-1}\partial_{kjl}(\mathbf{J})\right\}$$

$$+ \frac{1}{4}\mathrm{Tr}\left\{[\mathbf{J} + \Delta_2 + \lambda_{W_2}N\mathbf{I}]^{-1}\left[\mathbf{J} + \frac{\widetilde{\Delta}_3}{n}\right][\mathbf{J} + \Delta_2 + \lambda_{W_2}N\mathbf{I}]^{-1}\partial_{kjl}(\mathbf{J})\right\}$$

$$+ \lambda_H\mathbf{e}_l^\top\mathbf{h}_{k,j} + \mathcal{P}_{kjl}$$

$$= -\frac{1}{2}\mathrm{Tr}\left\{[\mathbf{J} + \Delta_2 + \lambda_{W_2}N\mathbf{I}]^{-1}\left[\mathbf{e}_l\left(\left(\frac{p-q}{p+q}\right)^2\overline{\mathbf{h}}_k + \frac{4pq}{(p+q)^2}\overline{\mathbf{h}}_G\right)^\top\right]\right\}$$

$$+ \frac{1}{2}\mathrm{Tr}\left\{[\mathbf{J} + \Delta_2 + \lambda_{W_2}N\mathbf{I}]^{-1}\left[\mathbf{J} + \frac{\widetilde{\Delta}_3}{n}\right][\mathbf{J} + \Delta_2 + \lambda_{W_2}N\mathbf{I}]^{-1}\right.$$

$$\left. \cdot \left[\mathbf{e}_l\left(\left(\frac{p-q}{p+q}\right)^2\overline{\mathbf{h}}_k + \frac{4pq}{(p+q)^2}\overline{\mathbf{h}}_G\right)^\top\right]\right\} + \lambda_H\mathbf{e}_l^\top\mathbf{h}_{k,j} + \mathcal{P}_{kjl}$$

Where $\mathcal{P}_{kjl}$ represents the remaining trace terms pertaining to the partial derivatives of $\Delta_2, \widetilde{\Delta}_3$:

$$\mathcal{P}_{kjl} = -\frac{1}{4}\mathrm{Tr}\left\{[\mathbf{J} + \Delta_2 + \lambda_{W_2}N\mathbf{I}]^{-1}\partial_{k,j,l}\left(\frac{\widetilde{\Delta}_3}{n}\right)\right\}$$

$$+ \frac{1}{4}\mathrm{Tr}\left\{[\mathbf{J} + \Delta_2 + \lambda_{W_2}N\mathbf{I}]^{-1}\left[\mathbf{J} + \frac{\widetilde{\Delta}_3}{n}\right][\mathbf{J} + \Delta_2 + \lambda_{W_2}N\mathbf{I}]^{-1}\partial_{kjl}(\Delta_2)\right\}$$

We now denote $\mathbf{M} := [\mathbf{J} + \Delta_2 + \lambda_{W_2}N\mathbf{I}]^{-1}$ to obtain:

$$\partial_{kjl}\widehat{\mathcal{R}}^{\mathcal{F}'}(\mathbf{H}) = -\frac{1}{2}\left(\left[\mathbf{M} - \mathbf{M}\left[\mathbf{J} + \frac{\widetilde{\Delta}_3}{n}\right]\mathbf{M}\right]\left[\left(\frac{p-q}{p+q}\right)^2\overline{\mathbf{h}}_k + \frac{4pq}{(p+q)^2}\overline{\mathbf{h}}_G\right] - 2\lambda_H\mathbf{h}_{k,j}\right)^\top\mathbf{e}_l$$

$$+ \mathcal{P}_{kjl}$$

$$(102)$$

Without loss of generality, since $\mathcal{P}_{kjl} \in \mathbb{R}$, we consider $\mathcal{P}_{kjl} = \mathbf{p}_{k,j}^\top\mathbf{e}_l$ where $\mathbf{p}_{k,j} \in \mathbb{R}^{d_L-1}$ is a random vector which represents the overall perturbation effect of $\mathcal{P}_{kjl}$. Note that the randomness is associated with the $\mathbf{E}$ matrix in $\Delta_2, \widetilde{\Delta}_3$. We can now represent $\partial_{kjl}\widehat{\mathcal{R}}^{\mathcal{F}'}(\mathbf{H}) = \langle\mathbf{R}_{k,j}, \mathbf{e}_l\rangle$, where:

$$\mathbf{R}_{k,j} = -\frac{1}{2}\left(\widetilde{\mathbf{M}}\left[\left(\frac{p-q}{p+q}\right)^2\overline{\mathbf{h}}_k + \frac{4pq}{(p+q)^2}\overline{\mathbf{h}}_G\right] - 2\lambda_H\mathbf{h}_{k,j} - 2\mathbf{p}_{k,j}\right)$$

$$\widetilde{\mathbf{M}} = \left[\mathbf{M} - \mathbf{M}\left[\mathbf{J} + \frac{\widetilde{\Delta}_3}{n}\right]\mathbf{M}\right]$$

$$(103)$$

- **Derivative of $\boldsymbol{\Sigma}_B$:** By denoting $\boldsymbol{\Sigma}_B(a,b) = \mathbf{e}_a^\top \boldsymbol{\Sigma}_B \mathbf{e}_b$ as the $(a,b)$-element of $\boldsymbol{\Sigma}_B$, we use the above result for $\partial_{kjl}\widehat{\mathcal{R}}^{\mathcal{F}'}(\mathbf{H})$ and the chain rule to compute $\frac{d\boldsymbol{\Sigma}_B}{dt}$ along the flow (14), as follows:

$$
\begin{aligned}
\frac{d\boldsymbol{\Sigma}_B(a,b)}{dt} &= \sum_{k,j,l} \partial_{k,j,l}\boldsymbol{\Sigma}_B(a,b)\frac{d\mathbf{h}_{k,j}[l]}{dt} = \sum_{k,j,l}\partial_{k,j,l}\boldsymbol{\Sigma}_B(a,b)\left(-\partial_{k,j,l}\widehat{\mathcal{R}}^{\mathcal{F}'}(\mathbf{H})\right)\\
&= \sum_{k,j}\sum_l -\frac{1}{2n}\left(\langle\mathbf{e}_a,\mathbf{e}_l\rangle\langle\mathbf{e}_b,\overline{\mathbf{h}}_k-\overline{\mathbf{h}}_G\rangle + \langle\mathbf{e}_a,\overline{\mathbf{h}}_k-\overline{\mathbf{h}}_G\rangle\langle\mathbf{e}_l,\mathbf{e}_b\rangle\right)\langle\mathbf{R}_{k,j},\mathbf{e}_l\rangle\\
&= \sum_{k,j}-\frac{1}{2n}\left(\langle\mathbf{e}_a,\mathbf{R}_{k,j}\rangle\langle\mathbf{e}_b,\overline{\mathbf{h}}_k-\overline{\mathbf{h}}_G\rangle + \langle\mathbf{e}_a,\overline{\mathbf{h}}_k-\overline{\mathbf{h}}_G\rangle\langle\mathbf{R}_{k,j},\mathbf{e}_b\rangle\right)\\
&= \frac{1}{2n}\mathbf{e}_a^\top\left(\sum_{k,j}-\mathbf{R}_{k,j}\left(\overline{\mathbf{h}}_k-\overline{\mathbf{h}}_G\right)^\top - \left(\overline{\mathbf{h}}_k-\overline{\mathbf{h}}_G\right)\mathbf{R}_{k,j}^\top\right)\mathbf{e}_b\\
&= \frac{1}{4n}\mathbf{e}_a^\top\left(\sum_{k,j}\left(\widetilde{\mathbf{M}}\left[\left(\frac{p-q}{p+q}\right)^2\overline{\mathbf{h}}_k + \frac{4pq}{(p+q)^2}\overline{\mathbf{h}}_G\right] - 2\lambda_H\mathbf{h}_{k,j} - 2\mathbf{p}_{k,j}\right)\left(\overline{\mathbf{h}}_k-\overline{\mathbf{h}}_G\right)^\top\right)\mathbf{e}_b\\
&\quad + \frac{1}{4n}\mathbf{e}_a^\top\left(\sum_{k,j}\left(\overline{\mathbf{h}}_k-\overline{\mathbf{h}}_G\right)\left(\widetilde{\mathbf{M}}\left[\left(\frac{p-q}{p+q}\right)^2\overline{\mathbf{h}}_k + \frac{4pq}{(p+q)^2}\overline{\mathbf{h}}_G\right] - 2\lambda_H\mathbf{h}_{k,j} - 2\mathbf{p}_{k,j}\right)^\top\right)\mathbf{e}_b
\end{aligned}
$$

For further simplification, let's consider the following term:

$$
\begin{aligned}
\sum_{k,j}&\left[\left(\frac{p-q}{p+q}\right)^2\overline{\mathbf{h}}_k + \frac{4pq}{(p+q)^2}\overline{\mathbf{h}}_G\right]\left[\overline{\mathbf{h}}_k-\overline{\mathbf{h}}_G\right]^\top\\
&= \frac{1}{(p+q)^2}\sum_{k,j}\left[(p-q)^2\,\overline{\mathbf{h}}_k + 4pq\overline{\mathbf{h}}_G\right]\left[\overline{\mathbf{h}}_k-\overline{\mathbf{h}}_G\right]^\top\\
&= \frac{1}{(p+q)^2}\sum_{k,j}\left[(p-q)^2\overline{\mathbf{h}}_k\overline{\mathbf{h}}_k^\top - (p-q)^2\overline{\mathbf{h}}_k\overline{\mathbf{h}}_G^\top + 4pq\overline{\mathbf{h}}_G\overline{\mathbf{h}}_k^\top - 4pq\overline{\mathbf{h}}_G\overline{\mathbf{h}}_G^\top\right]\\
&= \frac{1}{(p+q)^2}\left[(p-q)^2 2n\widetilde{\boldsymbol{\Sigma}}_B - 8npq\boldsymbol{\Sigma}_G - 2n\boldsymbol{\Sigma}_G\left((p-q)^2 - 4pq\right)\right]\\
&= \frac{1}{(p+q)^2}\left[2n(p-q)^2\boldsymbol{\Sigma}_B + 2n(p-q)^2\boldsymbol{\Sigma}_G - 8npq\boldsymbol{\Sigma}_G - 2n\boldsymbol{\Sigma}_G\left((p-q)^2 - 4pq\right)\right]\\
&= 2n\left(\frac{p-q}{p+q}\right)^2\boldsymbol{\Sigma}_B
\end{aligned}
$$

Next, we proceed with the simplification of $\sum_{k,j}-\lambda_H\mathbf{h}_{k,j}(\overline{\mathbf{h}}_k-\overline{\mathbf{h}}_G)^\top - \lambda_H(\overline{\mathbf{h}}_k-\overline{\mathbf{h}}_G)\mathbf{h}_{k,j}^\top$:

$$
\begin{aligned}
-\lambda_H&\left(\sum_{k,j}\mathbf{h}_{k,j}(\overline{\mathbf{h}}_k-\overline{\mathbf{h}}_G)^\top + (\overline{\mathbf{h}}_k-\overline{\mathbf{h}}_G)\mathbf{h}_{k,j}^\top\right)\\
&= -\lambda_H\left(n\overline{\mathbf{h}}_1(\overline{\mathbf{h}}_1-\overline{\mathbf{h}}_G)^\top + n\overline{\mathbf{h}}_2(\overline{\mathbf{h}}_2-\overline{\mathbf{h}}_G)^\top + n(\overline{\mathbf{h}}_1-\overline{\mathbf{h}}_G)\overline{\mathbf{h}}_1^\top + n(\overline{\mathbf{h}}_2-\overline{\mathbf{h}}_G)\overline{\mathbf{h}}_2^\top\right)\\
&= -\lambda_H\left(2n\overline{\mathbf{h}}_1\overline{\mathbf{h}}_1^\top + 2n\overline{\mathbf{h}}_2\overline{\mathbf{h}}_2^\top - n\overline{\mathbf{h}}_1\overline{\mathbf{h}}_G^\top - n\overline{\mathbf{h}}_2\overline{\mathbf{h}}_G^\top - n\overline{\mathbf{h}}_G\overline{\mathbf{h}}_1^\top - n\overline{\mathbf{h}}_G\overline{\mathbf{h}}_2^\top\right)\\
&= -\lambda_H\left(2n\overline{\mathbf{h}}_1\overline{\mathbf{h}}_1^\top + 2n\overline{\mathbf{h}}_2\overline{\mathbf{h}}_2^\top - 4n\overline{\mathbf{h}}_G\overline{\mathbf{h}}_G^\top\right) = -\lambda_H\left(4n\widetilde{\boldsymbol{\Sigma}}_B - 4n\boldsymbol{\Sigma}_G\right) = -4n\lambda_H\boldsymbol{\Sigma}_B
\end{aligned}
$$

These results now simplify $\frac{d\boldsymbol{\Sigma}_B}{dt}$ as:

$$
\frac{d\boldsymbol{\Sigma}_B}{dt} = \frac{1}{4n}\mathbf{e}_a^\top\left(2n\left(\frac{p-q}{p+q}\right)^2\widetilde{\mathbf{M}}\boldsymbol{\Sigma}_B + 2n\left(\frac{p-q}{p+q}\right)^2\boldsymbol{\Sigma}_B\widetilde{\mathbf{M}} - 8n\lambda_H\boldsymbol{\Sigma}_B - \widetilde{\mathbf{P}}_B\right)\mathbf{e}_b \tag{104}
$$

Where $\widetilde{\mathbf{P}}_B = 2\sum_{k,j} \mathbf{p}_{k,j}(\overline{\mathbf{h}}_k - \overline{\mathbf{h}}_G)^\top + (\overline{\mathbf{h}}_k - \overline{\mathbf{h}}_G)\mathbf{p}_{k,j}^\top$.

• **Derivative of $\mathbf{\Sigma}_W$:** By denoting $\mathbf{\Sigma}_W(a,b) = \mathbf{e}_a^\top \mathbf{\Sigma}_W \mathbf{e}_b$ as the $(a,b)$-element of $\mathbf{\Sigma}_W$, we use a similar line of analysis as above to compute $\frac{d\mathbf{\Sigma}_W}{dt}$ as follows:

$$
\frac{d\mathbf{\Sigma}_W(a,b)}{dt} = \sum_{k,j,l} \partial_{k,j,l}\mathbf{\Sigma}_W(a,b)\frac{d\mathbf{h}_{k,j}[l]}{dt} = \sum_{k,j,l} \partial_{k,j,l}\mathbf{\Sigma}_W(a,b)\left(-\partial_{k,j,l}\widehat{\mathcal{R}}^{\mathcal{F}'}(\mathbf{H})\right)
$$

$$
= \sum_{k,j}\sum_{l} -\frac{1}{2n}\left(\langle \mathbf{e}_a, \mathbf{e}_l\rangle\langle \mathbf{e}_b, \mathbf{h}_{k,j} - \overline{\mathbf{h}}_k\rangle + \langle \mathbf{e}_a, \mathbf{h}_{k,j} - \overline{\mathbf{h}}_k\rangle\langle \mathbf{e}_l, \mathbf{e}_b\rangle\right)\langle \mathbf{R}_{k,j}, \mathbf{e}_l\rangle
$$

$$
= \sum_{k,j} -\frac{1}{2n}\left(\langle \mathbf{e}_a, \mathbf{R}_{k,j}\rangle\langle \mathbf{e}_b, \mathbf{h}_{k,j} - \overline{\mathbf{h}}_k\rangle + \langle \mathbf{e}_a, \mathbf{h}_{k,j} - \overline{\mathbf{h}}_k\rangle\langle \mathbf{R}_{k,j}, \mathbf{e}_b\rangle\right)
$$

$$
= \frac{1}{2n}\mathbf{e}_a^\top\left(\sum_{k,j} -\mathbf{R}_{k,j}\left(\mathbf{h}_{k,j} - \overline{\mathbf{h}}_k\right)^\top - \left(\mathbf{h}_{k,j} - \overline{\mathbf{h}}_k\right)\mathbf{R}_{k,j}^\top\right)\mathbf{e}_b
$$

$$
= \frac{1}{4n}\mathbf{e}_a^\top\left(\sum_{k,j}\left(\widetilde{\mathbf{M}}\left[\left(\frac{p-q}{p+q}\right)^2\overline{\mathbf{h}}_k + \frac{4pq}{(p+q)^2}\overline{\mathbf{h}}_G\right] - 2\lambda_H\mathbf{h}_{k,j} - 2\mathbf{p}_{k,j}\right)\left(\mathbf{h}_{k,j} - \overline{\mathbf{h}}_k\right)^\top\right)\mathbf{e}_b
$$

$$
+ \frac{1}{4n}\mathbf{e}_a^\top\left(\sum_{k,j}\left(\mathbf{h}_{k,j} - \overline{\mathbf{h}}_k\right)\left(\widetilde{\mathbf{M}}\left[\left(\frac{p-q}{p+q}\right)^2\overline{\mathbf{h}}_k + \frac{4pq}{(p+q)^2}\overline{\mathbf{h}}_G\right] - 2\lambda_H\mathbf{h}_{k,j} - 2\mathbf{p}_{k,j}\right)^\top\right)\mathbf{e}_b
$$

For further simplification, let's consider the following term:

$$
\sum_{k,j}\left[\left(\frac{p-q}{p+q}\right)^2\overline{\mathbf{h}}_k + \frac{4pq}{(p+q)^2}\overline{\mathbf{h}}_G\right]\left[\mathbf{h}_{k,j} - \overline{\mathbf{h}}_k\right]^\top
$$

$$
= \sum_{k}\left[\left(\frac{p-q}{p+q}\right)^2\overline{\mathbf{h}}_k + \frac{4pq}{(p+q)^2}\overline{\mathbf{h}}_G\right]\left[\sum_{j}\left(\mathbf{h}_{k,j} - \overline{\mathbf{h}}_k\right)\right]^\top = \mathbf{0}
$$

Next, we simplify $\sum_{k,j} -\lambda_H\mathbf{h}_{k,j}(\mathbf{h}_{k,j} - \overline{\mathbf{h}}_k)^\top - \lambda_H(\mathbf{h}_{k,j} - \overline{\mathbf{h}}_k)\mathbf{h}_{k,j}^\top$ as follows:

$$
-\lambda_H\sum_{k,j}\left(\mathbf{h}_{k,j}(\mathbf{h}_{k,j} - \overline{\mathbf{h}}_k)^\top + (\mathbf{h}_{k,j} - \overline{\mathbf{h}}_k)\mathbf{h}_{k,j}^\top\right)
$$

$$
= -\lambda_H\sum_{k,j}\left(\mathbf{h}_{k,j}\mathbf{h}_{k,j}^\top - \mathbf{h}_{k,j}\overline{\mathbf{h}}_k^\top + \mathbf{h}_{k,j}\mathbf{h}_{k,j}^\top - \overline{\mathbf{h}}_k\mathbf{h}_{k,j}^\top\right)
$$

$$
= -\lambda_H\sum_{k,j}\left(\mathbf{h}_{k,j}\mathbf{h}_{k,j}^\top - \mathbf{h}_{k,j}\overline{\mathbf{h}}_k^\top + \mathbf{h}_{k,j}\mathbf{h}_{k,j}^\top - \overline{\mathbf{h}}_k\mathbf{h}_{k,j}^\top + \overline{\mathbf{h}}_k\overline{\mathbf{h}}_k^\top - \overline{\mathbf{h}}_k\overline{\mathbf{h}}_k^\top\right)
$$

$$
= -\lambda_H\left(2n\mathbf{\Sigma}_W + 2n\widetilde{\mathbf{\Sigma}}_T - 2n\widetilde{\mathbf{\Sigma}}_B\right) = -4n\lambda_H\mathbf{\Sigma}_W
$$

Where the last inequality is based on the fact that $\widetilde{\mathbf{\Sigma}}_T = \mathbf{\Sigma}_W + \widetilde{\mathbf{\Sigma}}_B$. These results simplify $\frac{d\mathbf{\Sigma}_W}{dt}$ as:

$$
\frac{d\mathbf{\Sigma}_W}{dt} = \frac{1}{4n}\mathbf{e}_a^\top\left(-8n\lambda_H\mathbf{\Sigma}_W - \widetilde{\mathbf{P}}_W\right)\mathbf{e}_b \tag{105}
$$

Where $\widetilde{\mathbf{P}}_W = 2\sum_{k,j}\mathbf{p}_{k,j}(\mathbf{h}_{k,j} - \overline{\mathbf{h}}_k)^\top + (\mathbf{h}_{k,j} - \overline{\mathbf{h}}_k)\mathbf{p}_{k,j}^\top$.

• **Trace of covariance matrices along the flow:** Taking the derivative of $\text{Tr}(\mathbf{\Sigma}_W)$ gives us:

$$
\frac{d\text{Tr}(\mathbf{\Sigma}_W)}{dt} = -2\lambda_H\text{Tr}(\mathbf{\Sigma}_W) - \frac{1}{4n}\text{Tr}\left(\widetilde{\mathbf{P}}_W\right)
$$

$$
= -2\lambda_H\text{Tr}(\mathbf{\Sigma}_W) - \frac{1}{n}\text{Tr}\left(\sum_{k,j}\mathbf{p}_{k,j}(\mathbf{h}_{k,j} - \overline{\mathbf{h}}_k)^\top\right) \tag{106}
$$

Similarly, the derivative of $\text{Tr}\left(\boldsymbol{\Sigma}_B\right)$ gives us:

$$
\begin{aligned}
\frac{d\text{Tr}\left(\boldsymbol{\Sigma}_B\right)}{dt} &= -2\lambda_H \text{Tr}(\boldsymbol{\Sigma}_B) + \frac{1}{4n}\text{Tr}\left(2n\left(\frac{p-q}{p+q}\right)^2\widetilde{\mathbf{M}}\boldsymbol{\Sigma}_B + 2n\left(\frac{p-q}{p+q}\right)^2\boldsymbol{\Sigma}_B\widetilde{\mathbf{M}} - \widetilde{\mathbf{P}}_B\right) \\
&= -2\lambda_H \text{Tr}(\boldsymbol{\Sigma}_B) + \frac{1}{2n}\text{Tr}\left(2n\left(\frac{p-q}{p+q}\right)^2\widetilde{\mathbf{M}}\boldsymbol{\Sigma}_B - 2\sum_{k,j}\mathbf{p}_{k,j}(\overline{\mathbf{h}}_k - \overline{\mathbf{h}}_G)^\top\right) \\
&= -2\lambda_H \text{Tr}(\boldsymbol{\Sigma}_B) + \text{Tr}\left(\left(\frac{p-q}{p+q}\right)^2\widetilde{\mathbf{M}}\boldsymbol{\Sigma}_B\right) - \frac{1}{n}\text{Tr}\left(\sum_{k,j}\mathbf{p}_{k,j}(\overline{\mathbf{h}}_k - \overline{\mathbf{h}}_G)^\top\right) \\
&= \text{Tr}\left(\left[\left(\frac{p-q}{p+q}\right)^2\widetilde{\mathbf{M}} - 2\lambda_H\mathbf{I}\right]\boldsymbol{\Sigma}_B\right) - \frac{1}{n}\text{Tr}\left(\sum_{k,j}\mathbf{p}_{k,j}(\overline{\mathbf{h}}_k - \overline{\mathbf{h}}_G)^\top\right)
\end{aligned}
$$
(107)

Observe that for small enough perturbation matrix $\mathbf{E}$, formally for $\|\mathbf{E}\| < E$ for sufficiently small $E$, the sign of $\frac{d\text{Tr}(\boldsymbol{\Sigma}_W)}{dt}$ and $\frac{d\text{Tr}(\boldsymbol{\Sigma}_B)}{dt}$ depends on the first term in each of them, and not on the second term that depends on $\mathbf{E}$. Therefore, since $\lambda_H > 0$, we have that $\text{Tr}\left(\boldsymbol{\Sigma}_W\right)$ decreases. It is left to show that $\text{Tr}\left(\boldsymbol{\Sigma}_B\right)$ increases in this regime.

Since the trace of the product of a positive definite matrix and a non-zero positive semidefinite matrix is positive by Von-Neumann trace inequality, we aim for conditions that allow $\left[\left(\frac{p-q}{p+q}\right)^2\widetilde{\mathbf{M}} - 2\lambda_H\mathbf{I}\right]$ to be positive definite. First, observe that $\widetilde{\mathbf{M}}$ is symmetric:

$$
\widetilde{\mathbf{M}} = \left[\mathbf{M} - \mathbf{M}\left[\mathbf{J} + \frac{\widetilde{\Delta}_3}{n}\right]\mathbf{M}\right]
$$
$$
\mathbf{M} = [\mathbf{J} + \Delta_2 + \lambda_{W_2}N\mathbf{I}]^{-1}
$$
$$
\mathbf{J} = 2n\left(\widetilde{\boldsymbol{\Sigma}}_B - \frac{4pq}{(p+q)^2}\boldsymbol{\Sigma}_B\right)
$$

Since $\widetilde{\boldsymbol{\Sigma}}_B, \boldsymbol{\Sigma}_B, \mathbf{I}, \Delta_2$ and $\widetilde{\Delta}_3$ are symmetric.

Observe that $\frac{4pq}{(p+q)^2} \leq 1$, because

$$
0 \leq (p-q)^2 = p^2 + 2pq + q^2 - 4pq = (p+q)^2 - 4pq.
$$

Thus,

$$
\mathbf{J} = 2n\left(\widetilde{\boldsymbol{\Sigma}}_B - \frac{4pq}{(p+q)^2}\boldsymbol{\Sigma}_B\right) \geq 2n\left(\widetilde{\boldsymbol{\Sigma}}_B - \boldsymbol{\Sigma}_B\right) = 2n\boldsymbol{\Sigma}_G \geq 0
$$

Thus also $\mathbf{M} > 0$ for small $\Delta_2$. Note also that $[\mathbf{J} + \lambda_{W_2}N\mathbf{I}]^{-1}\mathbf{J} < \mathbf{I}$. Therefore, for small enough $\mathbf{E}$ (and thus small $\Delta_2, \widetilde{\Delta}_3$), we have that

$$
\widetilde{\mathbf{M}} = \mathbf{M} - \mathbf{M}\left[\mathbf{J} + \frac{\widetilde{\Delta}_3}{n}\right]\mathbf{M} > 0.
$$

Thus, $\left[\left(\frac{p-q}{p+q}\right)^2\widetilde{\mathbf{M}} - 2\lambda_H\mathbf{I}\right]$ is positive definite when:

$$
2\lambda_H < \left(\frac{p-q}{p+q}\right)^2\lambda_{min}\left(\widetilde{\mathbf{M}}\right)
$$
(108)

Here $\lambda_{min}\left(\widetilde{\mathbf{M}}\right)$ represents the smallest eigenvalue of $\widetilde{\mathbf{M}}$.

# G   Proof of Theorem 4.1

Let's begin by calculating the expected value and covariance of features $\mathbf{X}^{(l)}$, which are obtained after the graph convolution operation based on equation 3.

• **Case** $c = 1$:

We begin by considering the features of a node belonging to class $c = 1$ as follows:

$$\mathbf{x}_{1,i}^{(l)} = \mathbf{W}_1^{*(l)}\mathbf{h}_{1,i}^{(l-1)} + \mathbf{W}_2^{*(l)}\mathbf{H}^{(l-1)}\widehat{\mathbf{A}}_t\mathbf{e}_{1,i} \tag{109}$$

By expanding the $\mathbf{H}^{(l-1)}\widehat{\mathbf{A}}_t$ term based on neighbors from all classes, we get:

$$\mathbf{x}_{1,i}^{(l)} = \mathbf{W}_1^{*(l)}\mathbf{h}_{1,i}^{(l-1)} + \mathbf{W}_2^{*(l)}\left(\frac{\sum_{v_{1,j}\in\mathcal{N}_1(v_{1,i})}\mathbf{h}_{1,j} + \sum_{v_{2,j}\in\mathcal{N}_2(v_{1,i})}\mathbf{h}_{2,j}}{|\mathcal{N}(v_{1,i})|}\right) \tag{110}$$

Now, by taking expectations on both sides with respect to features $\mathbf{h}_{1,i}^{(l-1)}$ and structure $\widehat{\mathbf{A}}_t$, we get:

$$\mathbb{E}_{\widehat{\mathbf{A}},\mathbf{h}}\mathbf{x}_{1,i}^{(l)} = \mathbf{W}_1^{*(l)}\mathbb{E}_{\widehat{\mathbf{A}},\mathbf{h}}\mathbf{h}_{1,i}^{(l-1)} + \mathbf{W}_2^{*(l)}\mathbb{E}_{\widehat{\mathbf{A}},\mathbf{h}}\left(\frac{\sum_{v_{1,j}\in\mathcal{N}_1(v_{1,i})}\mathbf{h}_{1,j} + \sum_{v_{2,j}\in\mathcal{N}_2(v_{1,i})}\mathbf{h}_{2,j}}{|\mathcal{N}(v_{1,i})|}\right)$$

$$= \mathbf{W}_1^{*(l)}\boldsymbol{\mu}_1^{(l-1)} + \mathbf{W}_2^{*(l)}\left(\frac{np\boldsymbol{\mu}_1^{(l-1)} + nq\boldsymbol{\mu}_2^{(l-1)}}{n(p+q)}\right)$$

$$= \left(\mathbf{W}_1^{*(l)} + \frac{p}{p+q}\mathbf{W}_2^{*(l)}\right)\boldsymbol{\mu}_1^{(l-1)} + \left(\frac{q}{p+q}\mathbf{W}_2^{*(l)}\right)\boldsymbol{\mu}_2^{(l-1)} \tag{111}$$

Similarly, the covariance $\mathbb{E}_{\widehat{\mathbf{A}},\mathbf{h}}\left(\left[\mathbf{x}_{1,i}^{(l)} - \mathbb{E}_{\widehat{\mathbf{A}},\mathbf{h}}\mathbf{x}_{1,i}^{(l)}\right]\left[\mathbf{x}_{1,i}^{(l)} - \mathbb{E}_{\widehat{\mathbf{A}},\mathbf{h}}\mathbf{x}_{1,i}^{(l)}\right]^{\top}\right)$ is based on:

$$\left[\mathbf{W}_1^{*(l)}\mathbf{h}_{1,i}^{(l-1)} + \mathbf{W}_2^{*(l)}\mathbf{H}^{(l-1)}\widehat{\mathbf{A}}_t\mathbf{e}_{1,i} - \mathbb{E}_{\widehat{\mathbf{A}},\mathbf{h}}\mathbf{x}_{1,i}^{(l)}\right]$$

$$\cdot \left[\mathbf{W}_1^{*(l)}\mathbf{h}_{1,i}^{(l-1)} + \mathbf{W}_2^{*(l)}\mathbf{H}^{(l-1)}\widehat{\mathbf{A}}_t\mathbf{e}_{1,i} - \mathbb{E}_{\widehat{\mathbf{A}},\mathbf{h}}\mathbf{x}_{1,i}^{(l)}\right]^{\top}$$

$$= \left[\mathbf{W}_1^{*(l)}\left(\mathbf{h}_{1,i}^{(l-1)} - \boldsymbol{\mu}_1^{(l-1)}\right) + \mathbf{W}_2^{*(l)}\left(\mathbf{H}^{(l-1)}\widehat{\mathbf{A}}_t\mathbf{e}_{1,i} - \frac{p\boldsymbol{\mu}_1^{(l-1)} + q\boldsymbol{\mu}_2^{(l-1)}}{p+q}\right)\right]$$

$$\cdot \left[\mathbf{W}_1^{*(l)}\left(\mathbf{h}_{1,i}^{(l-1)} - \boldsymbol{\mu}_1^{(l-1)}\right) + \mathbf{W}_2^{*(l)}\left(\mathbf{H}^{(l-1)}\widehat{\mathbf{A}}_t\mathbf{e}_{1,i} - \frac{p\boldsymbol{\mu}_1^{(l-1)} + q\boldsymbol{\mu}_2^{(l-1)}}{p+q}\right)\right]^{\top}$$

$$= \mathbf{W}_1^{*(l)}\left(\mathbf{h}_{1,i}^{(l-1)} - \boldsymbol{\mu}_1^{(l-1)}\right)\left(\mathbf{h}_{1,i}^{(l-1)} - \boldsymbol{\mu}_1^{(l-1)}\right)^{\top}\mathbf{W}_1^{*(l)\top}$$

$$+ \mathbf{W}_1^{*(l)}\left(\mathbf{h}_{1,i}^{(l-1)} - \boldsymbol{\mu}_1^{(l-1)}\right)\left(\mathbf{H}^{(l-1)}\widehat{\mathbf{A}}_t\mathbf{e}_{1,i} - \frac{p\boldsymbol{\mu}_1^{(l-1)} + q\boldsymbol{\mu}_2^{(l-1)}}{p+q}\right)^{\top}\mathbf{W}_2^{*(l)\top}$$

$$+ \mathbf{W}_2^{*(l)}\left(\mathbf{H}^{(l-1)}\widehat{\mathbf{A}}_t\mathbf{e}_{1,i} - \frac{p\boldsymbol{\mu}_1^{(l-1)} + q\boldsymbol{\mu}_2^{(l-1)}}{p+q}\right)\left(\mathbf{h}_{1,i}^{(l-1)} - \boldsymbol{\mu}_1^{(l-1)}\right)^{\top}\mathbf{W}_1^{*(l)\top}$$

$$+ \mathbf{W}_2^{*(l)}\left(\mathbf{H}^{(l-1)}\widehat{\mathbf{A}}_t\mathbf{e}_{1,i} - \frac{p\boldsymbol{\mu}_1^{(l-1)} + q\boldsymbol{\mu}_2^{(l-1)}}{p+q}\right)\left(\mathbf{H}^{(l-1)}\widehat{\mathbf{A}}_t\mathbf{e}_{1,i} - \frac{p\boldsymbol{\mu}_1^{(l-1)} + q\boldsymbol{\mu}_2^{(l-1)}}{p+q}\right)^{\top}\mathbf{W}_2^{*(l)\top} \tag{112}$$

The expectation of the term $\mathbf{W}_1^{*(l)}\left(\mathbf{h}_{1,i}^{(l-1)} - \boldsymbol{\mu}_1^{(l-1)}\right)\left(\mathbf{h}_{1,i}^{(l-1)} - \boldsymbol{\mu}_1^{(l-1)}\right)^{\top}\mathbf{W}_1^{*(l)\top}$ is given by:

$$\mathbb{E}_{\widehat{\mathbf{A}},\mathbf{h}}\left[\mathbf{W}_1^{*(l)}\left(\mathbf{h}_{1,i}^{(l-1)} - \boldsymbol{\mu}_1^{(l-1)}\right)\left(\mathbf{h}_{1,i}^{(l-1)} - \boldsymbol{\mu}_1^{(l-1)}\right)^{\top}\mathbf{W}_1^{*(l)\top}\right]$$

$$= \mathbf{W}_1^{*(l)}\mathbb{E}_{\widehat{\mathbf{A}},\mathbf{h}}\left[\left(\mathbf{h}_{1,i}^{(l-1)} - \boldsymbol{\mu}_1^{(l-1)}\right)\left(\mathbf{h}_{1,i}^{(l-1)} - \boldsymbol{\mu}_1^{(l-1)}\right)^{\top}\right]\mathbf{W}_1^{*(l)\top} \tag{113}$$

$$= \mathbf{W}_1^{*(l)}\boldsymbol{\Sigma}_1^{(l-1)}\mathbf{W}_1^{*(l)\top}$$

The expectation of $\mathbf{W}_1^{*(l)}\left(\mathbf{h}_{1,i}^{(l-1)} - \boldsymbol{\mu}_1^{(l-1)}\right)\left(\mathbf{H}^{(l-1)}\widehat{\mathbf{A}}_t\mathbf{e}_{1,i} - \frac{p\boldsymbol{\mu}_1^{(l-1)}+q\boldsymbol{\mu}_2^{(l-1)}}{p+q}\right)^\top \mathbf{W}_2^{*(l)\top}$ is given by the following:

$$
\mathbb{E}_{\widehat{\mathbf{A}},\mathbf{h}}\left[\mathbf{W}_1^{*(l)}\left(\mathbf{h}_{1,i}^{(l-1)} - \boldsymbol{\mu}_1^{(l-1)}\right)\left(\mathbf{H}^{(l-1)}\widehat{\mathbf{A}}_t\mathbf{e}_{1,i} - \frac{p\boldsymbol{\mu}_1^{(l-1)}+q\boldsymbol{\mu}_2^{(l-1)}}{p+q}\right)^\top \mathbf{W}_2^{*(l)\top}\right]
$$

$$
= \mathbb{E}_{\widehat{\mathbf{A}},\mathbf{h}}\left[\mathbf{W}_1^{*(l)}\mathbf{h}_{1,i}^{(l-1)}\left(\mathbf{H}^{(l-1)}\widehat{\mathbf{A}}_t\mathbf{e}_{1,i} - \frac{p\boldsymbol{\mu}_1^{(l-1)}+q\boldsymbol{\mu}_2^{(l-1)}}{p+q}\right)^\top \mathbf{W}_2^{*(l)\top}\right]
$$

$$
- \mathbb{E}_{\widehat{\mathbf{A}},\mathbf{h}}\left[\mathbf{W}_1^{*(l)}\boldsymbol{\mu}_1^{(l-1)}\left(\mathbf{H}^{(l-1)}\widehat{\mathbf{A}}_t\mathbf{e}_{1,i} - \frac{p\boldsymbol{\mu}_1^{(l-1)}+q\boldsymbol{\mu}_2^{(l-1)}}{p+q}\right)^\top \mathbf{W}_2^{*(l)\top}\right]
$$

$$
= \mathbf{W}_1^{*(l)}\mathbb{E}_{\mathbf{h}}\left[\mathbf{h}_{1,i}^{(l-1)}\left(\frac{p\sum_{j=1}^n \mathbf{h}_{1,j}^{(l-1)}+q\sum_{j=1}^n \mathbf{h}_{2,j}^{(l-1)}}{n(p+q)} - \frac{p\boldsymbol{\mu}_1^{(l-1)}+q\boldsymbol{\mu}_2^{(l-1)}}{p+q}\right)^\top\right]\mathbf{W}_2^{*(l)\top}
$$

$$
= \mathbf{W}_1^{*(l)}\mathbb{E}_{\mathbf{h}}\left[\mathbf{h}_{1,i}^{(l-1)}\left(\frac{p}{n(p+q)}\mathbf{h}_{1,i}^{(l-1)\top}\right)\right]\mathbf{W}_2^{*(l)\top}
$$

$$
+ \mathbf{W}_1^{*(l)}\mathbb{E}_{\mathbf{h}}\left[\mathbf{h}_{1,i}^{(l-1)}\left(\frac{p\sum_{j=1,\neq i}^n \mathbf{h}_{1,j}^{(l-1)}+q\sum_{j=1}^n \mathbf{h}_{2,j}^{(l-1)}}{n(p+q)} - \frac{p\boldsymbol{\mu}_1^{(l-1)}+q\boldsymbol{\mu}_2^{(l-1)}}{p+q}\right)^\top\right]\mathbf{W}_2^{*(l)\top}
$$

Since the features are independent draws from their normal distributions, we can simplify the expectation as follows:

$$
\mathbb{E}_{\widehat{\mathbf{A}},\mathbf{h}}\left[\mathbf{W}_1^{*(l)}\left(\mathbf{h}_{1,i}^{(l-1)} - \boldsymbol{\mu}_1^{(l-1)}\right)\left(\mathbf{H}^{(l-1)}\widehat{\mathbf{A}}_t\mathbf{e}_{1,i} - \frac{p\boldsymbol{\mu}_1^{(l-1)}+q\boldsymbol{\mu}_2^{(l-1)}}{p+q}\right)^\top \mathbf{W}_2^{*(l)\top}\right]
$$

$$
= \mathbf{W}_1^{*(l)}\left[\frac{p}{n(p+q)}\left(\boldsymbol{\Sigma}_1^{(l-1)} + \boldsymbol{\mu}_1^{(l-1)}\boldsymbol{\mu}_1^{(l-1)\top}\right)\right]\mathbf{W}_2^{*(l)\top}
$$

$$
+ \mathbf{W}_1^{*(l)}\left[\boldsymbol{\mu}_1^{(l-1)}\left(\frac{(n-1)p\boldsymbol{\mu}_1^{(l-1)}+nq\boldsymbol{\mu}_2^{(l-1)}}{n(p+q)} - \frac{p\boldsymbol{\mu}_1^{(l-1)}+q\boldsymbol{\mu}_2^{(l-1)}}{p+q}\right)^\top\right]\mathbf{W}_2^{*(l)\top} \tag{114}
$$

$$
= \mathbf{W}_1^{*(l)}\left[\frac{p}{n(p+q)}\boldsymbol{\Sigma}_1^{(l-1)}\right]\mathbf{W}_2^{*(l)\top}
$$

Similarly, the expectation of $\mathbf{W}_2^{*(l)}\left(\mathbf{H}^{(l-1)}\widehat{\mathbf{A}}_t\mathbf{e}_{1,i} - \frac{p\boldsymbol{\mu}_1^{(l-1)}+q\boldsymbol{\mu}_2^{(l-1)}}{p+q}\right)\left(\mathbf{h}_{1,i}^{(l-1)} - \boldsymbol{\mu}_1^{(l-1)}\right)^\top \mathbf{W}_1^{*(l)\top}$ is given by the following:

$$
\mathbb{E}_{\widehat{\mathbf{A}},\mathbf{h}}\left[\mathbf{W}_2^{*(l)}\left(\mathbf{H}^{(l-1)}\widehat{\mathbf{A}}_t\mathbf{e}_{1,i} - \frac{p\boldsymbol{\mu}_1^{(l-1)}+q\boldsymbol{\mu}_2^{(l-1)}}{p+q}\right)\left(\mathbf{h}_{1,i}^{(l-1)} - \boldsymbol{\mu}_1^{(l-1)}\right)^\top \mathbf{W}_1^{*(l)\top}\right]
$$

$$
= \mathbb{E}_{\widehat{\mathbf{A}},\mathbf{h}}\left[\mathbf{W}_1^{*(l)}\left(\mathbf{h}_{1,i}^{(l-1)} - \boldsymbol{\mu}_1^{(l-1)}\right)\left(\mathbf{H}^{(l-1)}\widehat{\mathbf{A}}_t\mathbf{e}_{1,i} - \frac{p\boldsymbol{\mu}_1^{(l-1)}+q\boldsymbol{\mu}_2^{(l-1)}}{p+q}\right)^\top \mathbf{W}_2^{*(l)\top}\right]^\top \tag{115}
$$

$$
= \mathbf{W}_2^{*(l)}\left[\frac{p}{n(p+q)}\boldsymbol{\Sigma}_1^{(l-1)}\right]\mathbf{W}_1^{*(l)\top}
$$

Next, the expectation of: $\mathbf{W}_2^{*(l)}\left(\mathbf{H}^{(l-1)}\widehat{\mathbf{A}}_t\mathbf{e}_{1,i} - \frac{p\boldsymbol{\mu}_1^{(l-1)}+q\boldsymbol{\mu}_2^{(l-1)}}{p+q}\right)\left(\mathbf{H}^{(l-1)}\widehat{\mathbf{A}}_t\mathbf{e}_{1,i} - \frac{p\boldsymbol{\mu}_1^{(l-1)}+q\boldsymbol{\mu}_2^{(l-1)}}{p+q}\right)^\top \mathbf{W}_2^{*(l)\top}$
can be computed as follows:

$$\mathbb{E}_{\widehat{\mathbf{A}},\mathbf{h}}\left[\mathbf{W}_2^{*(l)}\left(\mathbf{H}^{(l-1)}\widehat{\mathbf{A}}_t\mathbf{e}_{1,i} - \frac{p\boldsymbol{\mu}_1^{(l-1)}+q\boldsymbol{\mu}_2^{(l-1)}}{p+q}\right)\right.$$

$$\left.\cdot\left(\mathbf{H}^{(l-1)}\widehat{\mathbf{A}}_t\mathbf{e}_{1,i} - \frac{p\boldsymbol{\mu}_1^{(l-1)}+q\boldsymbol{\mu}_2^{(l-1)}}{p+q}\right)^\top \mathbf{W}_2^{*(l)\top}\right]$$

$$= \mathbb{E}_{\widehat{\mathbf{A}},\mathbf{h}}\left[\mathbf{W}_2^{*(l)}\left(\mathbf{H}^{(l-1)}\widehat{\mathbf{A}}_t\mathbf{e}_{1,i}\right)\left(\mathbf{H}^{(l-1)}\widehat{\mathbf{A}}_t\mathbf{e}_{1,i} - \frac{p\boldsymbol{\mu}_1^{(l-1)}+q\boldsymbol{\mu}_2^{(l-1)}}{p+q}\right)^\top \mathbf{W}_2^{*(l)\top}\right]$$

$$- \mathbb{E}_{\widehat{\mathbf{A}},\mathbf{h}}\left[\mathbf{W}_2^{*(l)}\left(\frac{p\boldsymbol{\mu}_1^{(l-1)}+q\boldsymbol{\mu}_2^{(l-1)}}{p+q}\right)\left(\mathbf{H}^{(l-1)}\widehat{\mathbf{A}}_t\mathbf{e}_{1,i} - \frac{p\boldsymbol{\mu}_1^{(l-1)}+q\boldsymbol{\mu}_2^{(l-1)}}{p+q}\right)^\top \mathbf{W}_2^{*(l)\top}\right]$$

Where the second term reduces to $0$. On expanding the first term, we get:

$$\mathbb{E}_{\widehat{\mathbf{A}},\mathbf{h}}\left[\mathbf{W}_2^{*(l)}\left(\mathbf{H}^{(l-1)}\widehat{\mathbf{A}}_t\mathbf{e}_{1,i}\right)\left(\mathbf{H}^{(l-1)}\widehat{\mathbf{A}}_t\mathbf{e}_{1,i} - \frac{p\boldsymbol{\mu}_1^{(l-1)}+q\boldsymbol{\mu}_2^{(l-1)}}{p+q}\right)^\top \mathbf{W}_2^{*(l)\top}\right]$$

$$= \mathbf{W}_2^{*(l)}\mathbb{E}_{\widehat{\mathbf{A}},\mathbf{h}}\left[\left(\mathbf{H}^{(l-1)}\widehat{\mathbf{A}}_t\mathbf{e}_{1,i}\right)\left(\mathbf{H}^{(l-1)}\widehat{\mathbf{A}}_t\mathbf{e}_{1,i}\right)^\top\right]\mathbf{W}_2^{*(l)\top}$$

$$- \mathbf{W}_2^{*(l)}\mathbb{E}_{\widehat{\mathbf{A}},\mathbf{h}}\left[\left(\mathbf{H}^{(l-1)}\widehat{\mathbf{A}}_t\mathbf{e}_{1,i}\right)\left(\frac{p\boldsymbol{\mu}_1^{(l-1)}+q\boldsymbol{\mu}_2^{(l-1)}}{p+q}\right)^\top\right]\mathbf{W}_2^{*(l)\top}$$

$$= \mathbf{W}_2^{*(l)}\mathbb{E}_{\mathbf{h}}\left[\left(\frac{p\sum_{j=1}^n\mathbf{h}_{1,j}^{(l-1)}+q\sum_{j=1}^n\mathbf{h}_{2,j}^{(l-1)}}{n(p+q)}\right)\left(\frac{p\sum_{j=1}^n\mathbf{h}_{1,j}^{(l-1)}+q\sum_{j=1}^n\mathbf{h}_{2,j}^{(l-1)}}{n(p+q)}\right)^\top\right]\mathbf{W}_2^{*(l)\top}$$

$$- \mathbf{W}_2^{*(l)}\left[\left(\frac{p\boldsymbol{\mu}_1^{(l-1)}+q\boldsymbol{\mu}_2^{(l-1)}}{p+q}\right)\left(\frac{p\boldsymbol{\mu}_1^{(l-1)}+q\boldsymbol{\mu}_2^{(l-1)}}{p+q}\right)^\top\right]\mathbf{W}_2^{*(l)\top}$$

$$= \mathbf{W}_2^{*(l)}\mathbb{E}_{\mathbf{h}}\left[\left(\frac{p^2\sum_{j=1}^n\mathbf{h}_{1,j}^{(l-1)}\mathbf{h}_{1,j}^{(l-1)\top}+q^2\sum_{j=1}^n\mathbf{h}_{2,j}^{(l-1)}\mathbf{h}_{2,j}^{(l-1)\top}}{n^2(p+q)^2}\right)\right]\mathbf{W}_2^{*(l)\top}$$

$$+ \mathbf{W}_2^{*(l)}\mathbb{E}_{\mathbf{h}}\left[\left(\frac{p^2\sum_{j=1}^n\sum_{j'=1,\neq j}^n\mathbf{h}_{1,j}^{(l-1)}\mathbf{h}_{1,j'}^{(l-1)\top}+q^2\sum_{j=1}^n\sum_{j'=1,\neq j}^n\mathbf{h}_{2,j}^{(l-1)}\mathbf{h}_{2,j'}^{(l-1)\top}}{n^2(p+q)^2}\right)\right]\mathbf{W}_2^{*(l)\top}$$

$$+ \mathbf{W}_2^{*(l)}\mathbb{E}_{\mathbf{h}}\left[\left(\frac{pq\sum_{j=1}^n\sum_{j'=1}^n\mathbf{h}_{1,j}^{(l-1)}\mathbf{h}_{2,j'}^{(l-1)\top}+pq\sum_{j=1}^n\sum_{j'=1}^n\mathbf{h}_{2,j}^{(l-1)}\mathbf{h}_{1,j'}^{(l-1)\top}}{n^2(p+q)^2}\right)\right]\mathbf{W}_2^{*(l)\top}$$

$$- \mathbf{W}_2^{*(l)}\left[\left(\frac{p\boldsymbol{\mu}_1^{(l-1)}+q\boldsymbol{\mu}_2^{(l-1)}}{p+q}\right)\left(\frac{p\boldsymbol{\mu}_1^{(l-1)}+q\boldsymbol{\mu}_2^{(l-1)}}{p+q}\right)^\top\right]\mathbf{W}_2^{*(l)\top}$$

Due to the independence of the features, we can simplify the expectation as follows:

$$\mathbb{E}_{\widehat{\mathbf{A}},\mathbf{h}}\left[\mathbf{W}_2^{*(l)}\left(\mathbf{H}^{(l-1)}\widehat{\mathbf{A}}_t\mathbf{e}_{1,i}\right)\left(\mathbf{H}^{(l-1)}\widehat{\mathbf{A}}_t\mathbf{e}_{1,i}-\frac{p\boldsymbol{\mu}_1^{(l-1)}+q\boldsymbol{\mu}_2^{(l-1)}}{p+q}\right)^\top\mathbf{W}_2^{*(l)\top}\right]$$

$$=\mathbf{W}_2^{*(l)}\left[\left(\frac{np^2\left(\boldsymbol{\Sigma}_1^{(l-1)}+\boldsymbol{\mu}_1^{(l-1)}\boldsymbol{\mu}_1^{(l-1)\top}\right)+nq^2\left(\boldsymbol{\Sigma}_2^{(l-1)}+\boldsymbol{\mu}_2^{(l-1)}\boldsymbol{\mu}_2^{(l-1)\top}\right)}{n^2(p+q)^2}\right)\right]\mathbf{W}_2^{*(l)\top}$$

$$+\mathbf{W}_2^{*(l)}\left[\left(\frac{p^2(n^2-n)\boldsymbol{\mu}_1^{(l-1)}\boldsymbol{\mu}_1^{(l-1)\top}+q^2(n^2-n)\boldsymbol{\mu}_2^{(l-1)}\boldsymbol{\mu}_2^{(l-1)\top}}{n^2(p+q)^2}\right)\right]\mathbf{W}_2^{*(l)\top}$$

$$+\mathbf{W}_2^{*(l)}\left[\left(\frac{pqn^2\boldsymbol{\mu}_1^{(l-1)}\boldsymbol{\mu}_2^{(l-1)\top}+pqn^2\boldsymbol{\mu}_2^{(l-1)}\boldsymbol{\mu}_1^{(l-1)\top}}{n^2(p+q)^2}\right)\right]\mathbf{W}_2^{*(l)\top}$$

$$-\mathbf{W}_2^{*(l)}\left[\left(\frac{p^2\boldsymbol{\mu}_1^{(l-1)}\boldsymbol{\mu}_1^{(l-1)\top}+pq\boldsymbol{\mu}_1^{(l-1)}\boldsymbol{\mu}_2^{(l-1)\top}+pq\boldsymbol{\mu}_2^{(l-1)}\boldsymbol{\mu}_1^\top+q^2\boldsymbol{\mu}_2^{(l-1)}\boldsymbol{\mu}_2^{(l-1)\top}}{(p+q)^2}\right)\right]\mathbf{W}_2^{*(l)\top}$$

$$=\mathbf{W}_2^{*(l)}\left[\frac{p^2\boldsymbol{\Sigma}_1^{(l-1)}+q^2\boldsymbol{\Sigma}_2^{(l-1)}}{n(p+q)^2}\right]\mathbf{W}_2^{*(l)\top}$$

$$(116)$$

Putting these results together, the covariance $\mathbb{E}_{\widehat{\mathbf{A}},\mathbf{h}}\left(\left[\mathbf{x}_{1,i}^{(l)}-\mathbb{E}_{\widehat{\mathbf{A}},\mathbf{h}}\mathbf{x}_{1,i}^{(l)}\right]\left[\mathbf{x}_{1,i}^{(l)}-\mathbb{E}_{\widehat{\mathbf{A}},\mathbf{h}}\mathbf{x}_{1,i}^{(l)}\right]^\top\right)$ is:

$$\mathbb{E}_{\widehat{\mathbf{A}},\mathbf{h}}\left(\left[\mathbf{x}_{1,i}^{(l)}-\mathbb{E}_{\widehat{\mathbf{A}},\mathbf{h}}\mathbf{x}_{1,i}^{(l)}\right]\left[\mathbf{x}_{1,i}^{(l)}-\mathbb{E}_{\widehat{\mathbf{A}},\mathbf{h}}\mathbf{x}_{1,i}^{(l)}\right]^\top\right)$$

$$=\mathbf{W}_1^{*(l)}\boldsymbol{\Sigma}_1^{(l-1)}\mathbf{W}_1^{*(l)\top}+\frac{p}{n(p+q)}\mathbf{W}_1^{*(l)}\boldsymbol{\Sigma}_1^{(l-1)}\mathbf{W}_2^{*(l)\top}$$

$$+\frac{p}{n(p+q)}\mathbf{W}_2^{*(l)}\boldsymbol{\Sigma}_1^{(l-1)}\mathbf{W}_1^{*(l)\top}+\mathbf{W}_2^{*(l)}\left[\frac{p^2\boldsymbol{\Sigma}_1^{(l-1)}+q^2\boldsymbol{\Sigma}_2^{(l-1)}}{n(p+q)^2}\right]\mathbf{W}_2^{*(l)\top}$$

$$(117)$$

To summarize, the means and covariance matrices for $\mathbf{X}^{(l)}$ can be given by:

$$\widetilde{\boldsymbol{\mu}}_1^{(l)}=\left(\mathbf{W}_1^{*(l)}+\frac{p}{p+q}\mathbf{W}_2^{*(l)}\right)\boldsymbol{\mu}_1^{(l-1)}+\left(\frac{q}{p+q}\mathbf{W}_2^{*(l)}\right)\boldsymbol{\mu}_2^{(l-1)}$$

$$\widetilde{\boldsymbol{\Sigma}}_1^{(l)}=\mathbf{W}_1^{*(l)}\boldsymbol{\Sigma}_1^{(l-1)}\mathbf{W}_1^{*(l)\top}+\frac{p}{n(p+q)}\mathbf{W}_1^{*(l)}\boldsymbol{\Sigma}_1^{(l-1)}\mathbf{W}_2^{*(l)\top}$$

$$+\frac{p}{n(p+q)}\mathbf{W}_2^{*(l)}\boldsymbol{\Sigma}_1^{(l-1)}\mathbf{W}_1^{*(l)\top}+\mathbf{W}_2^{*(l)}\left[\frac{p^2\boldsymbol{\Sigma}_1^{(l-1)}+q^2\boldsymbol{\Sigma}_2^{(l-1)}}{n(p+q)^2}\right]\mathbf{W}_2^{*(l)\top}$$

$$(118)$$

• **Case** $c=2$:

The analysis presented above can be extended for $c=2$ in a straightforward fashion to get $\widetilde{\boldsymbol{\mu}}_2^{(l)},\widetilde{\boldsymbol{\Sigma}}_2^{(l)}$ as follows:

$$\widetilde{\boldsymbol{\mu}}_2^{(l)}=\left(\mathbf{W}_1^{*(l)}+\frac{p}{p+q}\mathbf{W}_2^{*(l)}\right)\boldsymbol{\mu}_2^{(l-1)}+\left(\frac{q}{p+q}\mathbf{W}_2^{*(l)}\right)\boldsymbol{\mu}_1^{(l-1)}$$

$$\widetilde{\boldsymbol{\Sigma}}_2^{(l)}=\mathbf{W}_1^{*(l)}\boldsymbol{\Sigma}_2^{(l-1)}\mathbf{W}_1^{*(l)\top}+\frac{p}{n(p+q)}\mathbf{W}_1^{*(l)}\boldsymbol{\Sigma}_2^{(l-1)}\mathbf{W}_2^{*(l)\top}$$

$$+\frac{p}{n(p+q)}\mathbf{W}_2^{*(l)}\boldsymbol{\Sigma}_2^{(l-1)}\mathbf{W}_1^{*(l)\top}+\mathbf{W}_2^{*(l)}\left[\frac{p^2\boldsymbol{\Sigma}_2^{(l-1)}+q^2\boldsymbol{\Sigma}_1^{(l-1)}}{n(p+q)^2}\right]\mathbf{W}_2^{*(l)\top}$$

$$(119)$$

## G.1 Modelling an increase/decrease in between-class variability

Let $\Sigma_B^{(l-1)} = \left(\boldsymbol{\mu}_1^{(l-1)} - \boldsymbol{\mu}_2^{(l-1)}\right)\left(\boldsymbol{\mu}_1^{(l-1)} - \boldsymbol{\mu}_2^{(l-1)}\right)^{\top}$ indicate the between-class covariance matrix for features at layer $l-1$. Based on the equations 118, 119, we analyze $\Sigma_B(\mathbf{X}^{(l)})$ to understand the effect of the convolution layer on feature separation. First, observe that:

$$
\begin{aligned}
\boldsymbol{\Sigma}_B(\mathbf{X}^{(l)}) &= \left(\widetilde{\boldsymbol{\mu}}_1^{(l)} - \widetilde{\boldsymbol{\mu}}_2^{(l)}\right)\left(\widetilde{\boldsymbol{\mu}}_1^{(l)} - \widetilde{\boldsymbol{\mu}}_2^{(l)}\right)^{\top} \\
&= \left(\mathbf{W}_1^{*(l)} + \frac{p-q}{p+q}\mathbf{W}_2^{*(l)}\right)\left(\boldsymbol{\mu}_1^{(l-1)} - \boldsymbol{\mu}_2^{(l-1)}\right) \\
&\quad \cdot \left(\boldsymbol{\mu}_1^{(l-1)} - \boldsymbol{\mu}_2^{(l-1)}\right)^{\top}\left(\mathbf{W}_1^{*(l)} + \frac{p-q}{p+q}\mathbf{W}_2^{*(l)}\right)^{\top} \\
&= \left(\mathbf{W}_1^{*(l)} + \frac{p-q}{p+q}\mathbf{W}_2^{*(l)}\right)\boldsymbol{\Sigma}_B(\mathbf{H}^{(l-1)})\left(\mathbf{W}_1^{*(l)} + \frac{p-q}{p+q}\mathbf{W}_2^{*(l)}\right)^{\top}
\end{aligned}
\tag{120}
$$

By taking the trace on both sides, we get:

$$
\begin{aligned}
\mathrm{Tr}\left(\boldsymbol{\Sigma}_B(\mathbf{X}^{(l)})\right) &= \mathrm{Tr}\left(\left(\mathbf{W}_1^{*(l)} + \frac{p-q}{p+q}\mathbf{W}_2^{*(l)}\right)\boldsymbol{\Sigma}_B(\mathbf{H}^{(l-1)})\left(\mathbf{W}_1^{*(l)} + \frac{p-q}{p+q}\mathbf{W}_2^{*(l)}\right)^{\top}\right) \\
&= \mathrm{Tr}\left(\boldsymbol{\Sigma}_B(\mathbf{H}^{(l-1)})\left(\mathbf{W}_1^{*(l)} + \frac{p-q}{p+q}\mathbf{W}_2^{*(l)}\right)^{\top}\left(\mathbf{W}_1^{*(l)} + \frac{p-q}{p+q}\mathbf{W}_2^{*(l)}\right)\right)
\end{aligned}
\tag{121}
$$

Where $\left(\mathbf{W}_1^{*(l)} + \frac{p-q}{p+q}\mathbf{W}_2^{*(l)}\right)^{\top}\left(\mathbf{W}_1^{*(l)} + \frac{p-q}{p+q}\mathbf{W}_2^{*(l)}\right) \in \mathbb{R}^{d_{l-1}\times d_{l-1}}$ is a symmetric and positive semi-definite matrix. This matrix product formulation allows us to leverage the eigenvalue-based trace inequalities [41, 68]. Formally, let $\mathbf{T}_B = \left(\mathbf{W}_1^{*(l)} + \frac{p-q}{p+q}\mathbf{W}_2^{*(l)}\right)^{\top}\left(\mathbf{W}_1^{*(l)} + \frac{p-q}{p+q}\mathbf{W}_2^{*(l)}\right)$. We now leverage Corollary.6 in Zhang and Zhang [68], and get the following inequality based on the eigenvalues of $\boldsymbol{\Sigma}_B(\mathbf{H}^{(l-1)}), \mathbf{T}_B$ as:

$$
\begin{aligned}
\sum_{i=1}^{d_{l-1}} \lambda_{d_{l-1}-i+1}\left(\boldsymbol{\Sigma}_B(\mathbf{H}^{(l-1)})\right)\lambda_i\left(\mathbf{T}_B\right) & \\
&\leq \mathrm{Tr}\left(\boldsymbol{\Sigma}_B(\mathbf{H}^{(l-1)})\mathbf{T}_B\right) \\
&\leq \sum_{i=1}^{d_{l-1}} \lambda_i\left(\boldsymbol{\Sigma}_B(\mathbf{H}^{(l-1)})\right)\lambda_i\left(\mathbf{T}_B\right)
\end{aligned}
\tag{122}
$$

Where $\lambda_i(.)$ represents the $i^{th}$ largest eigenvalue of a matrix. Additionally, based on the standard trace equality: $\mathrm{Tr}\left(\boldsymbol{\Sigma}_B(\mathbf{H}^{(l-1)})\right) = \sum_{i=1}^{d_{l-1}} \lambda_i\left(\boldsymbol{\Sigma}_B(\mathbf{H}^{(l-1)})\right)$, the increase/decrease in $\mathrm{Tr}\left(\boldsymbol{\Sigma}_B(\mathbf{X}^{(l)})\right)$ with respect to $\mathrm{Tr}\left(\boldsymbol{\Sigma}_B(\mathbf{H}^{(l-1)})\right)$ boils down to:

$$
\frac{\sum_{i=1}^{d_{l-1}} \lambda_{d_{l-1}-i+1}\left(\boldsymbol{\Sigma}_B(\mathbf{H}^{(l-1)})\right)\lambda_i\left(\mathbf{T}_B\right)}{\sum_{i=1}^{d_{l-1}} \lambda_i\left(\boldsymbol{\Sigma}_B(\mathbf{H}^{(l-1)})\right)} \leq \frac{\mathrm{Tr}(\boldsymbol{\Sigma}_B(\mathbf{X}^{(l)}))}{\mathrm{Tr}(\boldsymbol{\Sigma}_B(\mathbf{H}^{(l-1)}))} \leq \frac{\sum_{i=1}^{d_{l-1}} \lambda_i\left(\boldsymbol{\Sigma}_B(\mathbf{H}^{(l-1)})\right)\lambda_i\left(\mathbf{T}_B\right)}{\sum_{i=1}^{d_{l-1}} \lambda_i\left(\boldsymbol{\Sigma}_B(\mathbf{H}^{(l-1)})\right)}
\tag{123}
$$

## G.2 Modelling an increase/decrease in within-class variability

Let $\boldsymbol{\Sigma}_W(\mathbf{H}^{(l-1)}) = \frac{1}{2}\left(\boldsymbol{\Sigma}_1^{(l-1)} + \boldsymbol{\Sigma}_2^{(l-1)}\right)$ represent the within-class covariance matrix for features $\mathbf{H}^{(l-1)}$ in our balanced class setting. Similar to the previous analysis, we leverage the results in

equations 118,119 to model $\boldsymbol{\Sigma}_W(\mathbf{X}^{(l)})$ as follows:

$$
\begin{aligned}
\boldsymbol{\Sigma}_W(\mathbf{X}^{(l)}) &= \frac{1}{2}\left(\widetilde{\boldsymbol{\Sigma}}_1^{(l)} + \widetilde{\boldsymbol{\Sigma}}_2^{(l)}\right) \\
&= \frac{1}{2}\left(\mathbf{W}_1^{*(l)}\left(\boldsymbol{\Sigma}_1^{(l-1)} + \boldsymbol{\Sigma}_2^{(l-1)}\right)\mathbf{W}_1^{*(l)\top}\right) \\
&\quad + \frac{1}{2}\left(\frac{p}{n(p+q)}\mathbf{W}_1^{*(l)}\left(\boldsymbol{\Sigma}_1^{(l-1)} + \boldsymbol{\Sigma}_2^{(l-1)}\right)\mathbf{W}_2^{*(l)\top}\right) \\
&\quad + \frac{1}{2}\left(\frac{p}{n(p+q)}\mathbf{W}_2^{*(l)}\left(\boldsymbol{\Sigma}_1^{(l-1)} + \boldsymbol{\Sigma}_2^{(l-1)}\right)\mathbf{W}_1^{*(l)\top}\right) \\
&\quad + \frac{1}{2}\left(\mathbf{W}_2^{*(l)}\left[\frac{(p^2+q^2)\left(\boldsymbol{\Sigma}_1^{(l-1)} + \boldsymbol{\Sigma}_2^{(l-1)}\right)}{n(p+q)^2}\right]\mathbf{W}_2^{*(l)\top}\right)
\end{aligned}
\tag{124}
$$

By taking trace on both sides, we get:

$$
\begin{aligned}
\mathrm{Tr}\left(\boldsymbol{\Sigma}_W(\mathbf{X}^{(l)})\right) &= \mathrm{Tr}\left(\boldsymbol{\Sigma}_W(\mathbf{H}^{(l-1)})\mathbf{W}_1^{*(l)\top}\mathbf{W}_1^{*(l)}\right) \\
&\quad + \mathrm{Tr}\left(\frac{p}{n(p+q)}\boldsymbol{\Sigma}_W(\mathbf{H}^{(l-1)})\mathbf{W}_2^{*(l)\top}\mathbf{W}_1^{*(l)}\right) \\
&\quad + \mathrm{Tr}\left(\frac{p}{n(p+q)}\boldsymbol{\Sigma}_W(\mathbf{H}^{(l-1)})\mathbf{W}_1^{*(l)\top}\mathbf{W}_2^{*(l)}\right) \\
&\quad + \mathrm{Tr}\left(\frac{(p^2+q^2)}{n(p+q)^2}\boldsymbol{\Sigma}_W(\mathbf{H}^{(l-1)})\mathbf{W}_2^{*(l)\top}\mathbf{W}_2^{*(l)}\right) \\
&= \mathrm{Tr}\left(\boldsymbol{\Sigma}_W(\mathbf{H}^{(l-1)})\left[\mathbf{W}_1^{*(l)\top}\mathbf{W}_1^{*(l)} + \frac{p}{n(p+q)}\left[\mathbf{W}_2^{*(l)\top}\mathbf{W}_1^{*(l)} + \mathbf{W}_1^{*(l)\top}\mathbf{W}_2^{*(l)}\right]\right.\right. \\
&\quad \left.\left. + \frac{(p^2+q^2)}{n(p+q)^2}\mathbf{W}_2^{*(l)\top}\mathbf{W}_2^{*(l)}\right]\right)
\end{aligned}
\tag{125}
$$

Let $\mathbf{T}_W = \mathbf{W}_1^{*(l)\top}\mathbf{W}_1^{*(l)} + \frac{p}{n(p+q)}\left[\mathbf{W}_2^{*(l)\top}\mathbf{W}_1^{*(l)} + \mathbf{W}_1^{*(l)\top}\mathbf{W}_2^{*(l)}\right] + \frac{(p^2+q^2)}{n(p+q)^2}\mathbf{W}_2^{*(l)\top}\mathbf{W}_2^{*(l)}$. Then, observe that $\mathbf{T}_W \in \mathbb{R}^{d_{l-1} \times d_{l-1}}$ is symmetric and positive semi-definite. To this end, the increase/decrease in $\mathrm{Tr}\left(\boldsymbol{\Sigma}_W(\mathbf{X}^{(l)})\right)$ with respect to $\mathrm{Tr}\left(\boldsymbol{\Sigma}_W(\mathbf{H}^{(l-1)})\right)$ boils down to:

$$
\frac{\displaystyle\sum_{i=1}^{d_{l-1}} \lambda_{d_{l-1}-i+1}\left(\boldsymbol{\Sigma}_W(\mathbf{H}^{(l-1)})\right)\lambda_i\left(\mathbf{T}_W\right)}{\displaystyle\sum_{i=1}^{d_{l-1}} \lambda_i\left(\boldsymbol{\Sigma}_W(\mathbf{H}^{(l-1)})\right)} \leq \frac{\mathrm{Tr}(\boldsymbol{\Sigma}_W(\mathbf{X}^{(l)}))}{\mathrm{Tr}(\boldsymbol{\Sigma}_W(\mathbf{H}^{(l-1)}))} \leq \frac{\displaystyle\sum_{i=1}^{d_{l-1}} \lambda_i\left(\boldsymbol{\Sigma}_W(\mathbf{H}^{(l-1)})\right)\lambda_i\left(\mathbf{T}_W\right)}{\displaystyle\sum_{i=1}^{d_{l-1}} \lambda_i\left(\boldsymbol{\Sigma}_W(\mathbf{H}^{(l-1)})\right)}
\tag{126}
$$

# H  Addition Experiments

• **Infrastructure details:** We perform experiments on a virtual machine with 8 Intel(R) Xeon(R) Platinum 8268 CPUs, 32GB of RAM, and 1 Quadro RTX 8000 GPU with 32GB of allocated memory. Our Python package *'gnn_collapse'* leverages PyTorch 1.12.1 and PyTorch-Geometric (PyG) 2.1.0 frameworks. For reproducible experiments and consistency with previous research, we extend the SBM generator by Chen et al. [16] and NC metrics by Zhu et al. [74].

## H.1  Experiments with GNNs to track penultimate layer features

• **Datasets:** We consider a variety of SSBM graph datasets as follows:

**D1:** $C = 2, N = 1000, p = 0.025, q = 0.0017, K = 1000$

**D2:** $C = 4, N = 1500, p = 0.072, q = 0.0048, K = 1000$

• **GNNs:** In our experiments, we empirically track the NC metrics of penultimate layer features during training for both the GNN designs $\psi_\Theta^{\mathcal{F}}, \psi_\Theta^{\mathcal{F}'}$. The number of layers is set to 32 and the hidden dimension is set to 8 across layers for datasets with $C = 2$ and set to 16 for datasets with $C = 4$.

• **Optimization:** The GNNs are trained for 8 epochs using stochastic gradient descent (SGD) with momentum 0.9, weight decay $5 \times 10^{-4}$, and a learning rate set to 0.004 for **D1** and 0.006 for **D2**.

• **Observations:** Figures 7, 8 illustrate the training loss, overlap and all the NC metrics that we defined in our setup for $\psi_\Theta^{\mathcal{F}}, \psi_\Theta^{\mathcal{F}'}$ on dataset **D1**. Note that when $C = 2$, the re-centering of the 2 class-means by subtracting the global mean, always leads to separation with maximal angle, irrespective of the configuration of the non-centered class-means. Thus, we skip the corresponding $\mathcal{NC}_2$ plots for $\mathbf{H}, \mathbf{H}\widehat{\mathbf{A}}$ when $C = 2$. Additionally, we can observe similar trends in NC metrics from Figures 9, 10 even after increasing $N, C$ in dataset **D2**.

Additionally, in all these experiments, notice that $\mathcal{NC}_2, \mathcal{NC}_3$ metrics do not show a significant reduction. In this context, a reduction indicates that a simplex equiangular tight-frame (simplex ETF) or an orthogonal frame (OF) is the desired configuration for weights and penultimate layer feature (re-centered) class-means. This behaviour can be linked to the presence of $\widehat{\mathbf{A}}$ in the risk formulation. However, our understanding of the role of $\widehat{\mathbf{A}}$ in determining these alignments towards simplex ETF or an OF is still unclear and would be a valuable future effort.

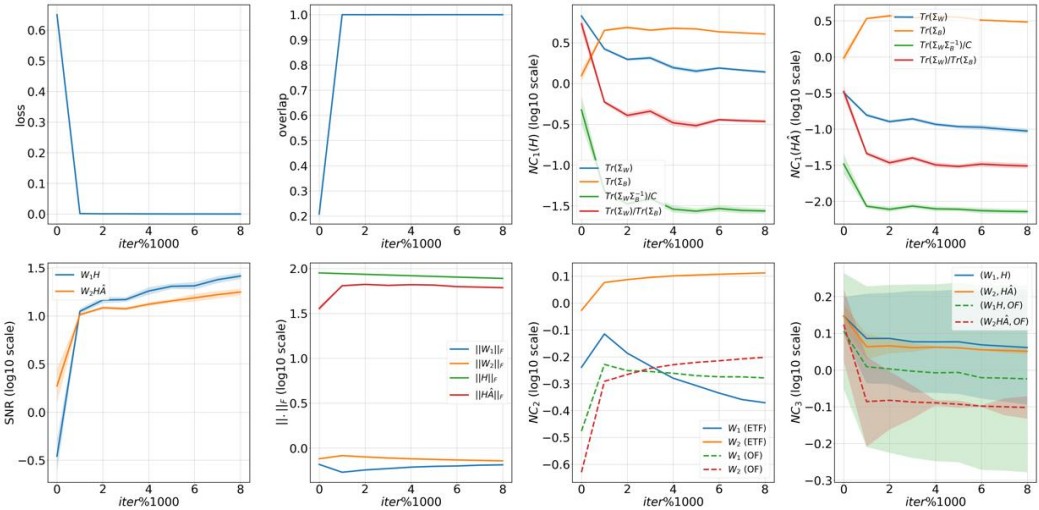

Figure 7: GNN $\psi_\Theta^{\mathcal{F}}$ on **D1:** $C = 2, N = 1000, p = 0.025, q = 0.0017, K = 1000$. Illustration of training loss, training overlap, $\mathcal{NC}_1$ plots for $\mathbf{H}, \mathbf{H}\widehat{\mathbf{A}}$, $SNR(\mathcal{NC}_1)$ for $\mathbf{H}, \mathbf{H}\widehat{\mathbf{A}}$, Frobenius norms of $\mathbf{W}_1, \mathbf{W}_2, \mathbf{H}, \mathbf{H}\widehat{\mathbf{A}}$, $\mathcal{NC}_2$ plots for $\mathbf{W}_1, \mathbf{W}_2$, $\mathcal{NC}_3$ plots for $\mathbf{W}_1, \mathbf{H}$ and $\mathbf{W}_2, \mathbf{H}\widehat{\mathbf{A}}$.

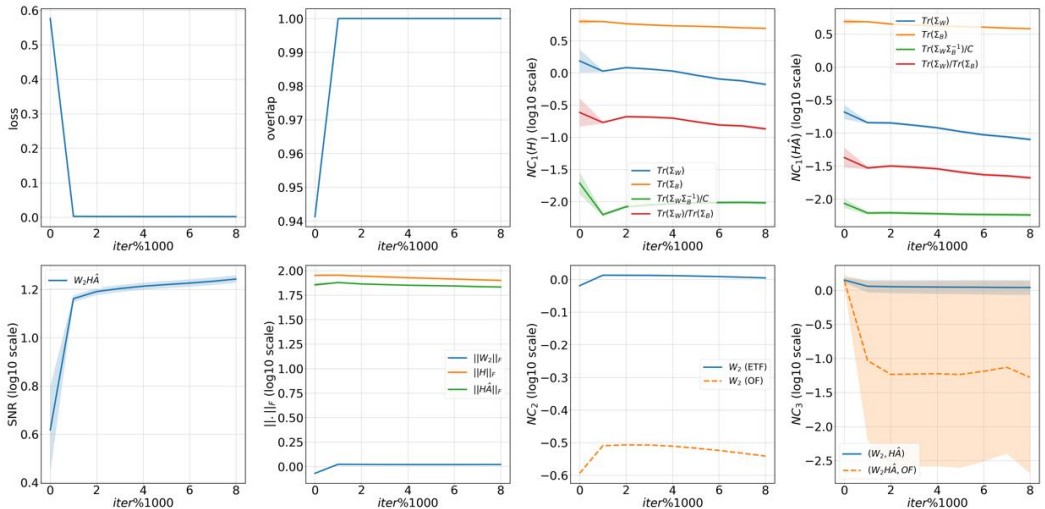

Figure 8: GNN $\psi_\Theta^{\mathcal{F}'}$ on **D1:** $C = 2, N = 1000, p = 0.025, q = 0.0017, K = 1000$. Illustration of training loss, training overlap, $\mathcal{NC}_1$ plots for $\mathbf{H}, \mathbf{H}\widehat{\mathbf{A}}$, $SNR(\mathcal{NC}_1)$ for $\mathbf{H}\widehat{\mathbf{A}}$, Frobenius norms of $\mathbf{W}_2, \mathbf{H}, \mathbf{H}\widehat{\mathbf{A}}$, $\mathcal{NC}_2$ plots for $\mathbf{W}_2$, and $\mathcal{NC}_3$ plots for $\mathbf{W}_2, \mathbf{H}\widehat{\mathbf{A}}$.

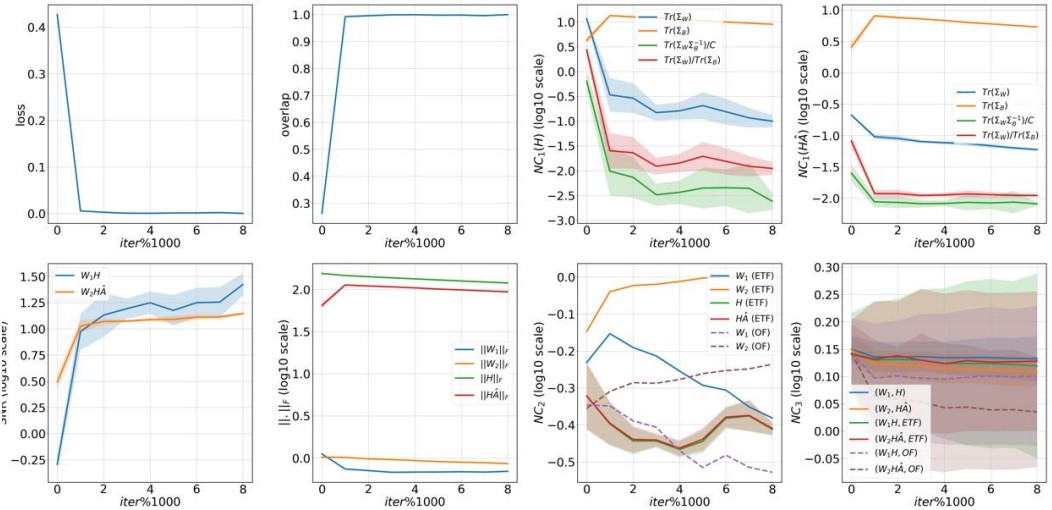

Figure 9: GNN $\psi_\Theta^{\mathcal{F}}$ on **D2:** $C = 4, N = 1500, p = 0.072, q = 0.0048, K = 1000$. Illustration of training loss, training overlap, $\mathcal{NC}_1$ plots for $\mathbf{H}, \mathbf{H}\widehat{\mathbf{A}}$, $SNR(\mathcal{NC}_1)$ for $\mathbf{H}, \mathbf{H}\widehat{\mathbf{A}}$, Frobenius norms of $\mathbf{W}_1, \mathbf{W}_2, \mathbf{H}, \mathbf{H}\widehat{\mathbf{A}}$, $\mathcal{NC}_2$ plots for $\mathbf{W}_1, \mathbf{W}_2, \mathbf{H}, \mathbf{H}\widehat{\mathbf{A}}$, $\mathcal{NC}_3$ plots for $\mathbf{W}_1, \mathbf{H}$ and $\mathbf{W}_2, \mathbf{H}\widehat{\mathbf{A}}$.

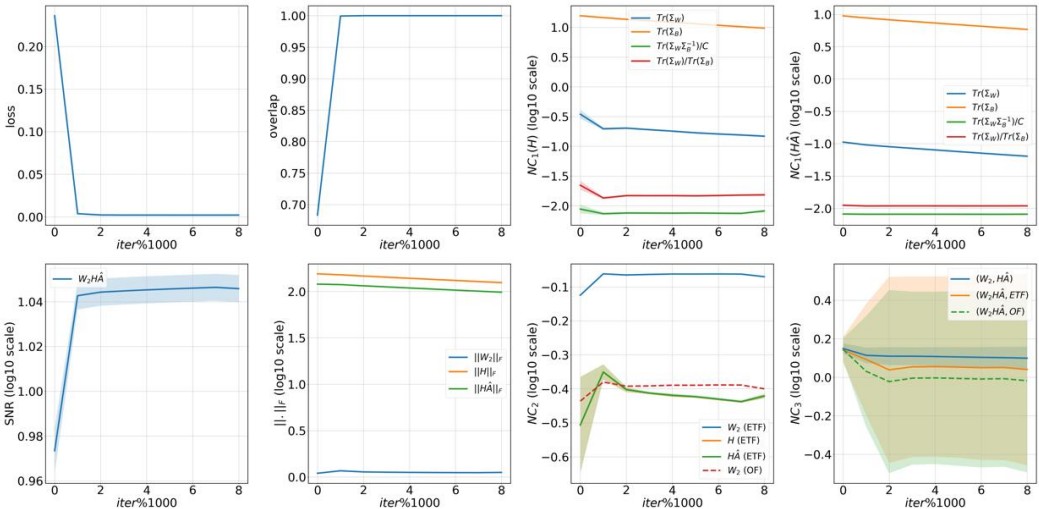

Figure 10: GNN $\psi_\Theta^{\mathcal{F}'}$ on **D2:** $C = 4, N = 1500, p = 0.072, q = 0.0048, K = 1000$. Illustration of training loss, training overlap, $\mathcal{NC}_1$ plots for $\mathbf{H}, \mathbf{H\widehat{A}}$, $SNR(\mathcal{NC}_1)$ for $\mathbf{H\widehat{A}}$, Frobenius norms of $\mathbf{W}_2, \mathbf{H}, \mathbf{H\widehat{A}}$, $\mathcal{NC}_2$ plots for $\mathbf{W}_2, \mathbf{H}, \mathbf{H\widehat{A}}$, and $\mathcal{NC}_3$ plots for $\mathbf{W}_2, \mathbf{H\widehat{A}}$.

## H.2 Experiments with UFM to model penultimate layer behavior

To improve our understanding of the empirical behavior of GNNs and to validate our theoretical results with the gUFM, we prepare datasets based on two strategies:

- Case $\mathbf{C}^+$: This case represents a graph that satisfies condition $\mathbf{C}$. Without loss of generality, we leverage the expected degrees of nodes and consider $t_{cc} = \lceil n * p \rceil, c \in [C]$ and $t_{cc'} = \lceil n * q \rceil, c \neq c' \in [C]$. Based on our notation, recall that $\Omega_c, \Omega_{c'}$ represents the set of nodes belonging to classes(communities) $c, c' \in [C]$ respectively. To this end, observe that the following conditions should be satisfied[12]:

    1. The sub-graph formed by nodes $\Omega_c$ should be $t_{cc}$-regular, for all $c \in [C]$.
    2. The bipartite sub-graph formed by $\Omega_c, \Omega_{c'}$ should be $t_{cc'}$-regular, for all $c, c' \in [C]$.

- Case $\mathbf{C}^-$: This case represents a graph that does not satisfy condition $\mathbf{C}$. Since any random graph sampled from $\text{SSBM}(N, C, p, q)$ satisfies this requirement with a high probability (as per theorem 3.2), we simply use this randomly sampled graph. As a simple sanity check, one can verify if the sampled graph satisfies the condition $\mathbf{C}$ and re-sample.

Especially, we consider the $\mathbf{C}^+, \mathbf{C}^-$ variants of SSBM graphs with following parameters:

**D1:** $C = 2, N = 1000, p = 0.025, q = 0.0017, K = 10$

**D2:** $C = 4, N = 1500, p = 0.072, q = 0.0048, K = 10$

• **Optimization:** The gUFMs are trained using plain gradient descent for 50000 epochs with a learning rate of 0.1 and L2 regularization parameters $\lambda_{W_1} = \lambda_{W_2} = \lambda_H = 5 \times 10^{-3}$.[13]

• **Observations:** Figures 12, 14, and 16, 18 illustrate the training loss, overlap and the NC metrics for the gUFM acting on $\mathbf{C}^-$ variants of datasets **D1, D2** respectively. Although gUFM is an optimistic mathematical model, we can observe a close resemblance of the values and trends of NC metrics to those of the penultimate layer features of the actual GNNs. This observation is justified as any random SSBM graph fails to satisfy condition $\mathbf{C}$ with a high probability. To this end, observe from Figures 11, 13, 15, 17 that when graphs satisfy condition $\mathbf{C}$, the NC1 metrics tend to reduce drastically (for gUFM designs based on $\psi_\Theta^{\mathcal{F}}, \psi_\Theta^{\mathcal{F}'}$ and $C = 2, 4$). Thus proving our theoretical results. Furthermore,

---

[12]We utilize NetworkX python libraries to generate SSBM graphs that satisfy these conditions.

[13]$\lambda_{W_1}$ is not applicable for gUFM based on $\psi_\Theta^{\mathcal{F}'}$.

observe from Figures 15, 17 that the (re-centered) class means for $\mathbf{H}, \mathbf{H}\widehat{\mathbf{A}}$ tend to align very closely to a simplex ETF and tend to converge at such a configuration. Based on our previous observations for GNN training, we underscore this observation for gUFM and emphasize that a rigorous theoretical analysis on the role of $\widehat{\mathbf{A}}$ in determining the structures of $\mathbf{H}$ can be a crucial future effort.

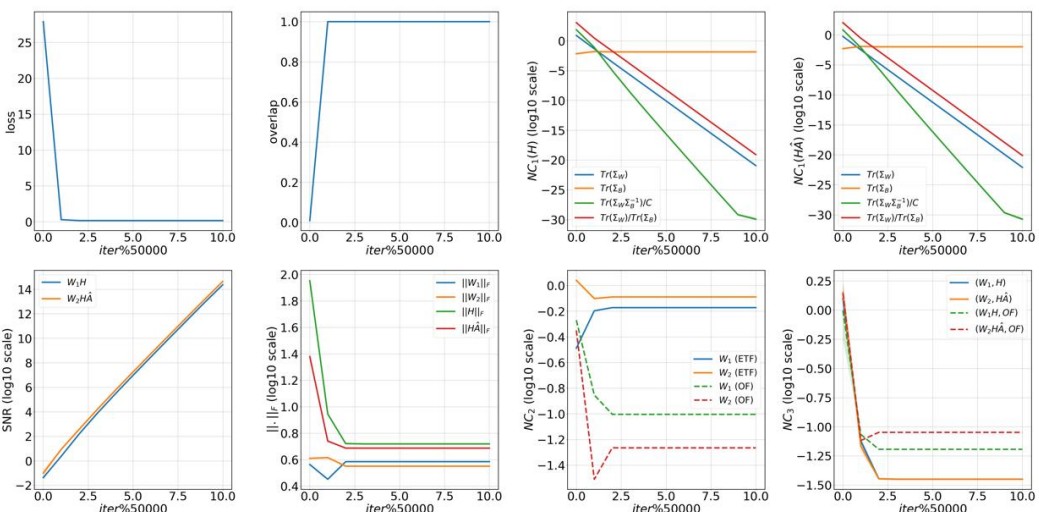

Figure 11: gUFM $\psi_{\Theta}^{\mathcal{F}}$ on $\mathbf{C}^+$ variant of $\mathbf{D1}$: $C = 2, N = 1000, p = 0.025, q = 0.0017, K = 10$. Illustration of training loss, training overlap, $\mathcal{NC}_1$ plots for $\mathbf{H}, \mathbf{H}\widehat{\mathbf{A}}$, $SNR(\mathcal{NC}_1)$ for $\mathbf{H}, \mathbf{H}\widehat{\mathbf{A}}$, Frobenius norms of $\mathbf{W}_1, \mathbf{W}_2, \mathbf{H}, \mathbf{H}\widehat{\mathbf{A}}$, $\mathcal{NC}_2$ plots for $\mathbf{W}_1, \mathbf{W}_2$, $\mathcal{NC}_3$ plots for $\mathbf{W}_1, \mathbf{H}$ and $\mathbf{W}_2, \mathbf{H}\widehat{\mathbf{A}}$.

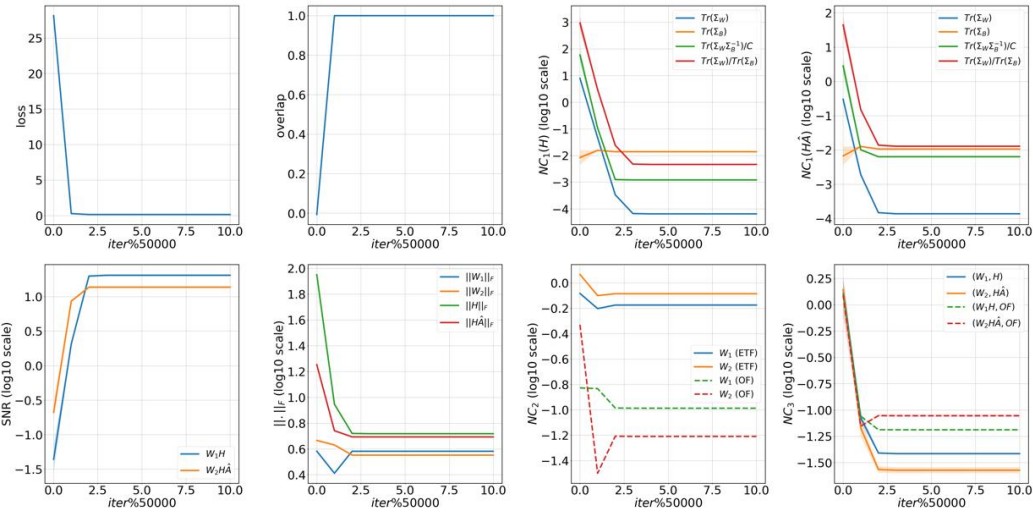

Figure 12: gUFM $\psi_{\Theta}^{\mathcal{F}}$ on $\mathbf{C}^-$ variant of $\mathbf{D1}$: $C = 2, N = 1000, p = 0.025, q = 0.0017, K = 10$. Illustration of training loss, training overlap, $\mathcal{NC}_1$ plots for $\mathbf{H}, \mathbf{H}\widehat{\mathbf{A}}$, $SNR(\mathcal{NC}_1)$ for $\mathbf{H}, \mathbf{H}\widehat{\mathbf{A}}$, Frobenius norms of $\mathbf{W}_1, \mathbf{W}_2, \mathbf{H}, \mathbf{H}\widehat{\mathbf{A}}$, $\mathcal{NC}_2$ plots for $\mathbf{W}_1, \mathbf{W}_2$, $\mathcal{NC}_3$ plots for $\mathbf{W}_1, \mathbf{H}$ and $\mathbf{W}_2, \mathbf{H}\widehat{\mathbf{A}}$.

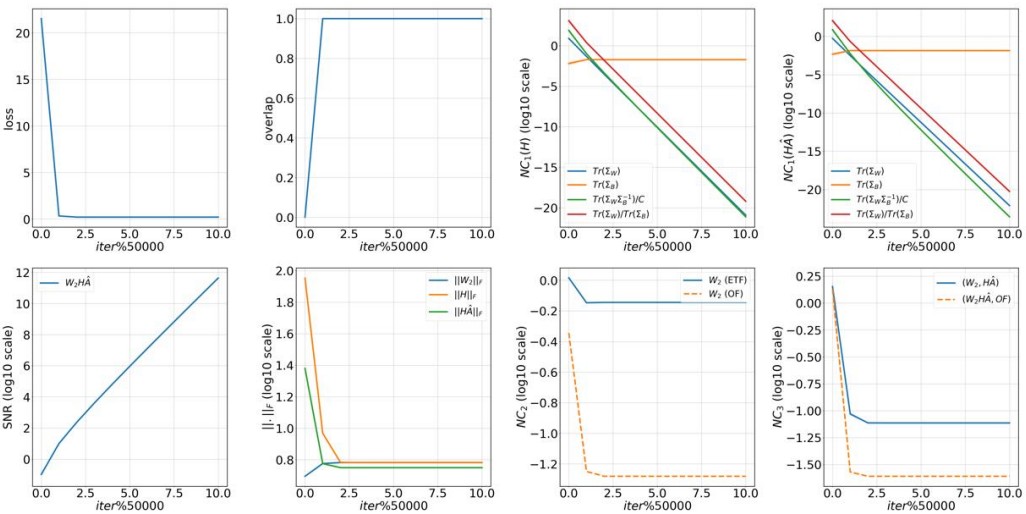

Figure 13: gUFM $\psi_{\Theta}^{\mathcal{F}'}$ on $\mathbf{C}^+$ variant of $\mathbf{D1}$: $C = 2, N = 1000, p = 0.025, q = 0.0017, K = 10$. Illustration of training loss, training overlap, $\mathcal{NC}_1$ plots for $\mathbf{H}, \mathbf{H\widehat{A}}, SNR(\mathcal{NC}_1)$ for $\mathbf{H\widehat{A}}$, Frobenius norms of $\mathbf{W}_2, \mathbf{H}, \mathbf{H\widehat{A}}, \mathcal{NC}_2$ plots for $\mathbf{W}_2$, and $\mathcal{NC}_3$ plots for $\mathbf{W}_2, \mathbf{H\widehat{A}}$.

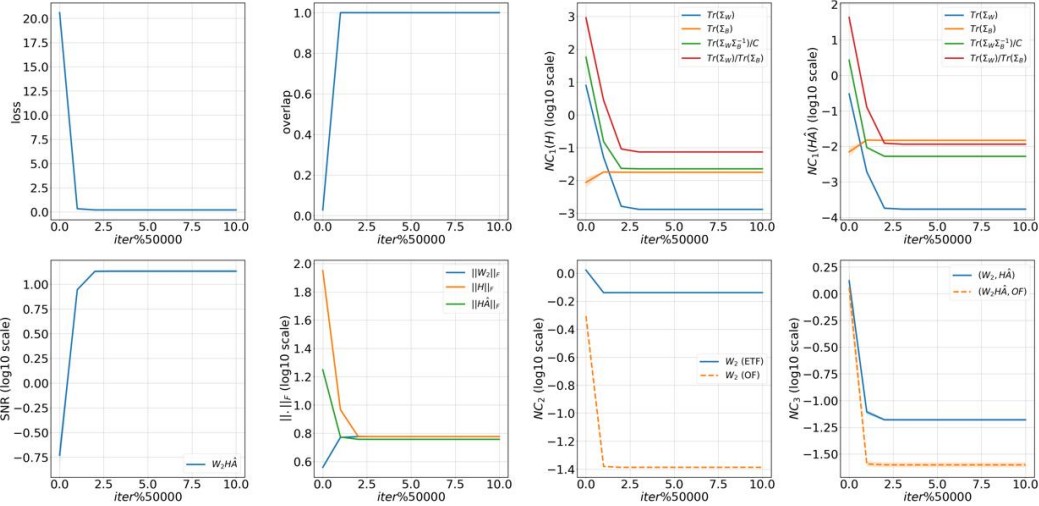

Figure 14: gUFM $\psi_{\Theta}^{\mathcal{F}'}$ on $\mathbf{C}^-$ variant of $\mathbf{D1}$: $C = 2, N = 1000, p = 0.025, q = 0.0017, K = 10$. Illustration of training loss, training overlap, $\mathcal{NC}_1$ plots for $\mathbf{H}, \mathbf{H\widehat{A}}, SNR(\mathcal{NC}_1)$ for $\mathbf{H\widehat{A}}$, Frobenius norms of $\mathbf{W}_2, \mathbf{H}, \mathbf{H\widehat{A}}, \mathcal{NC}_2$ plots for $\mathbf{W}_2$, and $\mathcal{NC}_3$ plots for $\mathbf{W}_2, \mathbf{H\widehat{A}}$.

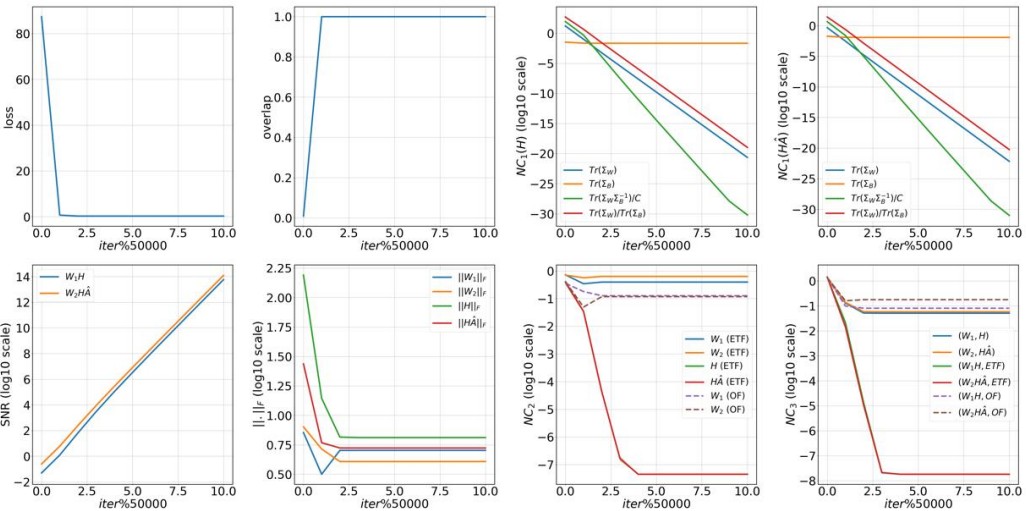

Figure 15: gUFM $\psi_\Theta^{\mathcal{F}}$ on $\mathbf{C}^+$ variant of **D2:** $C = 4, N = 1500, p = 0.072, q = 0.0048, K = 10$. Illustration of training loss, training overlap, $\mathcal{NC}_1$ plots for $\mathbf{H}, \mathbf{H}\widehat{\mathbf{A}}$, $SNR(\mathcal{NC}_1)$ for $\mathbf{H}, \mathbf{H}\widehat{\mathbf{A}}$, Frobenius norms of $\mathbf{W}_1, \mathbf{W}_2, \mathbf{H}, \mathbf{H}\widehat{\mathbf{A}}$, $\mathcal{NC}_2$ plots for $\mathbf{W}_1, \mathbf{W}_2, \mathbf{H}, \mathbf{H}\widehat{\mathbf{A}}$, $\mathcal{NC}_3$ plots for $\mathbf{W}_1, \mathbf{H}$ and $\mathbf{W}_2, \mathbf{H}\widehat{\mathbf{A}}$.

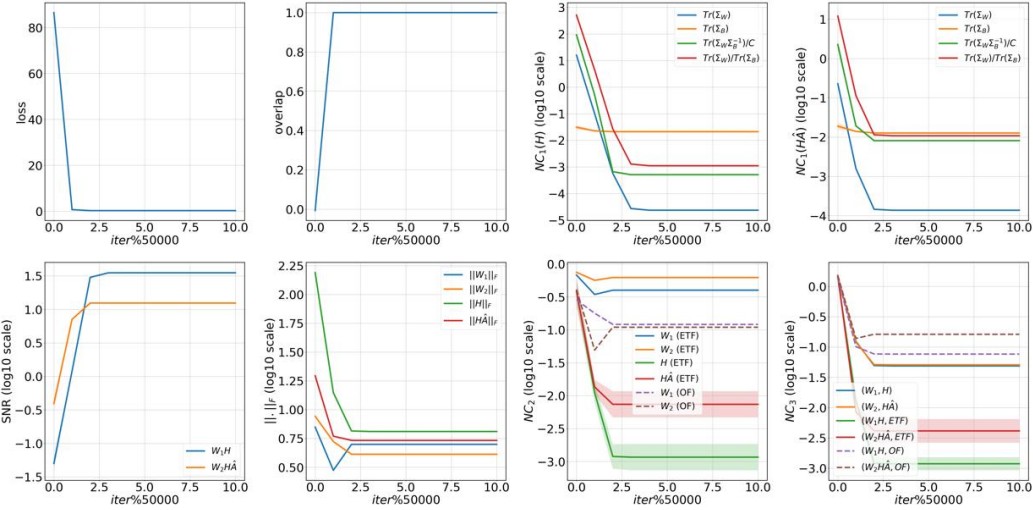

Figure 16: gUFM $\psi_\Theta^{\mathcal{F}}$ on $\mathbf{C}^-$ variant of **D2:** $C = 4, N = 1500, p = 0.072, q = 0.0048, K = 10$. Illustration of training loss, training overlap, $\mathcal{NC}_1$ plots for $\mathbf{H}, \mathbf{H}\widehat{\mathbf{A}}$, $SNR(\mathcal{NC}_1)$ for $\mathbf{H}, \mathbf{H}\widehat{\mathbf{A}}$, Frobenius norms of $\mathbf{W}_1, \mathbf{W}_2, \mathbf{H}, \mathbf{H}\widehat{\mathbf{A}}$, $\mathcal{NC}_2$ plots for $\mathbf{W}_1, \mathbf{W}_2, \mathbf{H}, \mathbf{H}\widehat{\mathbf{A}}$, $\mathcal{NC}_3$ plots for $\mathbf{W}_1, \mathbf{H}$ and $\mathbf{W}_2, \mathbf{H}\widehat{\mathbf{A}}$.

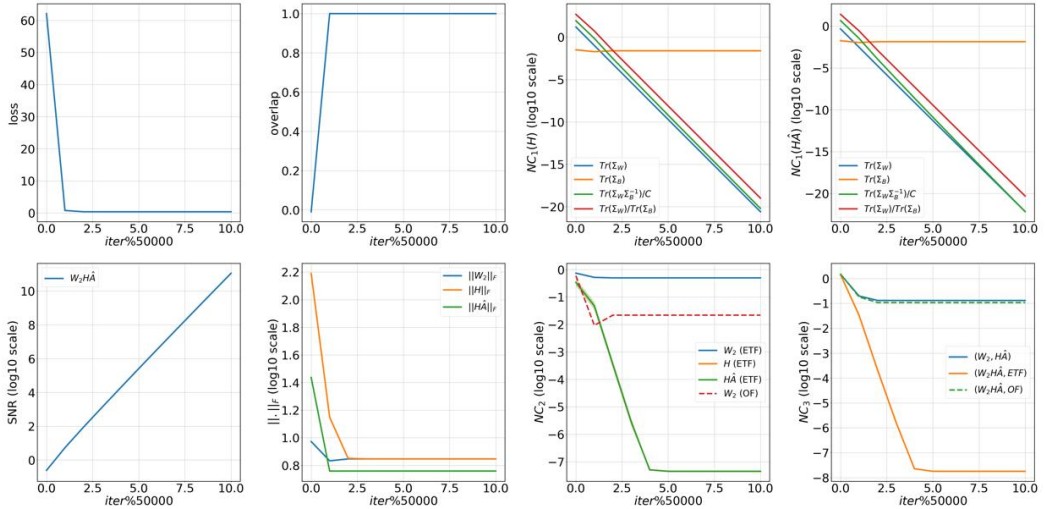

Figure 17: gUFM $\psi_\Theta^{\mathcal{F}'}$ on $\mathbf{C}^+$ variant of **D2:** $C = 4, N = 1500, p = 0.072, q = 0.0048, K = 10$. Illustration of training loss, training overlap, $\mathcal{NC}_1$ plots for $\mathbf{H}, \mathbf{H}\widehat{\mathbf{A}}$, $SNR(\mathcal{NC}_1)$ for $\mathbf{H}\widehat{\mathbf{A}}$, Frobenius norms of $\mathbf{W}_2, \mathbf{H}, \mathbf{H}\widehat{\mathbf{A}}$, $\mathcal{NC}_2$ plots for $\mathbf{W}_2, \mathbf{H}, \mathbf{H}\widehat{\mathbf{A}}$, and $\mathcal{NC}_3$ plots for $\mathbf{W}_2, \mathbf{H}\widehat{\mathbf{A}}$.

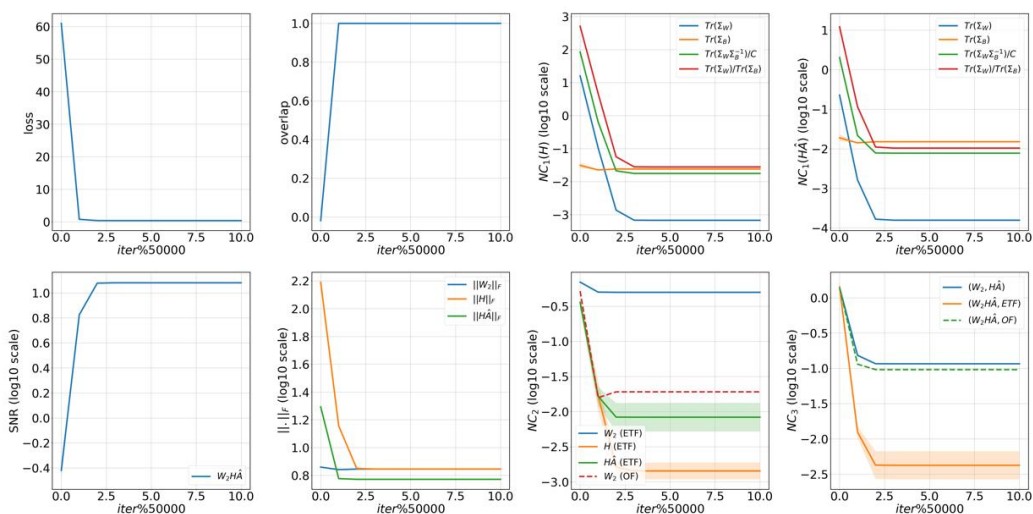

Figure 18: gUFM $\psi_\Theta^{\mathcal{F}'}$ on $\mathbf{C}^-$ variant of **D2:** $C = 4, N = 1500, p = 0.072, q = 0.0048, K = 10$. Illustration of training loss, training overlap, $\mathcal{NC}_1$ plots for $\mathbf{H}, \mathbf{H}\widehat{\mathbf{A}}$, $SNR(\mathcal{NC}_1)$ for $\mathbf{H}\widehat{\mathbf{A}}$, Frobenius norms of $\mathbf{W}_2, \mathbf{H}, \mathbf{H}\widehat{\mathbf{A}}$, $\mathcal{NC}_2$ plots for $\mathbf{W}_2, \mathbf{H}, \mathbf{H}\widehat{\mathbf{A}}$, and $\mathcal{NC}_3$ plots for $\mathbf{W}_2, \mathbf{H}\widehat{\mathbf{A}}$.

### H.3 Experiments with GNNs and Spectral Methods

The main focus of this section is to emphasize the differences between power iterations-based spectral methods and GNNs. To this end, we consider the following datasets and GNNs as follows:

**D1:** $C = 2, N = 1000, p = 0.025, q = 0.0017, K(train) = 1000, K(test) = 100$

**D2:** $C = 2, N = 1000, p = 0.0017, q = 0.025, K(train) = 1000, K(test) = 100$

• **GNNs:** In our experiments, we empirically track the NC metrics of penultimate layer features during training for both the GNN designs $\psi_\Theta^{\mathcal{F}}, \psi_\Theta^{\mathcal{F}'}$. The number of layers is varied based on $L = 64, 128$ and the hidden dimension is set to 8 across layers.

• **Spectral methods:** The number of projected power-iterations are varied based on $L = 64, 128$ for a fair comparison with GNNs.

• **Optimization:** The GNNs are trained for 8 epochs using stochastic gradient descent (SGD) with learning rate of $0.004$, momentum $0.9$ and a weight decay of $5 \times 10^{-4}$.

• **Observations:** For the dataset $\mathbf{D}1$ with homophilic graphs, we first plot the training metrics for GNNs $\psi_\Theta^{\mathcal{F}}, \psi_\Theta^{\mathcal{F}'}$ with $L = 64$ in Figures 19, 20 respectively and ensure that they reach TPT. Now, from Figures 21, 22 observe that the ratios $\mathrm{Tr}(\mathbf{\Sigma}_B(\mathbf{x}^{(l)}))/\mathrm{Tr}(\mathbf{\Sigma}_B(\mathbf{w}^{(l-1)})), \mathrm{Tr}(\mathbf{\Sigma}_W(\mathbf{x}^{(l)}))/\mathrm{Tr}(\mathbf{\Sigma}_W(\mathbf{w}^{(l-1)}))$ tend to be constant throughout the power iterations for spectral methods, whereas, the GNNs behave differently as $\mathrm{Tr}(\mathbf{\Sigma}_B(\mathbf{X}^{(l)}))/\mathrm{Tr}(\mathbf{\Sigma}_B(\mathbf{H}^{(l-1)})), \mathrm{Tr}(\mathbf{\Sigma}_W(\mathbf{X}^{(l)}))/\mathrm{Tr}(\mathbf{\Sigma}_W(\mathbf{H}^{(l-1)}))$ tend to decrease across depth. We make similar observations for GNNs $\psi_\Theta^{\mathcal{F}}, \psi_\Theta^{\mathcal{F}'}$ with $L = 128$ in Figures 23,24,25,26 and note that the behaviour is the same as $L = 32$ case illustrated in the main text. However, when considering the dataset $\mathbf{D}2$ with heterophilic graphs, even though the GNNs reach TPT during training (Figures 27,28,29,30 ), the evolution of ratios $\mathrm{Tr}(\mathbf{\Sigma}_B(\mathbf{X}^{(l)}))/\mathrm{Tr}(\mathbf{\Sigma}_B(\mathbf{H}^{(l-1)}))$, $\mathrm{Tr}(\mathbf{\Sigma}_W(\mathbf{X}^{(l)}))/\mathrm{Tr}(\mathbf{\Sigma}_W(\mathbf{H}^{(l-1)}))$ tends to differ especially for the GNN $\psi_\Theta^{\mathcal{F}'}$ when $L = 64$ (Figures 31,32) and $L = 128$ (Figures 33,34). Thus, highlighting the empirical role of depth in GNN design which requires further investigations in future efforts.

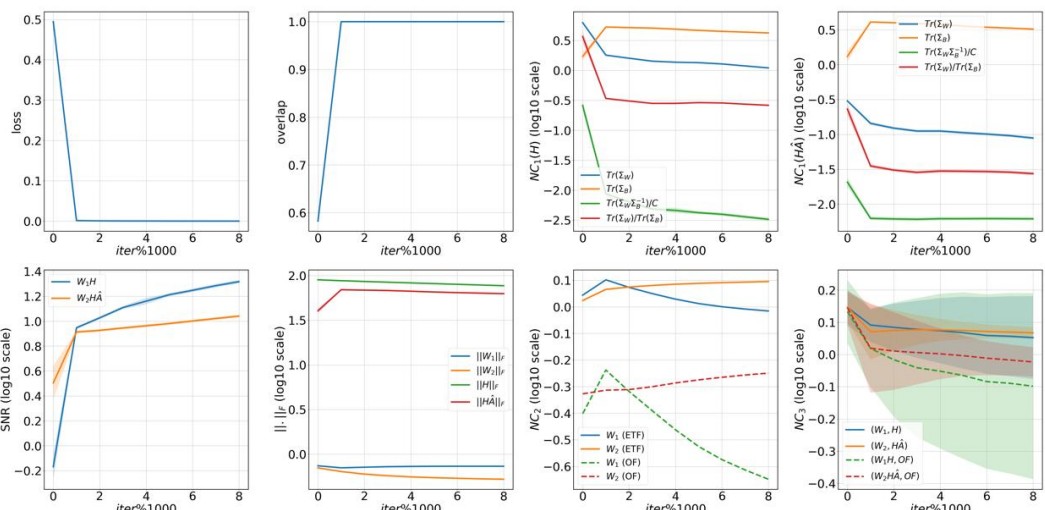

Figure 19: GNN $\psi_\Theta^{\mathcal{F}}$ with $L = 64$ on $\mathbf{D}1$: $C = 2, N = 1000, p = 0.025, q = 0.0017, K = 1000$. Illustration of training loss, training overlap, $\mathcal{NC}_1$ plots for $\mathbf{H}, \mathbf{H}\widehat{\mathbf{A}}$, $SNR(\mathcal{NC}_1)$ for $\mathbf{H}, \mathbf{H}\widehat{\mathbf{A}}$, Frobenius norms of $\mathbf{W}_1, \mathbf{W}_2, \mathbf{H}, \mathbf{H}\widehat{\mathbf{A}}, \mathcal{NC}_2$ plots for $\mathbf{W}_1, \mathbf{W}_2, \mathcal{NC}_3$ plots for $\mathbf{W}_1, \mathbf{H}$ and $\mathbf{W}_2, \mathbf{H}\widehat{\mathbf{A}}$.

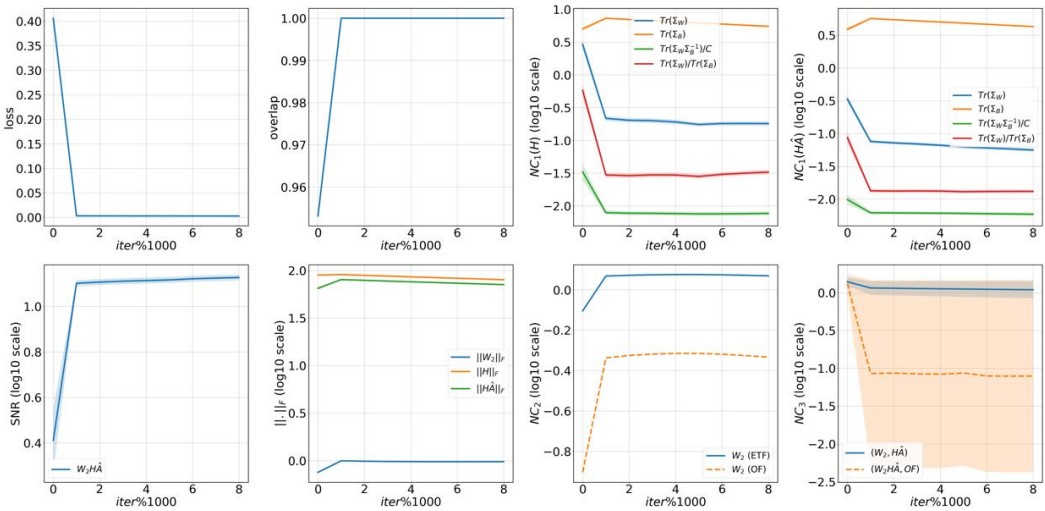

Figure 20: GNN $\psi_\Theta^{\mathcal{F}'}$ with $L = 64$ on **D1:** $C = 2, N = 1000, p = 0.025, q = 0.0017, K = 1000$. Illustration of training loss, training overlap, $\mathcal{NC}_1$ plots for $\mathbf{H}, \mathbf{H}\widehat{\mathbf{A}}$, $SNR(\mathcal{NC}_1)$ for $\mathbf{H}\widehat{\mathbf{A}}$, Frobenius norms of $\mathbf{W}_2, \mathbf{H}, \mathbf{H}\widehat{\mathbf{A}}$, $\mathcal{NC}_2$ plots for $\mathbf{W}_2$, and $\mathcal{NC}_3$ plots for $\mathbf{W}_2, \mathbf{H}\widehat{\mathbf{A}}$.

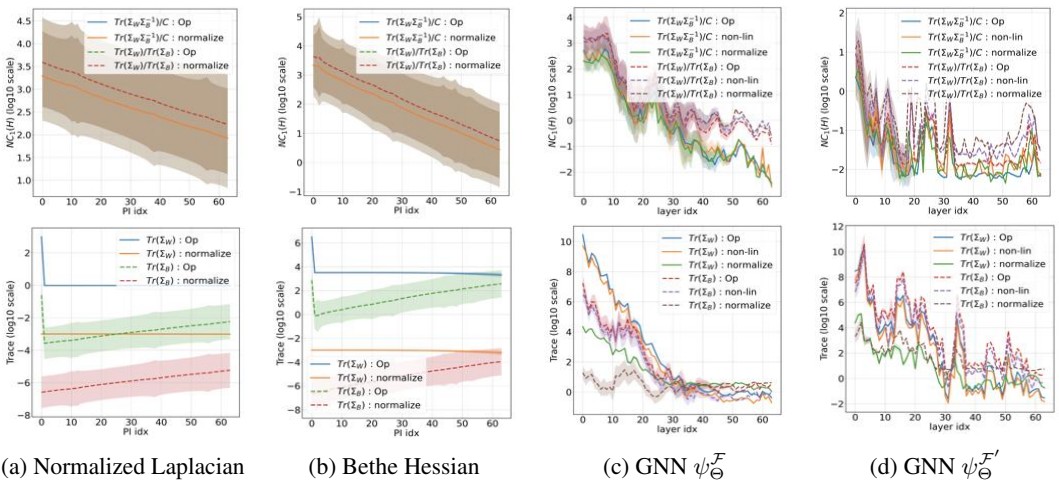

(a) Normalized Laplacian  (b) Bethe Hessian  (c) GNN $\psi_\Theta^{\mathcal{F}}$  (d) GNN $\psi_\Theta^{\mathcal{F}'}$

Figure 21: $\mathcal{NC}_1(\mathbf{H}), \widetilde{\mathcal{NC}}_1(\mathbf{H})$ metrics (top) and traces of covariance matrices (bottom) across 64 projected power iterations for NL and BH (a,b), and across 64 layers for GNNs $\psi_\Theta^{\mathcal{F}}$ and $\psi_\Theta^{\mathcal{F}'}$ (c,d) on **D1:** $C = 2, N = 1000, p = 0.025, q = 0.0017, K(test) = 100$.

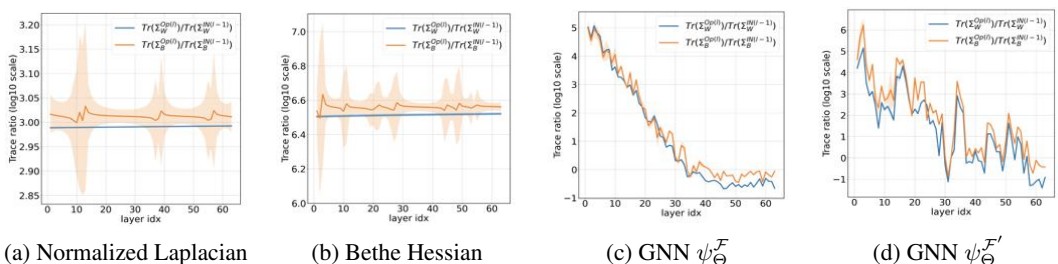

(a) Normalized Laplacian  (b) Bethe Hessian  (c) GNN $\psi_\Theta^{\mathcal{F}}$  (d) GNN $\psi_\Theta^{\mathcal{F}'}$

Figure 22: Ratio of traces of covariance matrices across 64 projected power iterations for NL and BH (a,b), and across 64 layers for GNNs $\psi_\Theta^{\mathcal{F}}$ and $\psi_\Theta^{\mathcal{F}'}$ (c,d) on **D1:** $C = 2, N = 1000, p = 0.025, q = 0.0017, K(test) = 100$.

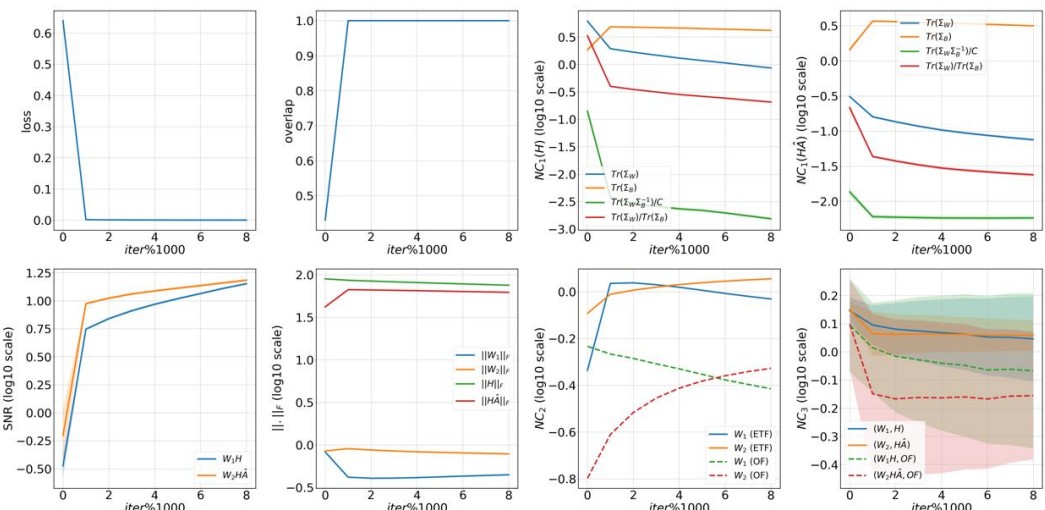

Figure 23: GNN $\psi_\Theta^{\mathcal{F}}$ with $L = 128$ on **D1:** $C = 2, N = 1000, p = 0.025, q = 0.0017, K = 1000$. Illustration of training loss, training overlap, $\mathcal{NC}_1$ plots for $\mathbf{H}, \mathbf{H}\widehat{\mathbf{A}}$, $SNR(\mathcal{NC}_1)$ for $\mathbf{H}, \mathbf{H}\widehat{\mathbf{A}}$, Frobenius norms of $\mathbf{W}_1, \mathbf{W}_2, \mathbf{H}, \mathbf{H}\widehat{\mathbf{A}}$, $\mathcal{NC}_2$ plots for $\mathbf{W}_1, \mathbf{W}_2$, $\mathcal{NC}_3$ plots for $\mathbf{W}_1, \mathbf{H}$ and $\mathbf{W}_2, \mathbf{H}\widehat{\mathbf{A}}$.

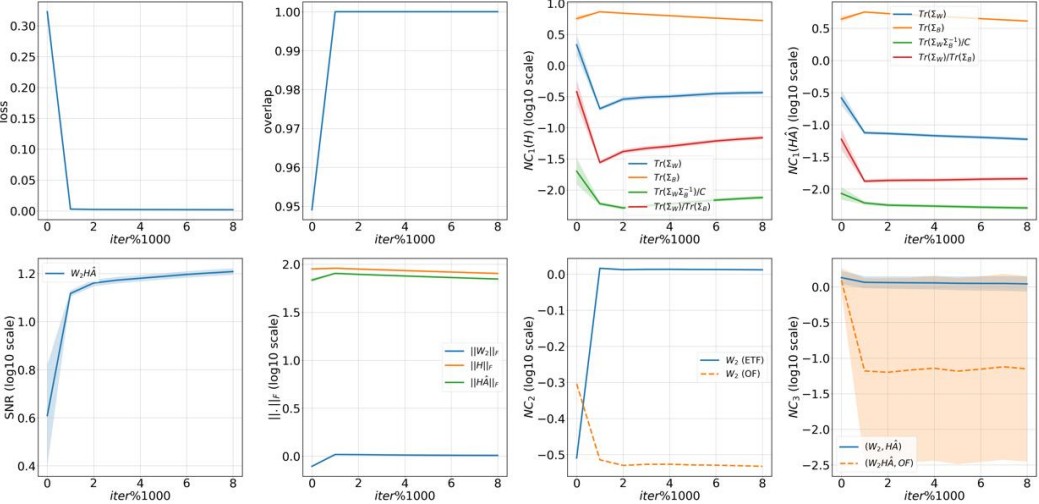

Figure 24: GNN $\psi_\Theta^{\mathcal{F}'}$ with $L = 128$ on **D1:** $C = 2, N = 1000, p = 0.025, q = 0.0017, K(train) = 1000$. Illustration of training loss, training overlap, $\mathcal{NC}_1$ plots for $\mathbf{H}, \mathbf{H}\widehat{\mathbf{A}}$, $SNR(\mathcal{NC}_1)$ for $\mathbf{H}\widehat{\mathbf{A}}$, Frobenius norms of $\mathbf{W}_2, \mathbf{H}, \mathbf{H}\widehat{\mathbf{A}}$, $\mathcal{NC}_2$ plots for $\mathbf{W}_2$, and $\mathcal{NC}_3$ plots for $\mathbf{W}_2, \mathbf{H}\widehat{\mathbf{A}}$.

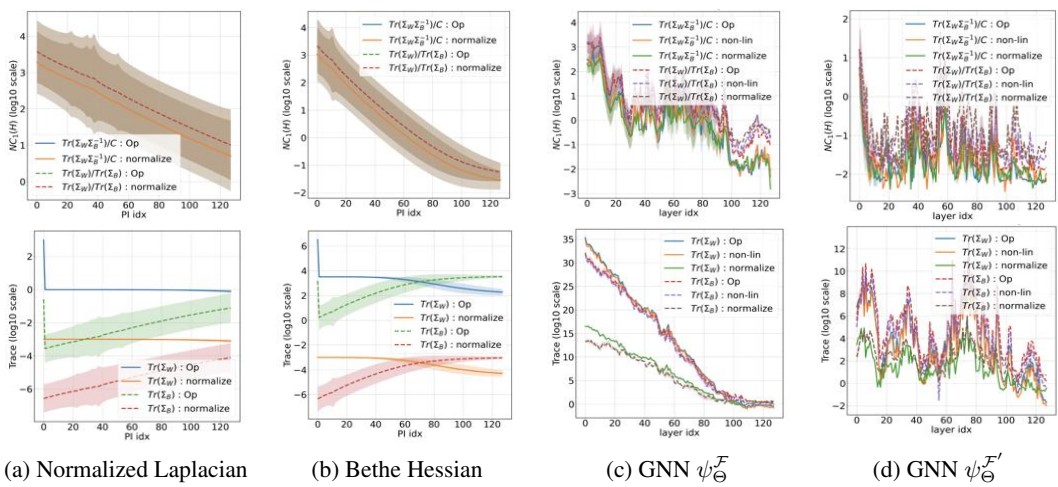

Figure 25: $\mathcal{NC}_1(\mathbf{H}), \widetilde{\mathcal{NC}}_1(\mathbf{H})$ metrics (top) and traces of covariance matrices (bottom) across 128 projected power iterations for NL and BH (a,b), and across 128 layers for GNNs $\psi_\Theta^\mathcal{F}$ and $\psi_\Theta^{\mathcal{F}'}$ (c,d) on **D1:** $C = 2, N = 1000, p = 0.025, q = 0.0017, K(test) = 100$.

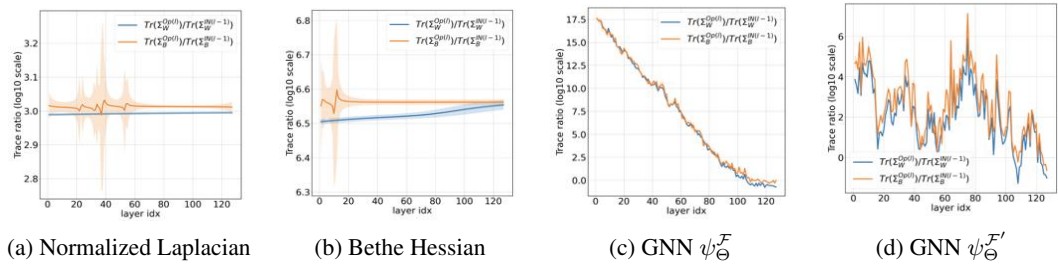

Figure 26: Ratio of traces of covariance matrices across 128 projected power iterations for NL and BH (a,b), and across 128 layers for GNNs $\psi_\Theta^\mathcal{F}$ and $\psi_\Theta^{\mathcal{F}'}$ (c,d) on **D1:** $C = 2, N = 1000, p = 0.025, q = 0.0017, K(test) = 100$.

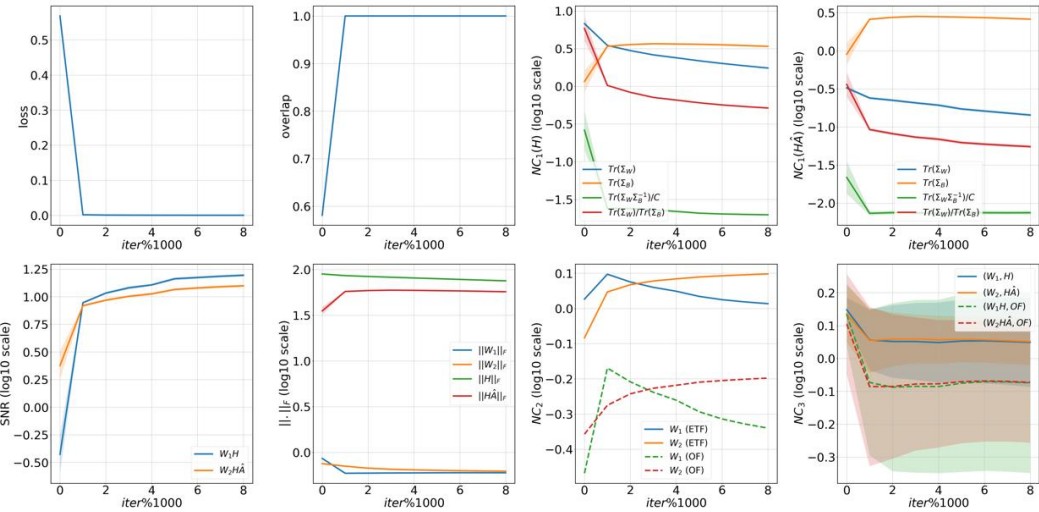

Figure 27: GNN $\psi_\Theta^\mathcal{F}$ with $L = 64$ on **D2:** $C = 2, N = 1000, p = 0.0017, q = 0.025, K = 1000$. Illustration of training loss, training overlap, $\mathcal{NC}_1$ plots for $\mathbf{H}, \mathbf{H}\widehat{\mathbf{A}}$, $SNR(\mathcal{NC}_1)$ for $\mathbf{H}, \mathbf{H}\widehat{\mathbf{A}}$, Frobenius norms of $\mathbf{W}_1, \mathbf{W}_2, \mathbf{H}, \mathbf{H}\widehat{\mathbf{A}}$, $\mathcal{NC}_2$ plots for $\mathbf{W}_1, \mathbf{W}_2$, $\mathcal{NC}_3$ plots for $\mathbf{W}_1, \mathbf{H}$ and $\mathbf{W}_2, \mathbf{H}\widehat{\mathbf{A}}$.

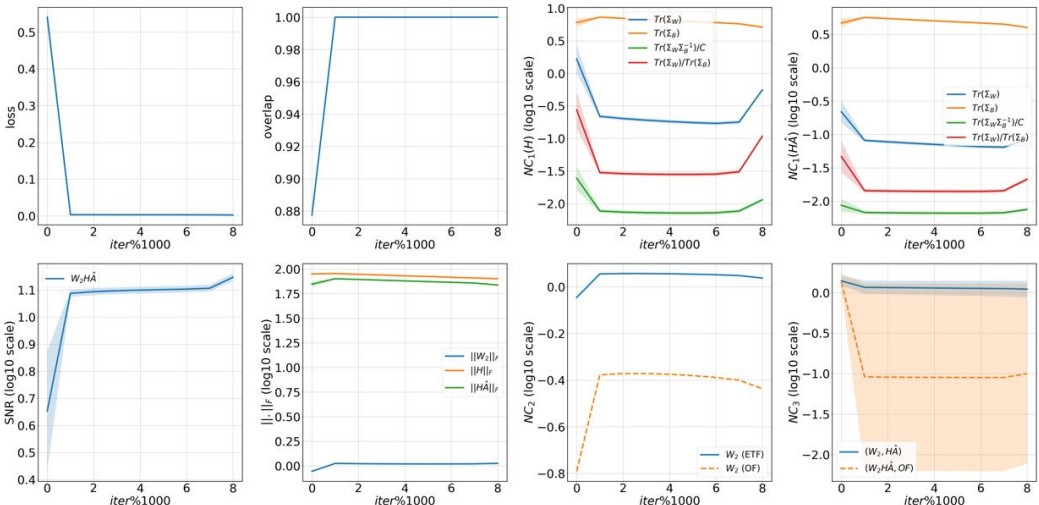

Figure 28: GNN $\psi_\Theta^{\mathcal{F}'}$ with $L = 64$ on **D2:** $C = 2, N = 1000, p = 0.0017, q = 0.025, K = 1000$. Illustration of training loss, training overlap, $\mathcal{NC}_1$ plots for $\mathbf{H}, \mathbf{H\hat{A}}$, $SNR(\mathcal{NC}_1)$ for $\mathbf{H\hat{A}}$, Frobenius norms of $\mathbf{W}_2, \mathbf{H}, \mathbf{H\hat{A}}$, $\mathcal{NC}_2$ plots for $\mathbf{W}_2$, and $\mathcal{NC}_3$ plots for $\mathbf{W}_2, \mathbf{H\hat{A}}$.

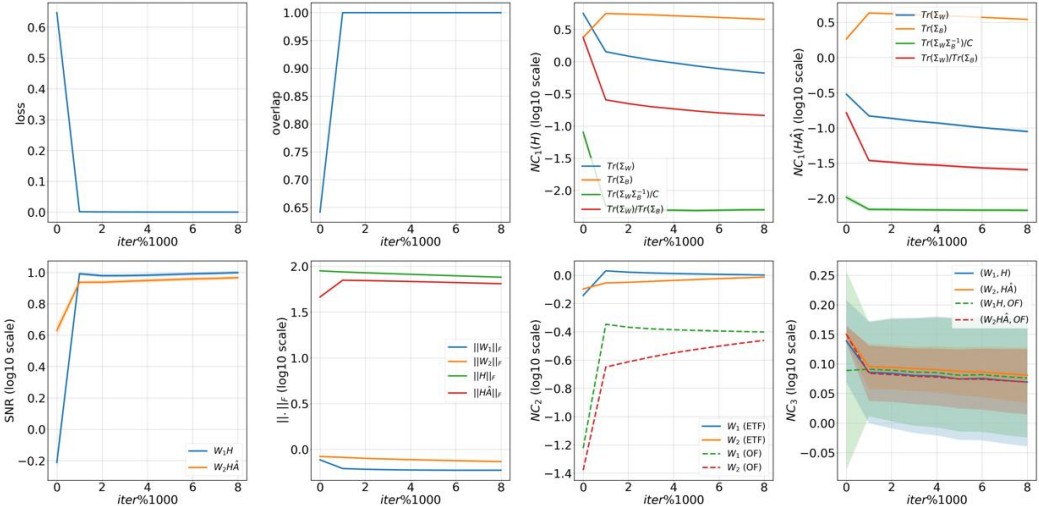

Figure 29: GNN $\psi_\Theta^{\mathcal{F}}$ with $L = 128$ on **D2:** $C = 2, N = 1000, p = 0.0017, q = 0.025, K = 1000$. Illustration of training loss, training overlap, $\mathcal{NC}_1$ plots for $\mathbf{H}, \mathbf{H\hat{A}}$, $SNR(\mathcal{NC}_1)$ for $\mathbf{H}, \mathbf{H\hat{A}}$, Frobenius norms of $\mathbf{W}_1, \mathbf{W}_2, \mathbf{H}, \mathbf{H\hat{A}}$, $\mathcal{NC}_2$ plots for $\mathbf{W}_1, \mathbf{W}_2$, $\mathcal{NC}_3$ plots for $\mathbf{W}_1, \mathbf{H}$ and $\mathbf{W}_2, \mathbf{H\hat{A}}$.

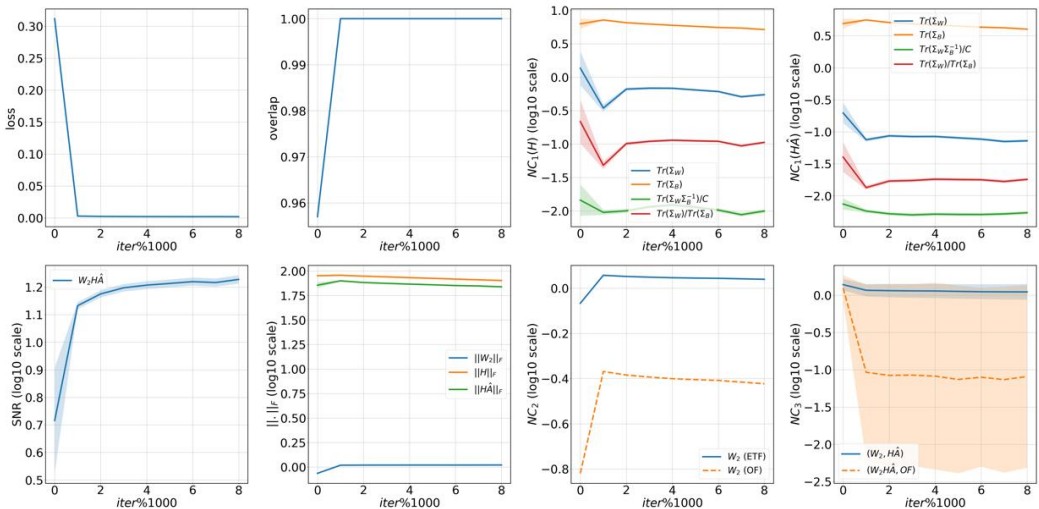

Figure 30: GNN $\psi_\Theta^{\mathcal{F}'}$ with $L = 128$ on **D2:** $C = 2, N = 1000, p = 0.0017, q = 0.025, K = 1000$. Illustration of training loss, training overlap, $\mathcal{NC}_1$ plots for $\mathbf{H}, \mathbf{H}\widehat{\mathbf{A}}$, $SNR(\mathcal{NC}_1)$ for $\mathbf{H}\widehat{\mathbf{A}}$, Frobenius norms of $\mathbf{W}_2, \mathbf{H}, \mathbf{H}\widehat{\mathbf{A}}$, $\mathcal{NC}_2$ plots for $\mathbf{W}_2$, and $\mathcal{NC}_3$ plots for $\mathbf{W}_2, \mathbf{H}\widehat{\mathbf{A}}$.

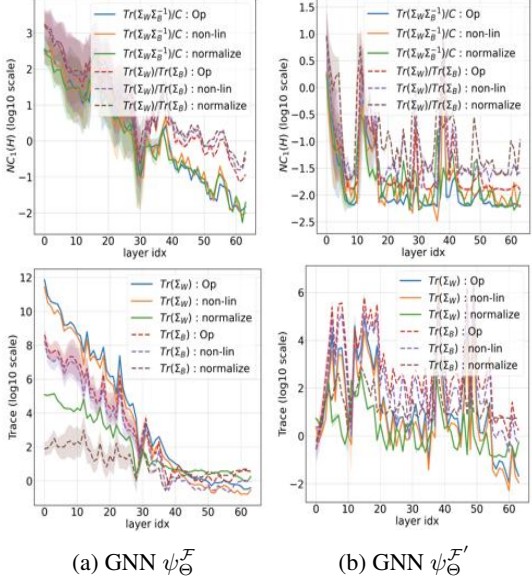

(a) GNN $\psi_\Theta^{\mathcal{F}}$      (b) GNN $\psi_\Theta^{\mathcal{F}'}$

Figure 31: $\mathcal{NC}_1(\mathbf{H}), \widetilde{\mathcal{NC}}_1(\mathbf{H})$ metrics (top) and traces of covariance matrices (bottom) across 64 layers for $\psi_\Theta^{\mathcal{F}}$ and $\psi_\Theta^{\mathcal{F}'}$ (a,b) on **D2:** $C = 2, N = 1000, p = 0.0017, q = 0.025, K(test) = 100$.

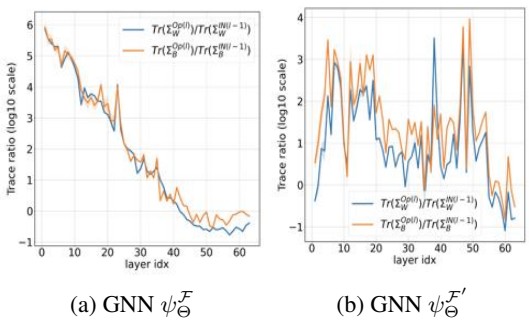

(a) GNN $\psi_\Theta^\mathcal{F}$        (b) GNN $\psi_\Theta^{\mathcal{F}'}$

Figure 32: Ratio of traces of covariance matrices across 64 layers for GNNs $\psi_\Theta^\mathcal{F}$ and $\psi_\Theta^{\mathcal{F}'}$ (a,b) on **D2:** $C = 2, N = 1000, p = 0.0017, q = 0.025, K(test) = 100$.

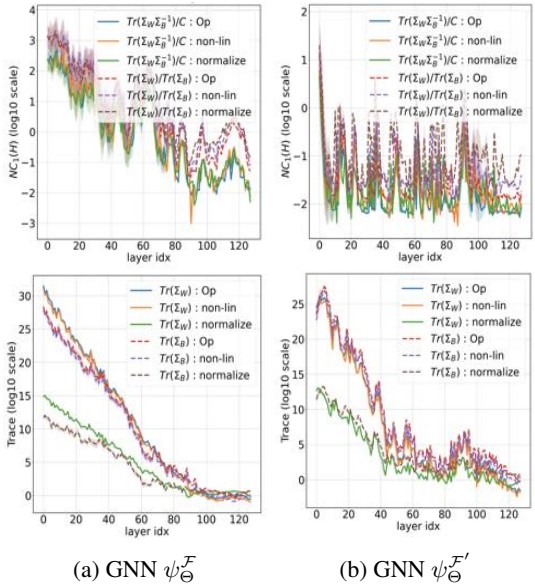

(a) GNN $\psi_\Theta^\mathcal{F}$        (b) GNN $\psi_\Theta^{\mathcal{F}'}$

Figure 33: $\mathcal{NC}_1(\mathbf{H}), \widetilde{\mathcal{NC}}_1(\mathbf{H})$ metrics (top) and traces of covariance matrices (bottom) across 128 layers for $\psi_\Theta^\mathcal{F}$ and $\psi_\Theta^{\mathcal{F}'}$ (a,b) on **D2:** $C = 2, N = 1000, p = 0.0017, q = 0.025, K(test) = 100$.

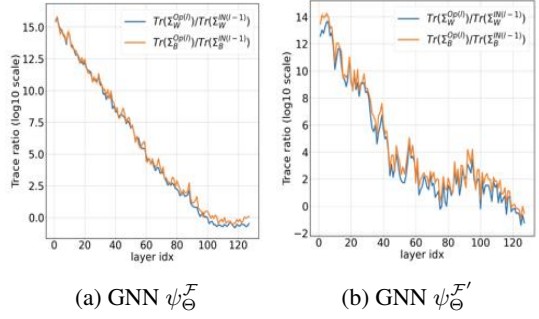

(a) GNN $\psi_\Theta^\mathcal{F}$        (b) GNN $\psi_\Theta^{\mathcal{F}'}$

Figure 34: Ratio of traces of covariance matrices across 128 layers for GNNs $\psi_\Theta^\mathcal{F}$ and $\psi_\Theta^{\mathcal{F}'}$ (a,b) on **D2:** $C = 2, N = 1000, p = 0.0017, q = 0.025, K(test) = 100$.

