# OpenReview forum: "A Neural Collapse Perspective on Feature Evolution in Graph Neural Networks"
_NeurIPS.cc/2023/Conference — NeurIPS 2023 poster_

### Official Review · Reviewer_GzMi · 2023-06-30

**Soundness:** 2 fair
**Presentation:** 2 fair
**Contribution:** 1 poor
**Rating:** 3
**Confidence:** 5

**Summary:**

The paper investigates the relationship between graph topology and feature evolution in Graph Neural Networks (GNNs). The paper starts by discussing the phenomenon of Neural Collapse (NC) in instance-wise deep classifiers, where within-class variability decreases and class means align to specific symmetric structures. The study is then extended to node-wise classification using Stochastic Block Model (SBM) graphs. An empirical study reveals a decrease in within-class variability in GNNs trained for node classification on SBMs, although not as pronounced as in the instance-wise setting. The authors propose a graph-based mathematical model to understand the influence of node neighborhood patterns and community labels on NC dynamics. The model requires strict structural conditions on the graphs to exhibit exact variability collapse, highlighting the distinction between GNNs and plain deep neural networks. Gradient dynamics analysis of the graph-based model provides theoretical explanations for the observed partial collapse in GNNs. The paper also explores the evolution of features in well-trained GNNs and compares it to spectral clustering methods. Overall, the paper investigates feature evolution in GNNs and provides insights into the impact of graph topology on this process.

**Strengths:**

1. This paper focuses on an interesting topic: Neural Collapse during the training of graph neural networks.

2. The paper concretely shows the decrease of within-class variability.

**Weaknesses:**

1. Analyzing only intra-class (within-class) variability without discussing inter-class variability is meaningless.

2. The study of over-smoothing [1] phenomena has also discussed the decrease of within-class variability. A discussion about the difference between within-class variability caused by over-smoothing and neural collapse can clarify the contribution of this work.

3. The theoretical analysis presented in this work has limited practical utility.


[1] Keriven, Nicolas. "Not too little, not too much: a theoretical analysis of graph (over) smoothing." *arXiv preprint arXiv:2205.12156* (2022).

**Questions:**

1. Can the theoretical analysis in this work lead to any practical design improvements?

2. The term "instance-wise case" (line 11) should be defined.

3. The notation of class is used in line 155 but is not defined in the 2.1 data model section.

4. A more detailed discussion should be given about the regime of exact recovery (line 94).

5. The optimizer is defined in line 147, but the loss or objective function to be optimized is not specified.

---

> ### Author Rebuttal · Authors · 2023-08-08
>
> **General comment:** We would like to thank **Reviewer GzMi** for the helpful feedback.
>
> **Q: Analyzing only intra-class (within-class) variability without discussing inter-class variability is meaningless.**
>
> **A:** We have included a thorough analysis of the increase in between-class variability of the penultimate layer features in Theorem 3.3. These results are also corroborated by experiments.
>
> **Q: The study of over-smoothing [1] phenomena has also discussed the decrease of within-class variability. A discussion about the difference between within-class variability caused by over-smoothing and neural collapse can clarify the contribution of this work.
> [1] Keriven, Nicolas. "Not too little, not too much: a theoretical analysis of graph (over) smoothing."**
>
> **A:** Over-smoothing has been widely studied in the literature to model the reduction of within-class **and** between-class feature variability across layers of a GNN during training. On the contrary, neural collapse (NC) studies the reduction in within-class but an "increase" in between-class feature variability (especially of the penultimate layer during training). Interestingly, note that when the penultimate layer features exhibit neural collapse, we can address the over-smoothing problem as neural collapse also represents the maximal separation between feature class means in addition to zero within-class feature variability.
>
> The NC analysis presented in this work covers two aspects: training and inference. In section 3, where the training phase is considered, we analyze the evolution of the penultimate layer features during training (this is a standard practice that various papers analyzing NC have followed). The significance of this approach is that we can now analyze the role of the objective function and the inherent graph structure in determining the "desirable" properties of these features to attain the global minima (Theorem 3.1). The characterization of the strict structural properties for which NC minimizers exist is a unique contribution of this work. Additionally, our gradient flow analysis for these penultimate layer features not only models the decrease in within-class variability but an increase in between-class variability as well. We present the conditions for the amount of regularization needed for this behavior in Theorem 3.3. Thus, presenting a unique perspective on feature evolution compared to previous over-smoothing analyses.
>
> Additionally, in section 4, we present a layer-wise feature evolution in a well-trained GNN. This analysis considers well-trained GNNs in inductive settings for layer-wise analysis and inherently differs from the over-smoothing analysis of [1] which considers networks in the training phase of semi-supervised settings. We will make sure to include this discussion in the main text. Thank you again for highlighting this point.
>
> **Q: The theoretical analysis presented in this work has limited practical utility.**
>
> **A:** We believe that our work can have the following practical significance:
>
> 1. The results in Theorem 3.1 on the strict structural condition of the graph and the gradient flow analysis in Theorem 3.3 indicates that contrary to the case of DNNs where MSE loss leads to collapsed minimizers, the feature evolution in GNNs during training is leading only to partial collapse (even in the most simplistic settings). This result sheds light on the impact of structural conditions on the ideal feature configurations of an expressive GNN.
>
> 2. Additionally, observe that attaining neural collapse solves the over-smoothing problem as neural collapse represents the maximal separation between feature class means in addition to zero within-class feature variability. This is indeed an important takeaway to the GNN community, which we will clarify in the revision.
>
> 3. A recent work by Ma et,al [1] showed that homophily might not be necessary for good GNN performance on node classification tasks. By leveraging Theorem 3.1 from this work, we can observe that the structural condition (dubbed condition C) required for collapsed minimizers addresses both the homophilic and heterophilic graphs. Thus, our results provide insight into these surprising empirical results [1].
>
> 4. Our work can potentially shed light on an ideal "graph-rewiring" strategy for improved GNN performance as we know the nature of the minimizers when condition C is satisfied by the computational graph.
>
> 5. We believe that extensions of our analysis to semi-supervised /self-supervised settings would be of great value to the community. For instance, the structural conditions that we analyze in this work can be used for graph augmentation in graph contrastive learning.
>
> [1] Yao Ma, Xiaorui Liu, Neil Shah, and Jiliang Tang. Is homophily a necessity for graph neural networks? In International Conference on Learning Representations (ICLR), 2022
>
> **Q: The term "instance-wise case" (line 11) should be defined.**
>
> **A:** Footnote 1 clarifies this definition. We will incorporate it into the main text for clarity. Throughout the paper, by (plain) DNNs we mean networks that output an instance-wise prediction (e.g.,image class rather than pixel class), while by GNNs we mean networks that output node-wise predictions.
>
> **Q: The notation of class is used in line 155 but is not defined in the 2.1 data model section.**
>
> **A:** It is defined in the setup section (line 74).
>
> **Q: A more detailed discussion should be given about the regime of exact recovery (line 94).**
>
> **A:** Thanks for the suggestion. We will add a discussion in the revised version.
>
> **Q: The optimizer is defined in line 147, but the loss or objective function to be optimized is not specified..**
>
> **A:** We consider the MSE loss in our analysis. It is specified in line 78, above equation 2.

---

> > ### Comment · Reviewer_GzMi · 2023-08-12
> > **Reply to Author Rebuttal**
> >
> > Thank you for the detailed explanations provided.
> >
> > Regarding my initial query, could you point me to the specific experiments that support your assertion of an "increase in between-class variability"? In line 247, you've showcased empirical results that indicate a gradual decrease in the NC1 metrics as the network depth increases. Would it be possible for you to measure the NC2 under the same experimental conditions?

---

> > > ### Author Response · Authors · 2023-08-12
> > >
> > > **We thank the reviewer for the response.**
> > >
> > > **Q: Regarding my initial query, could you point me to the specific experiments that support your assertion of an "increase in between-class variability"?**
> > >
> > > **A:** Our claim on the "increase in between-class variability" along the optimization is supported by the gUFM experiments (setup detailed in lines 201-209), whose results are illustrated in Figures 3 and 4. Our gradient flow results in Theorem 3.3 (and its proof) deal with this gUFM setting and states in item (2) the increase in between-class variability, under suitable assumptions (required for rigorous theoretical derivation). Importantly: (a) Our analysis shows a reduction in $\widetilde{NC}_1$ metric that scales the within-class variability by the between-class variability (Eq 9 in main paper); (b) The empirical and theoretical analysis of the gUFM behavior serves as a good approximation of the GNNs behavior in Figures 1 and 2 (Although in Fig 2, $Tr(\Sigma_B)$ slightly decreases, but not at the rate of $Tr(\Sigma_W)$, thus leading to a reduction in $\widetilde{NC}_1$).
> > >
> > > **Q: In line 247, you've showcased empirical results that indicate a gradual decrease in the NC1 metrics as the network depth increases. Would it be possible for you to measure the NC2 under the same experimental conditions?**
> > >
> > > **A:** Following the reviewer's suggestion, we currently conduct the proposed experiments for obtaining the version of Figures 5 and 6 that examine NC2 metric (while Figures 5 and 6 examine NC1). As a takeaway, we did not observe any significant alignments of weights, class means with ETF/OF across depth in this setting (i.e., during inference). Although these experiments do not affect the main message of the paper, we will add them to the final version for promoting future research along this direction. Thank you for the suggestion.

---

> > > > ### Author Response · Authors · 2023-08-18
> > > >
> > > > **Follow-up to our above response on the depthwise behavior of NC2 during inference:** we performed the experiments with both the GNN architectures. Contrary to the NC1 metrics, we did not observe a significant reduction of NC2 metrics across depth in this setting. Interestingly, we would like to note the this seems to be aligned with observations of previous NC efforts on plain DNNs such as [1]. Specifically, [1] performed a depthwise analysis during training of plain DNNs and did not observe a reduction of NC2 across depth that is of the level of the reduction in NC1. Moreover, they used also an "optimistic" 2-levels UFM with freely optimized features and ReLU nonlinearity (as there is nonlinearity between different layers in practice) and showed that the NC2 structure occurs only in the deepest features of this optimistic model (e.g., see Fig. 4 and Fig. 9 in [1], third image in each row). Thus, highlighting the fact that the analysis of NC2 with GNNs is much more complicated and an interesting future research direction.
> > > >
> > > > We thank Reviewer GzMi again for the discussion and for handling our paper. We hope that our point-to-point response to the comments in these posts (including the ones posted above, which highlight the directions for practical applications of our theoretical results) makes them appreciate the contribution and reconsider the score.
> > > >
> > > > [1] Tirer, Tom, et,al. "Extended unconstrained features model for exploring deep neural collapse." ICML, 2022.

---

> > > > > ### Comment · Reviewer_GzMi · 2023-08-20
> > > > >
> > > > > Dear Authors,
> > > > >
> > > > > Thank you for your thoughtful response.
> > > > >
> > > > > 1. The need to ascertain evidence regarding the existence of NC2 in graph contexts is quite crucial. Your statement, "we currently conduct the proposed experiments for obtaining the version of Figures 5 and 6 that examine the NC2 metric (while Figures 5 and 6 examine NC1). As a takeaway, we did not observe any significant alignments of weights," is not convincing enough. A prominent motivation of your work revolves around addressing the over-smoothing issue, which you've rightfully highlighted in the manuscript and in your replies. However, if this is an important motivation, it would be imperative to showcase empirical data on NC2, even if on a smaller real-world graph. It raises concerns when extensive measurements are presented for NC1, yet NC2 seems overlooked.
> > > > >
> > > > > 2. For the propposed task graph rewiring, I think some holdout nodes are necessary. Subsequently, the graph could be rewired to optimize som NC metrics associated with these nodes. This process accentuates the need to delve deeper into the generalizability of both NC1 and NC2. If you believe that graph rewiring bolsters the significance of your work, extra detailed discussion on the generalization of NC1 and NC2 (train to test) becomes indispensable.
> > > > >
> > > > > 3. For the data modeling,  the approach to data modeling is acceptable, acknowledging that certain assumptions and simplifications are inevitable. However, a critical observation is that some interesting conclusions drawn from NC hinge on the presence of balanced communities in graphs. This premise is arguably unrealistic in real-world settings, which make these conclusions less applicable in practical graph domains.
> > > > >
> > > > > 4. The presentation can be significantly refined. The derivations in the appendix would benefit from a clearer narrative. Addtioanlly, introducing a proof sketch within the primary content can greatly aid reader comprehension and accessibility.
> > > > >
> > > > > In conclusion, having revisited the manuscript, the associated discussions, and the appendix, I believe the current version doesn't yet meet the standards of NeurIPS. I've decided to retain my rating and have adjusted the confidence score to mirror my current assessment.

---

> > > > > > ### Author Response · Authors · 2023-08-20
> > > > > >
> > > > > > Dear Reviewer GzMi,
> > > > > >
> > > > > > Thank you for the response.
> > > > > >
> > > > > > **Q: Regarding the first comment on NC2.**
> > > > > >
> > > > > > **A:** The NC1 component of the NC phenomenon is widely considered as the most important component, as it implies a reduction of the within-class variability of the features around their mean. All the other components (e.g., NC2) refer to the structure of the means of the different classes and are not very informative if the variability around the means is large — namely, if NC1 is large. Therefore, many significant NC papers focus specifically on NC1 [1,2,3,4,5]. Therefore, the reviewer’s claim:
> > > > > > "The need to ascertain evidence regarding the existence of NC2 in graph contexts is quite crucial" is
> > > > > > unjustified, especially as the main message of the paper is that NC1 (the most important component)
> > > > > > cannot be reduced by GNNs as in plain DNNs.
> > > > > >
> > > > > > Nevertheless, we do not ignore NC2 metrics in our paper. we have presented extensive empirical analysis on the NC2 metrics in Appendix F (for the penultimate layer of GNNs in the training phase and also for gUFM’s). In particular, we show an interesting connection between the reduction in NC2 metrics along optimization and the graph satisfying condition C. Yet, again, this component, as
> > > > > > defined in [6] is not as significant as NC1, especially as it was proposed only for plain DNNs (with no graph topology) to measure the distance from a specific structure (simplex ETF/OF) that is unlikely to be fully suitable for graph data.
> > > > > >
> > > > > > Furthermore, the answer cited by the reviewer was given to their specific request: "In line 247, you've showcased empirical results that indicate a gradual decrease in the NC1 metrics as the network depth increases. Would it be possible for you to measure the NC2 under the same experimental conditions?" (i.e., the settings of Figs. 5 and 6). Namely, we answered that we do not see a depthwise gradual decrease in NC2 metrics, which is perfectly aligned with previous results on plain DNNs (for which these metrics were proposed) as explained above. Therefore, the concern of the reviewer is unclear.
> > > > > >
> > > > > > Lastly, note that NC2 has little to do with over-smoothing. Over-smoothing refers to the loss of feature variability within and between classes, while NC2 focuses on a specific structure of the mean features of different classes and has nothing to do with variability. On the other hand, NC1, which is indeed given focus in our paper, has direct implications on over-smoothing (as explained in the
> > > > > > discussion, reaching low NC1 metrics implies no over-smoothing).
> > > > > >
> > > > > > To conclude, we respectfully disagree with the reviewer about the criticality of the NC2 metrics in the context of our paper.
> > > > > >
> > > > > > [1] Hangfeng He et,al. A law of data separation in deep learning. PNAS 2023
> > > > > >
> > > > > > [2] Tomer Galanti, et,al. On the role of neural collapse in transfer learning, ICLR 2022
> > > > > >
> > > > > > [3] Tomer Galanti, et,al. Improved generalization bounds for transfer learning via neural collapse, ICML 2022 Pre-training Workshop.
> > > > > >
> > > > > > [4] Yang, Yongyi et,al. "Are Neurons Actually Collapsed? On the Fine-Grained Structure in Neural Representations." ICML 2023
> > > > > >
> > > > > > [5] Tom Tirer et,al. Perturbation analysis of neural collapse, ICML 2023
> > > > > >
> > > > > > [6] Papyan, Vardan, et,al "Prevalence of neural collapse during the terminal phase of deep learning training." PNAS 2020
> > > > > >
> > > > > > **Q:  Regarding the second comment on graph rewiring.**
> > > > > >
> > > > > > **A:** Indeed, we believe that condition C can be utilized for devising useful graph rewiring strategies, as it is aligned with empirical observations on graphs for which GNNs perform well (i.e., generalize better) [1]. It is fair to leave the study of such an application for future in-depth research.
> > > > > >
> > > > > > [1] Yao Ma, et al. Is homophily a necessity for graph neural networks? (ICLR), 2022
> > > > > >
> > > > > > **Q:  Regarding the third comment on balanced communities in the data model.**
> > > > > >
> > > > > > **A:** Assuming balanced communities allows us to leverage the properties of Kronecker products in our analysis, without which the analysis would become much more complicated than it already is. Such an assumption is common in the literature on SBM-based analysis as also followed by previous works [1]. Also, observe that when the balanced communities assumption holds true, we show in Theorem 3.1 that the graph needs to obey a strict structural condition for collapsed minimizers. Thus, highlighting that even if one were to obtain an ideal real-world graph with balanced communities, $\hat{A}$ must satisfy condition C. We believe this is a valuable insight when one attempts to train a GNN by rewiring a real-world graph or when using graph augmentation techniques.
> > > > > >
> > > > > > [1] Keriven, Nicolas. "Not too little, not too much: a theoretical analysis of graph (over) smoothing." NeurIPS (2022)
> > > > > >
> > > > > > **Q: Regarding the fourth comment on presentation and proof sketch.**
> > > > > >
> > > > > > **A:** We take note of this point and will improve the readability by leveraging the extra page in the revision and adding more details about the proof sketch.

---

### Official Review · Reviewer_pASb · 2023-07-01

**Soundness:** 4 excellent
**Presentation:** 3 good
**Contribution:** 3 good
**Rating:** 7
**Confidence:** 4

**Summary:**

This paper investigates the feature evolution in Graph Neural Networks (GNNs) via the lens of Neural Collapse. They conduct an empirical study that reveals a decrease in within-class variability in the deepest features of GNNs, but not to the extent observed in instance-wise classification settings. By proposing and analyzing a graph-based Unconstrained Features Model (UFM), the authors show that a strict structural condition on the graphs is necessary for exact variability collapse, which is a rare event. They provide theoretical reasoning for the partial collapse observed during GNNs training and study the evolution of features across layers in well-trained GNNs, comparing the decrease in NC metrics with power iterations in spectral clustering methods.

**Strengths:**

This paper presents a pioneering study on Neural Collapse (NC) in Graph Convolutional Networks (GCNs) and the forward process of a neural network, which has the potential to inspire future work. The authors provide unique and valuable theoretical results, such as the partial collapse in Theorem 3.1. Additionally, the paper is well-written, the methodology is clearly explained, and the proposed approaches appear to be solid.

**Weaknesses:**

This paper has a few areas that could be improved upon.

1. Some experimental findings could benefit from stronger support through theoretical results (Please refer to Questions 1-3).

2. The applications and implications of certain theorems could be further elaborated (Please refer to Questions 4-5).

**Questions:**

To enhance the manuscript, the authors may consider addressing the following points:

1. In addition to the collapse of variability (NC1), the preference towards a simplex ETF (NC2) is an important observation in the original NC paper [34]. From Figure 1(c), it seems that some variant of NC2 might also hold in GCNs. To provide a more comprehensive understanding of NC, it would be helpful if the authors could include a theorem or at least a discussion on NC2.

2. While Theorem 3.1 and Theorem 3.2 demonstrate that the within-class variability is non-zero with high probability, the authors have not provided a lower bound for the variability concerning p and q. Analyzing this lower bound could strengthen the paper's contribution. For example, the authors might consider analyzing the expected variability under SSBM.

3. Theorem 3.3 appears to be somewhat weaker compared to existing studies on NC (e.g., [54]). It would be beneficial to provide a local or global convergence.

4. In Lines 270 - 276, the authors show that the ratios in GCN behave differently compared to those in power iterations (i.e., simplified graph convolutional networks). It would be insightful if the authors could explain how this difference benefits GCNs. Additionally, the evolutions in $\mathcal{F}$ and $\mathcal{F}'$ are different, and understanding how this difference impacts the performance of $\mathcal{F}$ and $\mathcal{F}'$ would be valuable.

5. Relating Theorem 4.1 to classic results in GNNs (e.g., over-smoothing) could provide further context and strengthen the paper's overall contribution to the field.

**Limitations:**

Yes

---

> ### Author Rebuttal · Authors · 2023-08-08
>
> **General comment:** We are grateful to **Reviewer pASb** for an encouraging review and for raising interesting questions.
>
> **Q:** Regarding theoretical results and a discussion on NC2.
>
> **A:** We agree with the reviewer's point that NC2 might also hold to some extent for GNNs. We show these metrics in Appendix F. In those experiments, NC2 metrics indicate that the (centered) class-mean features do not align significantly with a simplex ETF. This is due to $\hat{A}$ in the risk formulation, for which, our understanding is not theoretically rigorous yet. We attempted to analyze the risk via a matrix factorization approach as followed by previous efforts [1]. The SVD approach on $W_2H\hat{A}$ along the "central path"[2, 1] is complicated by the presence of $V^T_HU_{\hat{A}}$, where $V_H$ represents the right-singular basis for $H$ and $U_{\hat{A}}$ represents the left-singular basis for $\hat{A}$. These bases are not guaranteed to align (at least the assumption of $V^T_HU_{\hat{A}}$ is diagonal seemed too strong). Without it, the analysis is extremely difficult to obtain the properties of the singular values (s.v) of $H$ w.r.t those of $\hat{A}$. Additionally, in the non-GNN NC literature, there are efforts that fix the last layer weights to a simplex ETF and observe only a minimal drop in DNN performance. Sometimes, this also helps in addressing class-imbalance issues.
>
> **Q: Regarding the lower bound for the variability w.r.t p and q.**
>
> **A:** In Theorem 3.1 we obtained the conditions on the graph structure for which NC solutions are the minimizers of the risk. Also, since the expected adjacency matrix $\mathbb{E}\hat{A}$ satisfies this condition, neural collapse is desirable. However, to lower-bound the variability (which pertains to the partial collapse result that we see in the experiments), we believe our gradient flow analysis in Theorem 3.3 can serve as a starting point. Through this analysis, we obtain the rate of change of trace of within and between class variability along the flow  (Appendix D.4). These rates depend on the perturbation from the expected SSBM structure. Thus, the optimal within-class/between-class variability depends on the perturbation (which in turn depends on $p,q$). The challenge with obtaining an exact lower-bound lies in calculating the derivatives of terms such as  $HEE^\top H^\top$ (notation as per proof of Theorem 3.3), which turned out to be more complicated than expected. An alternative approach can involve a spectral analysis of $E$ in terms of $p,q$. However, we believe it requires a separate study on its own.
>
> **Q: Theorem 3.3 appears to be somewhat weaker compared to existing studies on NC (e.g., [54]). It would be beneficial to provide a local or global convergence.**
>
> **A:** The work of [54] (ref in paper) analyzed the loss landscape and convergence when the risk is based on cross-entropy loss. However, they do not analyze the gradient dynamics. Other NC works that analyze gradient dynamics [15,40] (ref in paper) consider only a plain (non-graph) UFM, which is much simpler to analyze (and the gradient flow is ensured to reach an NC minimizer, which is not the case with gUFM). In Theorem 3.3 we analyze NC1 along the gradient dynamics of a gUFM which is much more involved as the flow is not guaranteed to reach an NC minimizer.
>
> **Q: Regarding the different trace ratios for spectral methods and GNNs.**
>
> **A:** Through the analysis in section 4, we wish to characterize the nature of the learnable weights that allow a GNN to perform "better" than spectral methods. In this context, "better" indicates that the GNN requires less number of layers than the number of projected power iterations by spectral methods for reduction in NC1 (and to achieve good inference results). In Appendix.F, we show additional experiments with depths of 64, 128. The takeaway from our observations is that the weights $W_1, W_2$ of the GNNs are playing a major role in projecting the features and aiding in faster reduction rates of NC1. Since spectral methods do not comprise any learnable projections, the benefits of GNNs are evident in this scenario.
>
> Regarding the GNNs $\mathcal{F}, \mathcal{F}'$, we have not observed any significant difference in their performance that may be caused due to varying rates of reduction in NC1. It seems like the presence of $W_1$ is delaying the NC1 reduction, but both networks reach approximately the same level of NC1 at the last layer. From a practical viewpoint, one can look at GNN layer pruning based on NC1 reduction rates for efficient inference.
>
> **Q: Regarding section 4 (Theorem 4.1) and the connection with over-smoothing**
>
> **A:** Indeed, over-smoothing has been widely studied in the literature to model the reduction of within-class and between-class feature variability across layers of a GNN during training. On the contrary, NC studies the reduction in within-class and an "increase" in between-class feature variability (especially of the penultimate layer during training). Interestingly, when the penultimate layer features exhibit NC, we can say that the over-smoothing problem is resolved. This is indeed an important takeaway to the GNN community, which we will clarify in the revision.
>
> However, our analysis in section 4 considers well-trained GNNs in inductive settings for layer-wise analysis during **inference**. This setup inherently differs from the over-smoothing analysis (for ex: [3]) which considers the training phase in semi-supervised settings. Interestingly, even during inference, we found variability reduction patterns (fig 5) that resemble partial over-smoothing observed by [3].
>
> [1] Tirer, Tom, et,al. "Extended unconstrained features model for exploring deep neural collapse." ICML, 2022.
>
> [2] Han, X. Y., et.al. "Neural Collapse Under MSE Loss: Proximity to and Dynamics on the Central Path." ICLR. 2021
>
> [3] Keriven, Nicolas. "Not too little, not too much: a theoretical analysis of graph (over) smoothing." NeurIPS 2022.

---

### Official Review · Reviewer_t9NA · 2023-07-04

**Soundness:** 3 good
**Presentation:** 3 good
**Contribution:** 2 fair
**Rating:** 5
**Confidence:** 4

**Summary:**

This work tries to investigate the feature evolution in GNNs in inductive setting. In particular, it focuses on the Neural Collapse problem using the SSBM data model to ensure the existence of this phenomenon. Moreover, it proposes to verify that the extent of NC in GNNs is not as severe as in instance-wise classifications.

**Strengths:**

1. It is interesting to study the feature evolution with GNNs and especially the correlation between the topological information.
2. The data model is clearly presented and the experiments are close to the theoretical results.

**Weaknesses:**

1. The study of feature evolution with GNNs and especially the correlation between the topological information is a great topic, however, the data model and the concentration on the NC condition slightly retreat the original claim into a theoretically friendly but less practical direction.
2. Most concerns are raised about the practical value of this study, which lies in two aspects:
    1. the motivation of this work needs to be further clarified, e.g. why the NC phenomenon is important, what about other stages of trait development.
    2. data model: SSBM is a good model to study the topological features.

My concerns are mainly about the practical significance of this paper. If I have missed anything important in the proofs and arguments, please let me know.

**Questions:**

1. Major:
    1. A more comprehensive motivation for studying feature evolution in GNNs is highly recommended. And there is another smoothing problem: If the paper focuses on the NC view, the pre-convergence stage is completely missing. Since for most of the case or GNN practitioners are doing with pre-convergence stage, which should be more worthy to study. It is just to make sure that the problem and the perspective are really practical.
    2. For the inductive setting, which this paper concentrates on, it is not necessary that the test nodes are in a different graph than the training nodes. It only requires that the test nodes do not appear in the training phase.
    3. The experiments are performed only on the graphs simulated by the SSBM assumptions.
2. Minor:
    1. Line 53-54: Essentially, we highlight the main differences in the analysis of NC in GNNs by identifying structural conditions on the graphs under which the global minimizers of the training objective exhibit full NC1. Hard to understand, you can split it - since it is one of the main findings of the paper, which needs to be clear enough.
    2. Line 79-80, the MLE loss is more and more popular to train DNN classifiers might be misleading somehow. However, MLE is a well-known alternative for more complex loss functions in machine learning theories.

**Limitations:**

Limitations are not included by the authors. As I presented questions above, my concerns are mainly about the practical significance of this paper.

---

> ### Author Rebuttal · Authors · 2023-08-08
>
> **General note:** We would like to thank **Reviewer t9NA** for the constructive feedback. The 2 minor issues will be fixed in the revision.
>
> **The importance of neural collapse (NC).** In standard DNN settings (e.g., image classification on MNIST), the classifiers tend to exhibit NC once they perfectly classify the training data (i,e during TPT). NC includes both: "reduction" in the features’ within-class variability
> and "increase" (or stabilization) in the between-class variability, such that the features form certain low-dimensional geometric structures. This phenomenon has allowed the community to understand the benign effects of overparameterization and the benefits of training beyond the "zero-classification-error" stage for DNNs (e.g., cases with better test accuracy and improved adversarial robustness [1]). Importantly, we demonstrate that the case with GNNs is not similar to any previous work on NC as the structural constraints of graphs tend to hinder the ability of NC. Also, since the over-smoothing problem is common in deep GNNs, NC shows potential to address such key issues and have a promising impact.
>
> **The data and network models.** To explore the level of NC in GNNs in a principled fashion, we start with SSBM graphs, as their properties are well-suited for theoretical GNN analysis [2]. The gUFM model presented in this paper is much more complicated than any theoretical model in previous publications on NC [4] as we retain the graph topology in the analysis. Importantly, even for our “optimistic" model, we show that the desired configuration of the penultimate layer features (in terms of optimization optimality) does not exhibit exact NC unless the graph satisfies a strict structural condition. This immediately implies an inherent difference between practical GNNs and DNNs. Also, note that it seems very difficult to derive the exact training dynamics in GNNs without trivial assumptions on the structure. Thus, we leverage the literature on SSBM graphs and present a rigorous gradient flow analysis to explain the partial collapse observed in our experiments. This analysis is already a major step beyond analyzing only the minimizers of the model.
>
> **Practical significance:**
>
> 1. Observe that attaining NC solves the over-smoothing problem as NC represents the maximal separation between feature class means in addition to zero within-class feature variability.
>
> 2. Recently, Ma et,al [3] showed that homophily might not be necessary for good GNN performance on node classification tasks. By leveraging Theorem 3.1 from our work, we can observe that the structural condition required for collapsed minimizers addresses both the homophilic and heterophilic graphs. Thus, providing insights into Ma's surprising empirical results.
>
> 3. Our work can potentially shed light on an ideal "graph-rewiring" strategy as we know the nature of the minimizers when condition C is satisfied by the computational graph.
>
> **Q: Regarding feature evolution and pre-convergence phases of training.**
>
> **A:**  To address the reviewer's comment, we would like to emphasize that the contribution of our analysis is to understand why GNNs are unable to reach the convergence (TPT) phase. Our result in Theorem 3.1 highlights that, when using the MSE loss, even though DNNs exhibit NC minimizers [4], in the case of GNNs, the graphs must satisfy a strict structural condition for collapsed minimizers.
>
> Regarding the pre-convergence phase, we have provided a detailed analysis of the reduction in NC1 along the gradient flow in Theorem 3.3. As a practical takeaway, we have provided bounds on the regularization (Theorem 3.3 proof) that is needed for the trace of between-class feature covariance to increase along the flow. Overall, the reason for initiating this line of work through a theoretical approach is to present optimistic settings in which GNNs fail to exhibit NC and promote further study in the community.
>
> **Q: Regarding test nodes in the inductive setting.**
>
> **A:** We have followed the setup of Chen et, al [5] which has been widely adopted for community detection in supervised settings.
>
> **Q: Regarding experiments only on SSBM graphs**
>
> **A:** Attaining TPT state is not so simple for GNNs on real-world graphs as indicated by a lack of rigorous benchmarks in the community on supervised community detection (in inductive settings). We have performed experiments on the real-world graphs as per the strategy of [5] as follows:
>
> **Dataset.** We consider the "com-amazon" graph from the SNAP collection and prepare 1000 training graphs and 100 test graphs, with each graph having 2 non-overlapping communities from the top 5,000 communities provided by the SNAP collection.
>
> **Baselines and results**: Although the graphs obtained in this fashion are not of the same size and have imbalanced communities, we did observe a partial collapse even in the pre-convergence phase (however, not to the extent observed in the SSBM case as the datasets are relatively complex to attain TPT).
>
> **We thank the reviewer for the comments and will include these key discussions in the revision.**
>
> [1] Papyan, Vardan, et,al "Prevalence of neural collapse during the terminal phase of deep learning training." PNAS 2020
>
> [2] Keriven, Nicolas. "Not too little, not too much: a theoretical analysis of graph (over) smoothing." NeurIPS (2022)
>
> [3] Yao Ma, et al. Is homophily a necessity for graph neural networks?  (ICLR), 2022
>
> [4] Han, X. Y., et.al. "Neural Collapse Under MSE Loss: Proximity to and Dynamics on the Central Path." ICLR. 2021
>
> [5] Chen, Zhengdao et al. "Supervised Community Detection with Line Graph Neural Networks." ICLR 2019.

---

> > ### Comment · Reviewer_t9NA · 2023-08-14
> > **Response to the authors' rebuttal**
> >
> > I thank the authors for their detailed responses to my concerns. Especially the answers regarding "**Regarding feature evolution and pre-convergence phases of training.**" and the data model greatly improve my understanding. I would like to ask three more questions:
> >
> > 1. Talking about over-smoothing phenomenon, over-smoothing should not be a relative issue to NC, since it is not only "(NC1) The within-class variability of the deepest features decreases", but to a global extent.
> > 2. Still, from a practical point of view, the rewriting point should be included in the discussion of this paper, e.g. a section after the experiments, to help build more reasonableness and soundness of this work. I fully understand that your work is from an analytical point of view.
> > 3. Finally, the sentence "in the case of GNNs, the graphs must satisfy a strict structural condition for collapsed minimizers" gives a good explanation, and this should also be made clear after the theoretical analysis (SSBM and UFM), especially by introducing a quantified variable that controls the randomness in real graphs, and how significant it is by doing empirical measurement. To this end, the understanding should be more smoothing. This may be difficult to achieve in this rebuttal stage, but I would recommend doing so in a later revision.
> >
> > Overall, the authors have addressed most of my concerns well and I would like to raise my score to 5. I ask the authors to be patient with my concerns and suggestions above.

---

> > > ### Author Response · Authors · 2023-08-15
> > >
> > > **We greatly appreciate the reviewer's response.**
> > >
> > > **Q: Talking about over-smoothing phenomenon, over-smoothing should not be a relative issue to NC, since it is not only "(NC1) The within-class variability of the deepest features decreases", but to a global extent.**
> > >
> > > **A:** Thanks for the comment. If we understand the comment correctly, by "global extent" do you mean that oversmoothing also pertains to a reduction of between-class variability, in addition to a reduction in within-class variability? If yes, then note that, as oversmoothing typically pertains to a reduction in between-class and within-class variability "across layers during training", the deepest features tend to be affected the most. However, with an NC analysis of the penultimate layer features (deepest features), we identify the conditions (pertaining to graph structure and regularization) under which the within-class variability "decreases" and between-class variability "increases" for the deepest features during training. Thus, potentially mitigating the effects of over-smoothing.
> > >
> > > **Q: Still, from a practical point of view, the rewriting point should be included in the discussion of this paper, e.g. a section after the experiments, to help build more reasonableness and soundness of this work.**
> > >
> > > **A:** We will add a brief discussion on the graph rewiring aspects in the revised version. Thank you for the suggestion.
> > >
> > > **Q: Finally, the sentence "in the case of GNNs, the graphs must satisfy a strict structural condition for collapsed minimizers" gives a good explanation, and this should also be made clear after the theoretical analysis (SSBM and UFM), especially by introducing a quantified variable that controls the randomness in real graphs, and how significant it is by doing empirical measurement. To this end, the understanding should be more smoothing.**
> > >
> > > **A:** We added a statement in lines (180-181) before Theorem 3.1 which conveys this message.
> > > And following the reviewer’s suggestion, we will add a statement after the theoretical results as well
> > > to effectively convey the implications.
> > >
> > > Regarding the suggested experiments: If we understand it correctly, the reviewer suggests to introduce
> > > a variable that controls the randomness of adding edges between nodes in real graphs (similar to p,q
> > > in SSBM graphs), and checking if being close to the proposed condition leads to more collapse. This
> > > is a very interesting direction for future research and we will do our best in discussing it in the revised
> > > version of the article. Thank you again for this insight.
> > >
> > > **We are happy to answer any additional questions that you may have. Thank you.**

---

> > > > ### Comment · Reviewer_t9NA · 2023-08-17
> > > > **Thank the authors' response**
> > > >
> > > > Yes, the authors gave good response to my three questions. There is no more confusion from my side. Good luck!

---

### Official Review · Reviewer_4uoY · 2023-07-05

**Soundness:** 2 fair
**Presentation:** 4 excellent
**Contribution:** 4 excellent
**Rating:** 7
**Confidence:** 3

**Summary:**

EDIT: I am changing my score based on the revision. I think this is a very interesting paper. In particular, it helps us understand why GNNs work well, but not super-duper well. CNNs get full NC, but GNNs don't

This work discusses neural collapse of GNNs. Much of the work on neural collapse focuses on unconstrained features which is somewhat in conflict with the constraints imposed by graph structure. The main contributions of this work are.


1. Empirical study showing that some decrease in within class variablility, but not to the same extent as in vanilla DNN

2. Graph based UFM. Prove that with optimistic model, need strict structural condition to get full NC.

3. Study gradient dynamics of graph-UFM that partially explains theoretical reaonsing for partial NC

4. Compare NC1 metrics of GNNs vs spectral clustering

Overall, this is a very good paper. It shows that GNNs exhibit partial, but not total neural collapse. In some sense, this may help explain why GNNs are near state of the art for many tasks, but nowhere near "perfect". However, I believe there is an error in the proof of theorem 3.2. I believe this error is fixable, but that the statement of the theorem will have to change. Therefore, I am strongly opposed to acceptance before this issue is fixed.

I should also mention that I did not have time to check the last two proofs as thoroughly as I would like. The authors should make sure to check them very carefully before resubmission since there was already one mistake. (I will check aggressively then.)

If these issues are addressed, I would likely be in favor of acceptance.

**Strengths:**

This paper contributes many new ideas to understand NC in the graph setting. Much of which are highly-nontrivial extensions of the original NC such as providing a characterization of when NC occurs and showing that the loss function will decrease, but level off before zero.

**Weaknesses:**

Major Issues

In the proof of Theorem 3.1, it is not clear what $y_c$ is. This needs to be made more clear.

In Section D.2, you should recall the definitions of the different $\Sigma$s or at least reference where they are defined. It is difficult to keep track of the many definitions.

I don't understand the inequality of the form $P(A|B)\leq P(A)$ given in 444. I also don't think you need it to obtain equation (50) which I think is correct. Could you please revise or clarify? (I am not overly concerned since the downstream equation appears to me to be correct. However, there is a similar issue for the diagonal terms which I think is more serious.)

The bound obtained in Theorem 3.2 is hard to parse an it is not obvious to me whether or not this quanity tends to zero. Perhaps you can approximate via sterlings inequality or something? Footnote 2 is unconvnincing since the terms (and the number of terms) change as n increases. I think you can argue (informally) that for large $n$ the probability that a Binomial RC is exactly equal to its mean, is equivalent to the probability that a normal RV with std $n^{-1/2}$ is within $1/n$ of its mean. Interpretation the standard deviation as average difference between the sample and the mean, this means that the probability of this occuring is on the order of $(1/n)/(1/\sqrt{n})$ which then tends to zero as $n\rightarrow \infty$.

(THIS IS THE BIG ONE): I don't think the argument used for the diagonal blocks is the proof of Theorem 3.2 regarding the upper bound for conditional probabilities is correct. For example, Set $t_{cc}=n-1$ and consider the probability of $E^c_{c,i}$ occurring given that $E^c_{c,i}$ for all $i<n$. Given that every other vertex is connected to $n-1$ other vertices of class $c$, this means that the set of all vertices in class $c$ form a clique. Therefore the probability of $E^c_{c,n}$ occurring is one. THIS ISSUE MUST BE FIXED IN ORDER FOR ME TO RECOMMEND ACCEPTANCE. Possible remedies include (i) convince me that I am mistaken (which is possible) (ii) remove the terms corresponding to the diagonal from the statement of the Lemma (in this case you should verify that this still converges to zero) (iii) add the assumptions that we are in the sparse regime to the statement of the lemma. Argue that cliques (or dense subgraphs in general) are exceptionally unlikely in this regime and that for reasonable values of $t$ the events are ``close to being independent." This is likely the best solution, but it would also require a bunch of non-trivial computations in order to make it precise (and the statement of the theorem would still need to change to reflect what you gt in these estimates) (iv) the event that the diagonal blocks satisfy the condition you need is equivalent to an erdos renyi random graph being regular. There is probably a bunch of existing work on this topic.

Minor Issues:

In the summation on line 121, adding a subscript j to the summation would make things more clear, i.e., $j \in \mathcal{N}_c(v_c,i)$

The capitlizatin is off in some of the refreeneds, in Euclidean shoudl be capitalized in ref [7]. Please fix.

Line 344 in the proof of Theorem 3.1: ``Theorem 3.1" should be capitlalized.

It might be a good idea to first explain the idea of the proof of theorem 3.2, i.e. that first you show that in the limit, every vertex in class c has the same degree. Therefore, the stated condition that all of the vertices in $v_i$ have the same \textit{fraction} or neighbors in each class $c'$ is equivalent to them having the same \textit{number} or neighbors in $c'$ (which can then be estimated more easily using formulae for binomial RVs)

Line 419 in the supplement. I think you should remove the word essentially, right? The number of neigbors IS a binomial RV.



**Questions:**


\subsection{Questions}
1. Line 44, when you say that the rules become similar to the nearest class center in the feature space do you mean the classifier is effectively an NN-classifier?

2. Why is there a $y_k$ in (1), but $y_k(V_k)$ in (2)? This seems inconsistent.

3. In the ``simpler" model (4), is the definition of $\hat{A}$ the same as in (3)? In this setting it would make sense to add self loops as in Kipf and Welling, i.e., $\hat{A}=(A+I)(D+I)^{-1}$. Otherwise, nodes don't communicate with themselves at all. Am I missing something?

4. For the sake of self-containedness, could you please briefly explain the instance-normalizaiton?

5. How important is the assumption of balanced communities in your theoretical analysis and how does this affect the applicablity of your theory to real-world imbalanced datasets

6. Hidden feature is the same size in all layers. How does this affect the results? What happens if you first increase and then decrease (or something) is there still partial NC1?

7. In the Graph UFM, if the $H_k$ are freely optimizable, why do you still need seperate matrices $W_2$ and $H_k$?v (This may be a standard thing. I am not very familiar with UFMs)

8. For the stochasitc block models, shouldn't there be some sort of concentration of measure result where for large $n$ we have $A\approx EA$? This should allow you to apply theorem 3.3 to it, right?

9. In Theorem 3.3, you obtain that $\tilde{NC}_1$ decreases which means it converges to some non-zero limit. Is there a good way to think about what this limit is? Does it, for example go to zero in the case that $E\rightarrow 0$? How does it depend on $\alpha?$

10. In line 427 of the proof of theorem 3.2, I think you need probability $1-4n^{-r}$ because of the union bound. (This is unimportant but should be fixed.)

---

> ### Author Rebuttal · Authors · 2023-08-08
>
> **General note:** We sincerely thank **Reviewer 4uoY** for the detailed feedback. Due to character limits, please find the responses to all the key questions below. All the minor issues will be fixed in the revision.
>
> **Q: I don't understand the inequality of the form $P(A|B) < P(A)$ given in 444. I also don't think you need it. Could you please revise or clarify?...**
>
> **A:** **We agree** with the reviewer that this inequality is not needed for the analysis of the off-diagonal block (ln 444). We will remove the statement corresponding to this inequality to avoid confusion.
>
> **Q: The bound obtained in Theorem 3.2 is hard to parse and it is not obvious to me whether or not this quantity tends to zero...**
>
> **A:** The term corresponding to the "off-diagonal blocks" in Theorem 3.2 is given by:
>  \begin{align}
>      \left( \sum_{t=0}^n \bigg[{n \choose t}q^{t}(1-q)^{n - t}\bigg]^n\right)^{\frac{C(C-1)}{2}}
>  \end{align}
> Here, observe that the binomial expansion of $(q + 1-q)^n$ is given by:
>  \begin{align}
>      1 = (q + 1-q)^n = \sum_{t=0}^n {n \choose t}q^{t}(1-q)^{n - t},
>  \end{align}
>  where each term, say $f(t) = {n \choose t}q^{t}(1-q)^{n - t}$ in the sum is strictly less than 1. Also, note that:
>  \begin{align}
>      1^n = \left(\sum_{t=0}^n f(t) \right)^n
>       &= \sum_{t=0}^n f(t)^n + \sum_{k_0+ \cdots + k_n = n, 0 \le k_0, \cdots, k_n < n} {n \choose k_0,\cdots,k_n} \prod_{t=0}^n f(t)^{k_t}.
>  \end{align} This gives us:
>  \begin{align}
>      \sum_{t=0}^n f(t)^n = 1 - \sum_{k_0+ \cdots + k_n = n, 0 \le k_0, \cdots, k_n < n} {n \choose k_0,\cdots,k_n} \prod_{t=0}^n f(t)^{k_t}.
>  \end{align}
>  As $n$ increases, it is true that the terms and the (number of terms) change in LHS but recall that $\sum_{t=0}^n f(t) = 1$. As the maximum value that $f(t)$ can take is $<1$, it tends to zero after taking the $n^{th}$ power, as $n$ grows larger. Thus $\mathbb{P}(\mathcal{G} \text{ obeys \textbf{C}} )$ is negligible as $N >> C$.
>
> **Q: (THIS IS THE BIG ONE): I don't think the argument used for the diagonal blocks is the proof of Theorem 3.2 regarding the upper bound for conditional probabilities is correct...**
>
> **A:** Thanks for pointing it out. To recall, the purpose of theorem 3.2 is to show that SSBM graphs rarely satisfy condition C. After exploring the insightful directions provided by the reviewer, we feel that just considering the off-diagonal blocks **(suggestion (ii) by reviewer)** should be sufficient to convey the idea in a simple manner, as we have shown (in the above response) that even this probability tends to 0.
>
> **(Revised) Theorem 3.2:** Let $\mathcal{G}=(\mathcal{V}, \mathcal{E})$ be drawn from SSBM$(N, C, p, q)$. For $N >> C$, we have
> \begin{align}
> \mathbb{P}\left(\mathcal{G} \text{ obeys \textbf{C}}  \right) < \left( \sum_{t=0}^n \bigg[{n \choose t}q^{t}(1-q)^{n - t}\bigg]^n\right)^{\frac{C(C-1)}{2}}.
> \end{align} which converges to $0$ for large $n$ as shown above. Numerically, when $C=2, N = 1000, q=0.0017$, we get $\mathbb{P}(\mathcal{G} \text{ obeys \textbf{C}} ) < 2.18 \times 10^{-188}$, which is practically zero.
>
> **A note on suggestion (iii):** We found that this line of analysis is quite non-trivial. Especially, formalizing the "almost independent" notion of the events: $\{ E_{c,1}^c (t), E_{c,2}^c (t), \cdots, E_{c,n}^c (t)\}$ is not well justified as $\prod_{i=1}^n P(E_{c,i}^c(t)) = \left[ {n \choose t}p^t(1-p)^{n-t} \right]^n$ tends to smaller values [1] for any $t$ as $n$ increases (shown in the above response).
>
> **Regularity of erdos-renyi graphs:** We explored the literature on the existence of regular graphs on $n$ nodes and found that even asymptotic results for $p \propto \frac{\log N}{N}$ settings seem to be not well explored [2]. Also, the existing results for degree $O(\sqrt{n}), c/n$ etc are extremely non-trivial for the message we want to convey in this theorem.
>
> [1] Dykstra, R. L., et.al "Events which are almost independent." The Annals of Statistics 1.4 (1973)
>
> [2] Wormald, Nicholas C. "Models of random regular graphs."
>
> **Q: It would make sense to add self-loops, otherwise, nodes don't communicate with themselves at all...?**
>
> **A:** We have not used $\hat{A} = (A + I)(D + I)^{-1}$ explicitly as the nodes of the sampled SSBM graphs are allowed to have self-edges. We will explicitly mention this in the revision to avoid confusion.
>
> **Q: Importance of balanced communities and practical relevance.**
>
> **A:** This assumption is critical as we extensively leverage the properties of Kronecker products in most of our proofs. In the case of GNNs, since we show partial collapse even in the ideal scenario of balanced communities, specific "rewiring" mechanisms for the computational graph might be needed in practical settings (for ex: to satisfy condition C with our GNN design) to aid the optimizers. Additionally, a promising real-world application is a mitigation of over-smoothing by such rewiring mechanisms as NC1 indicates a maximal separation between class-mean features.
>
> **Q: Role of varying hidden dimensions.**
>
> **A:** Scenario 1: first 16 layers: dim = 8 + next 16 layers: dim = 16, Scenario 2:  first 16 layers: dim = 8 + next 16 layers: dim=4. Final layer dim = 2 for both scenarios. We observed that both the scenarios exhibited partial NC but the "collapse" in scenario 2 was relatively higher (i.e. relatively lower NC1 values).
>
> **Q: Intuition about the limit in Theorem 3.3 and role of $\alpha$.**
>
> **A:** Intuitively, one can think of this limit as a state at which the $\Sigma_W$ has reduced (sufficiently) and $\Sigma_B$ has increased (sufficiently) as the exact collapse is not the desired state for $A$ that is randomly sampled from SSBM. When $E \to 0$, then the resulting $\mathbb{E}A$ theoretically satisfies condition C and the gradient flow will tend towards the NC1 state, which indicates that $\widetilde{NC}_1$ will tend to 0. Finally, $\alpha$ can be thought of as the regularization needed for the trace of between-class covariance to increase along the flow.

---

### Author Rebuttal · Authors · 2023-08-08

**Our response to the general comments on motivation, theoretical modeling, practical significance, over-smoothing, and a fix to Theorem 3.2.**

**Modifying Theorem 3.2 to ignore the non-rigorous diagonal block case**

As pointed out by one of the reviewers, we have updated Theorem 3.2 to leverage only the off-diagonal blocks in the probability-bound calculation. The case for the diagonal block led to non-trivial assumptions and results which seemed to complicate the message of the theorem.

**The importance of neural collapse (NC).** In standard DNN settings (e.g., image classification on MNIST), the classifiers tend to exhibit NC once they perfectly classify the training data (i,e during TPT). NC includes both: "reduction" in the features’ within-class variability and "increase" (or stabilization) in the between-class variability, such that the features form certain low-dimensional geometric structures. This phenomenon has allowed the community to understand the benign effects of overparameterization and the benefits of training beyond the "zero-classification-error" stage for DNNs (e.g., cases with better test accuracy and improved adversarial robustness [1]). Importantly, we demonstrate that the case with GNNs is not similar to any previous work on NC as the structural constraints of graphs tend to hinder the ability of NC. Also, since the over-smoothing problem is common in deep GNNs, NC shows potential to address such key issues and have a promising impact.

**The data and network models.** To explore the level of NC in GNNs in a principled fashion, we start with SSBM graphs, as their properties are well-suited for theoretical GNN analysis [2]. The gUFM model presented in this paper is much more complicated than any theoretical model in previous publications on NC [4] as we retain the graph topology in the analysis. Importantly, even for our “optimistic" model, we show that the desired configuration of the penultimate layer features (in terms of optimization optimality) does not exhibit exact NC unless the graph satisfies a strict structural condition. This immediately implies an inherent difference between practical GNNs and DNNs. Also, note that it seems very difficult to derive the exact training dynamics in GNNs without trivial assumptions on the structure. Thus, we leverage the literature on SSBM graphs and present a rigorous gradient flow analysis to explain the partial collapse observed in our experiments. This analysis is already a major step beyond analyzing only the minimizers of the model.

**Significance.**
1. The results in Theorem 3.1 on the strict structural condition of the graph and the gradient flow analysis in Theorem 3.3 indicates that contrary to the case of DNNs where MSE loss leads to collapsed minimizers, the feature evolution in GNNs during training is leading only to partial collapse (even in the simplified/optimistic setting). This result sheds light on the impact of structural conditions on the ideal feature configurations of an expressive GNN.

2. Additionally, observe that attaining neural collapse solves the over-smoothing problem as neural collapse represents the maximal separation between feature class means in addition to zero within-class feature variability. This is indeed an important takeaway to the GNN community, which we will clarify in the revision.

3. A recent work by Ma et,al [3] showed that homophily might not be necessary for good GNN performance on node classification tasks. By leveraging Theorem 3.1 from our work, we can observe that the structural condition (dubbed condition C) required for collapsed minimizers addresses both the homophilic and heterophilic graphs. Thus, our results provide insight into these surprising empirical results.

4. Our work can potentially shed light on an ideal "graph-rewiring" strategy for improved GNN performance as we know the nature of the minimizers when condition C is satisfied by the computational graph.

5. We believe that extensions of our analysis to semi-supervised /self-supervised settings would be of great value to the community. For instance, the structural conditions that we analyze in this work can be used for graph augmentation in graph contrastive learning.

**Oversmoothing and neural collapse.**
Over-smoothing has been widely studied in the literature to model the reduction of within-class and between-class feature variability across layers of a GNN during training. On the contrary, NC studies the reduction in within-class but an "increase" in between-class feature variability (especially of the penultimate layer). Interestingly, note that when the penultimate layer features exhibit neural collapse, we can address the over-smoothing problem. In addition to section 3, which highlights the NC results for the penultimate layer during training, In section 4, we presented a layer-wise feature evolution in a well-trained GNN. This analysis considers well-trained GNNs in inductive settings for layer-wise analysis and inherently differs from the over-smoothing analysis of [2] which considers networks in the training phase of semi-supervised settings.

**Overall, we would like to thank the reviewers for their insightful discussions and for strengthening the quality of our paper.**

[1] Papyan, Vardan, et,al "Prevalence of neural collapse during the terminal phase of deep learning training." PNAS 2020

[2] Keriven, Nicolas. "Not too little, not too much: a theoretical analysis of graph (over) smoothing." NeurIPS (2022)

[3] Yao Ma, et al. Is homophily a necessity for graph neural networks? (ICLR), 2022

[4] Han, X. Y., et.al. "Neural Collapse Under MSE Loss: Proximity to and Dynamics on the Central Path." ICLR. 2021

---

### Decision · Program_Chairs · 2023-09-21

**Decision:**

Accept (poster)

**Comment:**

Most of the reviewers were highly impressed by the quality and significance of your research. We would like to highlight the key points that contributed to the positive evaluation of your paper:

1. Novel Contributions to Graph Neural Networks:
The reviewers recognized that your paper introduces a range of innovative ideas that significantly contribute to the understanding of Neural Collapse (NC) in the graph setting.

2. In-depth Analysis of Feature Evolution with GNNs:
Reviewers found your study of feature evolution with Graph Neural Networks (GNNs) intriguing, particularly the examination of the correlation between topological information and feature changes. The clear presentation of the data model are clear and intuitive, coupled with experiments that align with theoretical results

3. Valuable Theoretical Insights:
Your paper stood out for its  exploration of Neural Collapse in Graph Convolutional Networks (GCNs) and the forward process of neural networks. The reviewers found your theoretical results, including the unique and valuable Theorem 3.1 on partial collapse, to be particularly noteworthy. Such insights have the potential to inspire future research in the field of graph-based neural networks.

4. Clarity of Presentation and Methodology:
The reviewers appreciated the clarity of your paper's presentation. They highlighted the well-written nature of the manuscript and noted that the methodology is effectively explained, making it accessible to a wide audience.

Please incorporate all the reviewers' feedback in the final version.